# Vimentin filaments integrate low-complexity domains in a complex helical structure

Matthias Eibauer [1] ✉, Miriam S. Weber [1], Rafael Kronenberg-Tenga[1], Charlie T. Beales[1], Rajaa Boujemaa-Paterski [1], Yagmur Turgay [1,4], Suganya Sivagurunathan[2], Julia Kraxner[3,5], Sarah Köster[3], Robert D. Goldman[2] & Ohad Medalia [1] ✉

Intermediate filaments (IFs) are integral components of the cytoskeleton. They provide cells with tissue-specific mechanical properties and are involved in numerous cellular processes. Due to their intricate architecture, a 3D structure of IFs has remained elusive. Here we use cryo-focused ion-beam milling, cryo-electron microscopy and tomography to obtain a 3D structure of vimentin IFs (VIFs). VIFs assemble into a modular, intertwined and flexible helical structure of 40 α-helices in cross-section, organized into five protofibrils. Surprisingly, the intrinsically disordered head domains form a fiber in the lumen of VIFs, while the intrinsically disordered tails form lateral connections between the protofibrils. Our findings demonstrate how protein domains of low sequence complexity can complement well-folded protein domains to construct a biopolymer with striking mechanical strength and stretchability.

IFs are expressed in a cell-type-specific manner, providing the cytoskeleton with cell-type-specific viscoelastic properties[1]. In humans, IF proteins are encoded by 70 genes[2], including lamins[3], keratins[4] and neurofilaments[5], and are associated with a wide range of functions[6,7] and at least 72 human pathologies[2,8].

Vimentin is mainly expressed in cells of mesenchymal origin, in which it assembles into extensive, dynamic and hyperelastic filament networks[9,10]. Individual VIFs can be stretched up to 3.5 times their initial length[11,12], tolerating much higher strain and exhibiting greater flexibility than F-actin and microtubules[13]. VIFs are involved in various cytoskeletal processes, such as maintenance of cell shape[14,15], development of focal adhesions[16], protrusion of lamellipodia[17], assembly of stress fibers[18,19] and signal transduction[20]. Vimentin expression and assembly of VIFs are markers and regulators of the epithelial–mesenchymal transition[21]. They are involved in cancer initiation, progression and metastasis[22–24] and several other pathophysiological conditions[25–28].

At the molecular level, vimentin monomers interact to form in-register, parallel dimers[29] that are ~48 nm in length. Like all IF proteins[30], the dimers contain a central coiled-coil rod domain composed of consecutive α-helical segments, connected by flexible linkers. The rod domain is flanked by intrinsically disordered amino-terminal head and carboxy-terminal tail domains[31]. The early steps in VIF assembly involve the formation of elongated tetramers consisting of two antiparallel, staggered dimers[32–34] that are ~62 nm in length. The vimentin tetramer is the basic building block for subsequent formation of unit-length filaments and VIF assembly[35–37].

Although the tetramer is well characterized, solving the three-dimensional (3D) structure of fully assembled VIFs, and that of IFs in general, using conventional methodologies has proven to be a major challenge. This is likely attributable to the unique elongated shape of their tetrameric building blocks, which contain a substantial fraction of intrinsically disordered domains and flexible linkers. IFs also exhibit extensive structural polymorphism[38–40]. Therefore, the 3D structure of a fully assembled IF has remained elusive[41].

Here we used cryo-focused ion-beam (cryo-FIB) milling and cryo-electron tomography (cryo-ET) to study VIFs in situ[42–44]. We

[1]Department of Biochemistry, University of Zurich, Zurich, Switzerland. [2]Department of Cell and Developmental Biology, Northwestern University Feinberg School of Medicine, Chicago, IL, USA. [3]Institute for X-Ray Physics, University of Göttingen, Göttingen, Germany. [4]Present address: Department of Health Sciences and Technology, ETH Zurich, Zurich, Switzerland. [5]Present address: MDC Berlin-Buch, Max-Delbrück-Centrum für Molekulare Medizin, Berlin, Germany. ✉e-mail: m.eibauer@bioc.uzh.ch; omedalia@bioc.uzh.ch

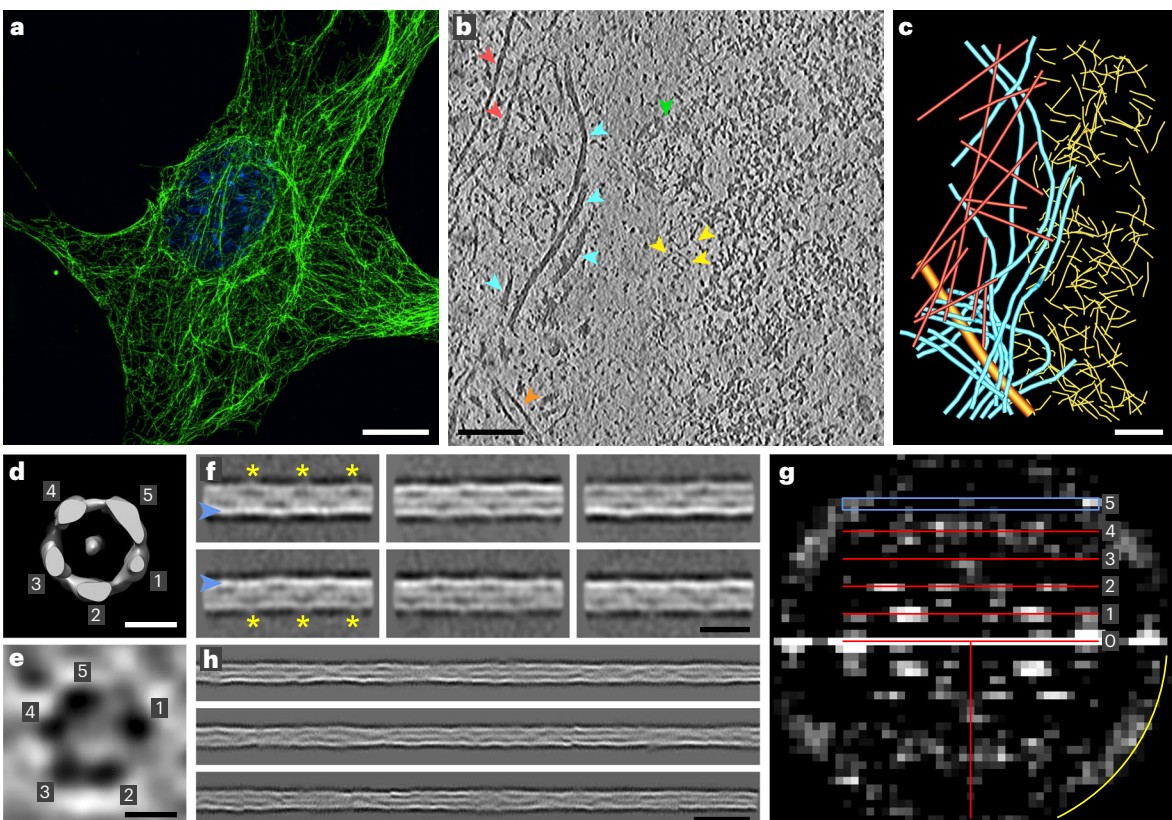

**Fig. 1 | VIFs are built from five protofibrils in cells. a**, Slice through a 3D-SIM image (*n* = 20) of a MEF, fixed and stained with anti-vimentin (green); the nucleus is stained with DAPI (blue). The VIF network extends over the whole cellular volume, with regions of lower and higher network density, and forms a cage around the nucleus. Scale bar, 10 μm. **b**, Slice (8.84 Å thick) through a tomogram of a cryo-FIB-milled MEF (*n* = 102), recorded in a region around the nuclear envelope. VIFs, cyan arrowheads; actin filaments, red arrowheads; lamin filaments, yellow arrowheads; microtubule, orange arrowhead; nuclear pore complex, green arrowhead. Scale bar, 100 nm. **c**, Segmentation (*n* = 1) of the biopolymers present in **b**. Scale bar, 100 nm. **d**, Averaging of 2,371 subtomograms showing that VIFs are assembled from five protofibrils in situ. Scale bar, 5 nm. **e**, Cross-section of a VIF (*n* = 150), 8.84 Å thick, extracted from the *xy* plane of a denoised tomogram, confirming the five-protofibril architecture of VIFs.

Scale bar, 5 nm. **f**, Class averages of VIFs (*n* = 8) show a helical pattern with a repeat distance of ~180 Å (repeating feature marked with yellow asterisks); one side of the filament boundary appears pronounced in projection (marked with blue arrowheads). Scale bar, 180 Å. **g**, Combined power spectrum (*n* = 1) of class averages, as shown in **f**. The presence of layer lines confirms the helical architecture of VIFs. The first layer line appears at ~1/185 Å and reflects the repeating pattern observed in the class averages. The layer line and peak distribution with a meridional reflection on the fifth layer line at ~1/37 Å is compatible with a helical assembly of five subunits per repeat. The yellow arc in the power spectrum indicates 1/26 Å. **h**, Gallery of three computationally assembled VIFs (*n* = 5,205). With this technique, the progression of VIFs can be followed with an improved signal-to-noise ratio over a substantial length. Scale bar, 35 nm.

analyzed data from mouse embryonic fibroblasts (MEFs) using subtomogram averaging[45–47] and averaging of VIF cross-section views[4]. In addition, we acquired tomograms of detergent-treated MEFs and found that VIFs have helical symmetry. Using single-particle cryo-electron microscopy (cryo-EM), we verified the helical-symmetry parameters of the filaments and obtained 3D structures of VIFs from human full-length and tail-less vimentin proteins that had been polymerized in vitro. On the basis of these structures, we constructed an atomic model of VIFs. This model shows a unique molecular architecture comprising a complex helical scaffold of intertwined vimentin tetramers that form protofibrils, which surround a central fiber composed of the low-complexity head domains of vimentin.

## Results

### Polymerization state in situ

MEFs are widely used to investigate the cellular organization and function of VIFs (Fig. 1a). To study VIFs in their native cellular environment, we cultured MEFs on electron microscopy grids, vitrified the cells without chemical fixation, prepared thin sections of the MEFs by cryo-FIB milling and imaged the resulting lamellas by cryo-ET (Fig. 1b and Supplementary Videos 1 and 2). A unique advantage of this approach is that it completely retains the molecular organization of the sample[48].

In regions around the nuclear envelope, the resulting tomograms show the complete set of filamentous biopolymers present in MEFs (Fig. 1c), including F-actin, microtubules and nuclear lamins[3]. VIFs can be clearly distinguished on the basis of their tubular shape and have a diameter of ~11 nm.

Contrary to expectations based on VIFs assembled in vitro[32,49], subtomogram averaging shows that VIFs are built from five protofibrils in situ (Fig. 1d, Table 1, Extended Data Fig. 1 and Supplementary Fig. 1). We also determined the protofibril stoichiometry in cross-section images of VIFs (Fig. 1e and Extended Data Figs. 2 and 3), confirming their five-protofibril architecture.

### Determination of helical symmetry

To verify the pentameric cross-sections of VIFs, MEFs were cultured on electron microscopy grids as described above, but the cells were treated with a permeabilization procedure that preserves the VIF network (Supplementary Fig. 2a), and soluble components were depleted from the cytoplasm and the nucleus[3]. Subsequently, the detergent-treated MEFs were vitrified and imaged by cryo-ET (Table 1 and Supplementary Fig. 2b and Supplementary Video 3). This sample preparation results in thinned MEFs; therefore, VIFs from different *z* heights are concentrated in the tomograms. This procedure allows for an extensive

**Table 1 | Data collection and validation statistics**

| | Cryo-FIB and cryo-ET in situ analysis of VIFs (EMD-19562) | Cryo-ET of detergent-treated MEFs | Cryo-EM of in-vitro-polymerized human VIFs (EMD-16844) | Cryo-EM of in-vitro-polymerized human VIFs-ΔT (EMD-19563) |
|---|---|---|---|---|
| **Data collection and processing** | | | | |
| Magnification | ×64,000 | ×42,000 | ×130,000 | ×130,000 |
| Voltage (kV) | 300 | 300 | 300 | 300 |
| Electron exposure (e⁻/Å²) | 160 | 125 | 62 | 62 |
| Electron dose per tilt image (e⁻/Å²) | 3.9 | 2.1 | – | – |
| Defocus range (μm) | –4 | –2 to –6 | –0.8 to –2.8 | –0.8 to –2.8 |
| Pixel size (Å) | 2.21 | 3.44 | 0.68 | 0.68 |
| Energy filter slit width (eV) | 20 | 20 | 20 | 20 |
| Tilt range (°), tilt increment (°) | –60°/60°, 3° | –60°/60°, 2° | 0° | 0° |
| Symmetry imposed | Helical | – | Helical | Helical |
| Helical rise (Å) / helical twist (°) | 42 / 72 | | 42.5 / 73.7 | 42.5 / 73.7 |
| Number of tomograms for cryo-ET or micrographs for cryo-EM | 102 | 225 | 12,160 | 19,534 |
| Initial subtomograms/particle images (no.) | 14,273 | 1,148,072 | 1,462,717 | 1,019,393 |
| Final subtomograms/particle images (no.) | 2,371 | 615,106 | 236,920 | 143,270 |
| Map resolution (Å) | 20.2 | – | 7.2 | 8.4 |
| FSC threshold | 0.143 | – | 0.143 | 0.143 |
| Map resolution range (Å) | 18–26 | – | 3–14 | 3–14 |

statistical analysis of the filaments in a fraction of the time required for the cryo-FIB approach.

In total, we extracted ~390,000 overlapping, 65 nm extended segments of VIFs from the tomograms and subjected the data to two-dimensional (2D) classification[47,50]. The class averages reveal a clear periodic pattern (Fig. 1f, yellow asterisks) with a repeat distance of ~180 Å (Supplementary Fig. 2c), as measured by autocorrelation analysis[51]. Additionally, class averages show that the projection of one filament wall is more pronounced than is that of its counterpart (Fig. 1f, blue arrowheads), further supporting the notion that VIFs comprise an odd number of protofibrils (Supplementary Fig. 2d).

Subsequently, we combined the periodic class averages into a power spectrum, which revealed a pattern of layer lines (Fig. 1g). This is direct proof that VIFs have helical symmetry[51]. The first layer line is positioned at ~1/185 Å, which corroborates the previous autocorrelation measurement (Supplementary Fig. 2c). Moreover, the Bessel peak distribution along the layer-line spectrum is compatible with a helical assembly of five building blocks in one helical pitch. Consequently, on the fifth layer line, positioned at ~1/37 Å (Fig. 1g, blue rectangle), a meridional peak can be detected. Therefore, we conclude that the unique pentameric cross-sections of VIFs, as observed in the in situ cryo-FIB and cryo-ET data, are formed by the specific helical symmetry of VIFs. This was independently verified using cryo-ET data obtained from the detergent-treated MEF preparations.

## Computational filament assembly

To improve the precision of the helical-symmetry measurement and to allow for higher-resolution analysis, we computationally assembled extended stretches of VIFs (Fig. 1h and Supplementary Fig. 3)[4]. Combining these long-range VIF observations into a single power spectrum improved the resolution of the layer lines substantial. As a result, the observed layer lines (Fig. 1g) decompose at high resolution into bundles of fine layer lines (Extended Data Fig. 4c). Comprehensive power-spectrum analysis yields a helical rise of 42.5 Å and a helical twist of 73.7° (Extended Data Fig. 4).

This analysis also indicates that VIFs are assembled from 4.88 tetramers, packed into a helical pitch with a length of 207.4 Å. This result recapitulates a characteristic feature of IFs, namely their axial periodicity of ~21 nm, which has been observed using rotary-shadowing techniques[1,30,52–54].

Because the mass per length of a helical filament is proportional to its helical rise[55], the observed VIFs have a mass per length of 50.5 kDa nm⁻¹, in agreement with data obtained by scanning transmission electron microscopy[38].

## Structure of VIFs

We next aimed to solve the 3D structure of fully assembled VIFs. To this end, we polymerized VIFs in vitro from bacterially expressed and purified human vimentin, and imaged the filaments using cryo-EM (Table 1 and Supplementary Fig. 4). We extracted ~1.5 million VIF segments from the data set, which were subjected to extensive 2D and 3D classifications (Supplementary Fig. 5)[56,57]. These calculations independently converged to the identical (up to two decimal places) helical parameter (Supplementary Fig. 5b,c), as measured by power spectrum analysis of mouse VIFs (see above).

The resulting 3D structure of VIFs reached a resolution of 7.2 Å (Supplementary Fig. 6). The helical filament is composed of five protofibrils (Fig. 2a and Supplementary Video 4), connected laterally by individual contact sites (Extended Data Fig. 5a,b). Surprisingly, the protofibrils are also connected through a fiber in the VIF lumen (Fig. 2b,c).

The repeating unit of a protofibril is assembled from three regions (Fig. 2d). The first is a relatively straight, tetrameric α-helix bundle (Fig. 2d, red region) that is ~21 nm in length and slightly tilted with respect to the helical axis. This region forms large parts of the outer surface of the filament. Furthermore, it is the least flexible part of the protofibril (Fig. 2e). The second region is the more curved tetrameric α-helix bundle (Fig. 2d, blue region), which is attached laterally to the center of the first bundle. Its ends extend below the first bundle and form large parts of the inner surface of the filament, facing the lumen.

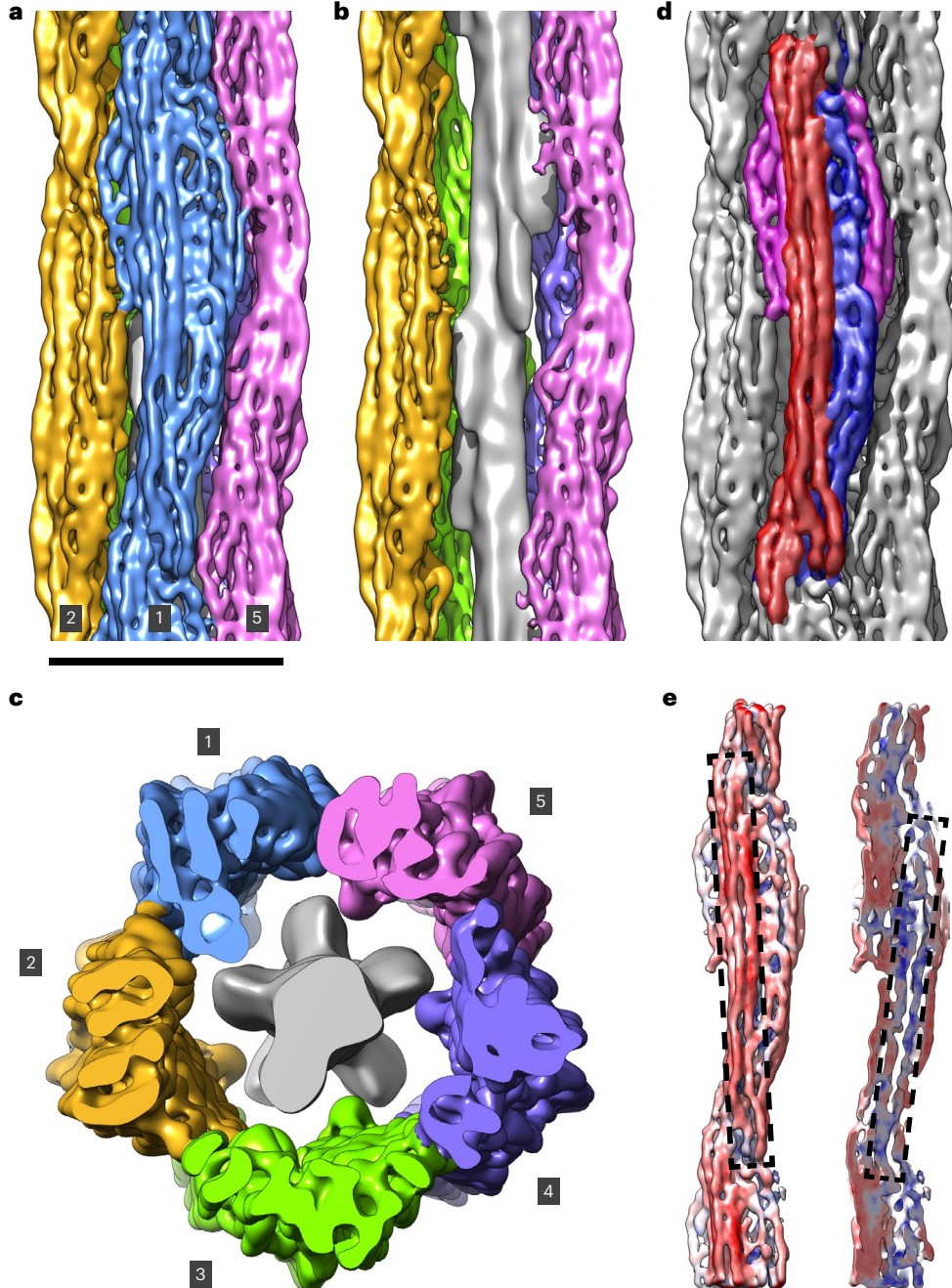

**Fig. 2 | The 3D structure of VIFs. a**, Isosurface rendering of the VIF 3D structure. The protofibrils are shown in different colors. Protofibrils 1, 2 and 5 are labelled. Scale bar, 10 nm. **b**, Omitting the front protofibril opens the view of the luminal fiber (gray density), which is located in the lumen of the VIFs. **c**, Cross-section of VIFs displaying all five protofibrils and the luminal fiber. **d**, Segmentation of the repeating unit of a protofibril. The frontal, straight tetrameric α-helix bundle is shown in red, the lateral, curved tetrameric α-helix bundle in blue, and the contact sites between adjacent protofibrils in magenta. **e**, The least flexible region of the repeating unit, shown in reddish colors, is the frontal, straight region (left dashed rectangle). A section view through the structure of the repeating unit shows that structural plasticity is increased mostly in the lateral, curved region (right dashed rectangle), shown in bluish colors.

Therefore, the central part of the repeating unit of a VIF protofibril is an octameric complex comprising eight α-helices. Finally, the third region is split between both sides of the repeating unit and establishes lateral contact sites between adjacent protofibrils (Fig. 2d, magenta regions).

### Position of the tail domains
To localize the tail domains in the cryo-EM density map, we polymerized recombinantly expressed human vimentin lacking the tail domain[38] and applied the above cryo-EM workflow to the tail-less VIFs. Surprisingly, the resulting VIF-ΔT structure clearly lacks contact sites between the

protofibrils (Extended Data Fig. 5c,d), but the octameric repeating unit and the luminal fiber are of similar volume to that of their counterparts in the full-length VIF structure. On the basis of this result, we conclude that the tail domains of vimentin, despite their low sequence complexity, can condense into a stable structure in VIFs and establish connections between the protofibrils[58].

### Atomic model of VIFs
To build an atomic model of VIFs based on our electron density map, we combined AlphaFold structure predictions[59] and molecular dynamics

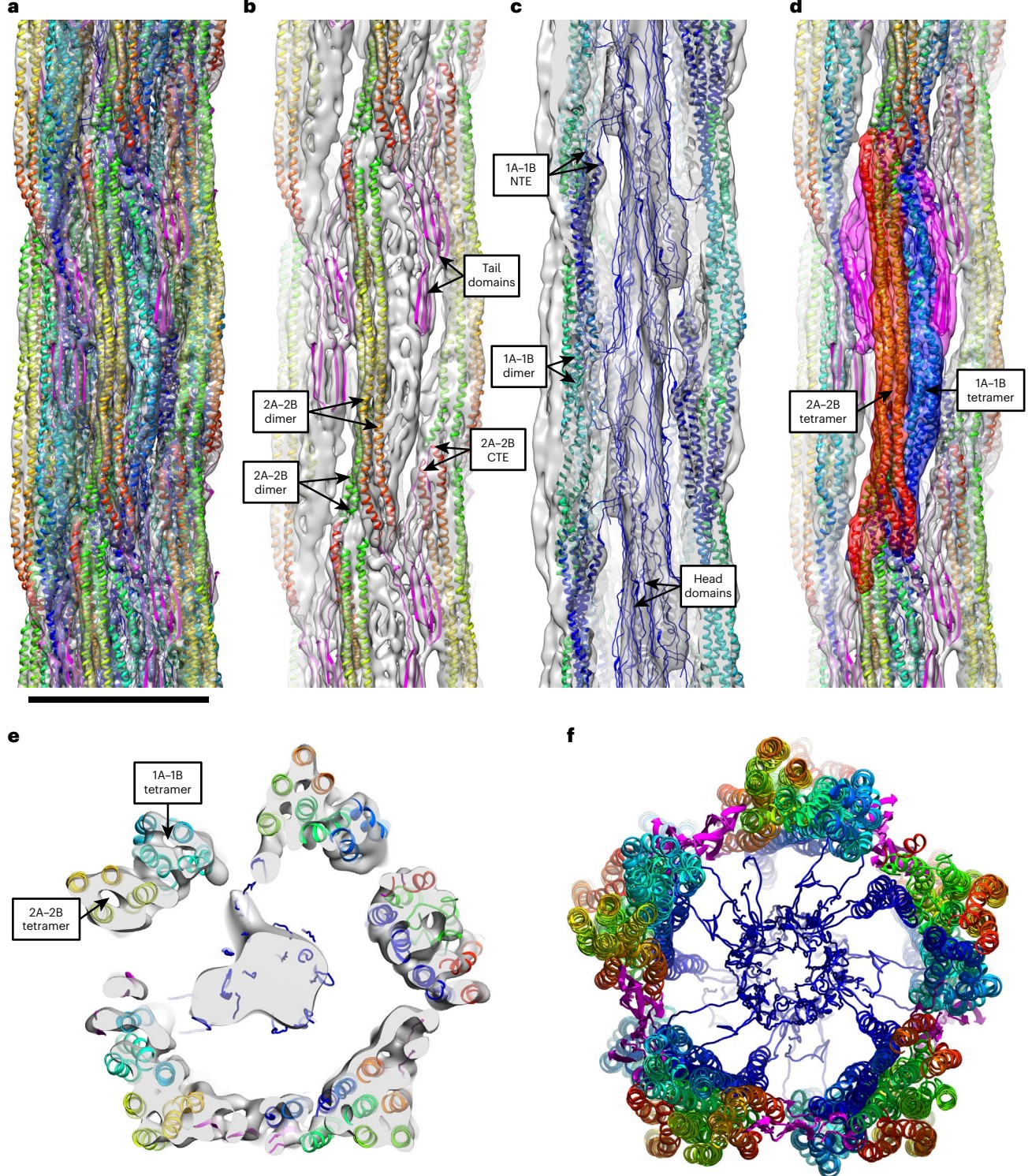

**Fig. 3 | Atomic model of VIFs. a**, Complete VIF 3D atomic model docked into the density map (transparent gray). Scale bar, 10 nm. **b**, The 2A–2B domains (α-helices colored from green (NTE) to red (CTE), and the tail domains (magenta chains) are shown within the density map. Two antiparallel 2A–2B dimers form a straight α-helix bundle, which constitutes about one half of a protofibril. The CTEs of each 2A–2B dimer position the tail domains to form the contact sites between the protofibrils. **c**, The 1A–1B domains (α-helices colored from blue (NTE) to green (CTE)), and the head domains (blue chains) are shown within the density map. Two antiparallel 1A–1B dimers compose a curved α-helix bundle, which approximately constitutes the second half of a protofibril. In the image, the frontal protofibril is omitted to reveal the inside of the filament. The NTEs

of each 1A–1B dimer protrude into the filament lumen, where the head domains aggregate to form the luminal fiber. **d**, The frontal straight (red transparent density) and lateral curved (blue transparent density) tetrameric regions in the electron density map are formed from an antiparallel 2A–2B tetramer and an antiparallel 1A–1B tetramer, respectively, and the contact sites between the protofibrils are formed by the tail domains (magenta transparent density). **e**, In VIFs, there are 40 polypeptide chains in cross-section, assembled into five protofibrils. **f**, The protofibrils interact laterally through the tail domains and centrally through the head domains. The 2A–2B dimers substantially shape the outer surface of VIFs, and the 1A–1B dimers predominantly coat their inner surface.

flexible fitting[60,61], and used reported cross-linking mass spectrometry data as distance restraints[62–64]. As an additional restraint, we incorporated the position of the tail domains into model building. The definition and initial folds of the vimentin domains, namely the 1A–1B (residues 86–253) and 2A–2B (265–411) domains, the linker L12 (254–264) and the head (1–85) and tail domains (412–466), were based on the AlphaFold prediction of the full-length vimentin dimer (Extended Data Fig. 6 and Supplementary Fig. 7)[31]. The initial folds were adapted to the cryo-EM density map by molecular dynamics flexible fitting.

The resulting 3D atomic model of VIFs (Fig. 3a, Table 2, Extended Data Fig. 7 and Supplementary Video 5) describes the cryo-EM density map in its entirety. The frontal, straight tetrameric α-helix bundle (Fig. 2d, red region) accommodates two antiparallel 2A–2B dimers (Fig. 3b and Extended Data Fig. 7, green to red colored α-helices). The C-terminal ends (CTEs) of each 2A–2B dimer position the tail domains to form the contact sites between the protofibrils (Fig. 3b and Extended Data Fig. 7, magenta chains), which contain a β-hairpin motif in the model (Extended Data Fig. 6).

The lateral, curved tetrameric α-helix bundle (Fig. 2d, blue region) accommodates two antiparallel 1A–1B dimers (Fig. 3c and Extended Data Fig. 7, blue to green colored α-helices). The N-terminal ends (NTEs) of each 1A–1B dimer position the head domains to protrude into the filament lumen, where they interact to form the luminal fiber (Fig. 3c and Extended Data Fig. 7, blue chains). This finding is in agreement with previous work showing that the isolated, low-complexity head domains of vimentin, and those of other types of IFs such as neurofilaments, self-assemble into amyloid-like fibers, which are stabilized by characteristic cross-β interactions, as detected by X-ray diffraction[65].

The frontal straight and lateral curved tetrameric regions in the electron density map are formed from two antiparallel 2A–2B dimers and two antiparallel 1A–1B dimers, respectively, and the contact sites between the protofibrils are formed by the tail domains (Fig. 3d). The cross-section view shows that VIFs comprise 40 polypeptide chains (Fig. 3e and Supplementary Video 6), which are partitioned into five octameric protofibrils. The protofibrils are connected laterally by interactions of the tail domains and connected centrally by interactions of the head domains (Fig. 3f). The 2A–2B dimers substantially shape the outer VIF surface, whereas the 1A–1B dimers predominantly coat the inner surface of the filaments (Extended Data Fig. 8 and Supplementary Video 7).

## Modular assembly of VIFs

The vimentin tetramer (Fig. 4a and Table 2) is the underlying asymmetric unit of the helical structure of VIFs. In general, the tetramer is straight, extending ~65 nm between CTEs of the flanking 2A–2B dimers. In particular, the α-helices of the 2A–2B dimers exhibit a mostly parallel geometry over a length of ~21 nm while rotating ~65° with respect to each other. By contrast, the four α-helices forming the central 1A–1B region exhibit a more pronounced coiled-coil geometry. This section is slightly curved, so the NTEs of the 1A–1B dimers protrude out of the plane of the tetramer and are positioned ~29 nm apart in pairs. The head domains extend parallel to the tetramer (within the luminal fiber in the mature filament). Our tetramer model predicts that the tail domains can exist in two conformations, either extending parallel to the long axis of the tetramer or folding back on one of the 2A–2B dimers. The possible range of motion of the tail domains is ~10 nm.

The full assembly of a protofibril, with eight polypeptide chains in the cross-section view, requires the sequential interaction of at least three tetramers (Fig. 4b and Extended Data Fig. 9). Starting from the first tetramer, t₁, a second tetramer, t₂, attaches with its 1A–1B section laterally to one of the flanking 2A–2B dimers of t₁. This creates an intermediate assembly, with six chains in cross-section. Subsequently, a third tetramer, t₃, binds to either side of this assembly. This creates a minimal-length protofibril (~110 nm between CTEs of the flanking 2A–2B dimers) with its basic repeating unit (eight chains in

**Table 2 | Refinement statistics**

|  | VIF model (PDB: 8RVE) | VIF tetramer model (PDBDEV-00000212) |
|---|---|---|
| **Refinement** | | |
| Model resolution (Å) | 7.9 | 7.9 |
| FSC threshold | 0.143 | 0.143 |
| Model resolution range (Å) | – | – |
| Map sharpening B factor (Å²) | Local B-factor sharpening | Local B-factor sharpening |
| **Model composition** | | |
| Chains | 78 | 4 |
| Non-hydrogen atoms | 54,788 | 7,456 |
| Protein residues | 13,697 | 1,864 |
| **B factors (Å²)** | | |
| Protein | – | – |
| **R.m.s. deviations** | | |
| Bond lengths (Å) | 0.017 | 0.017 |
| Bond angles (°) | 2.583 | 2.594 |
| **Validation** | | |
| MolProbity score | 2.59 | 1.52 |
| Clashscore | 30.1 | 1.49 |
| Poor rotamers (%) | 0 | 0 |
| **Ramachandran plot** | | |
| Favored (%) | 86.85 | 86.68 |
| Allowed (%) | 10.28 | 10.41 |
| Disallowed (%) | 2.87 | 2.91 |
| **Rama-Z** | | |
| whole | −4.58 | −4.61 |
| helix | −2.28 | −2.29 |
| sheet | −6.36 | −6.36 |
| loop | −4.43 | −4.43 |

cross-section) fully assembled in the center (Fig. 4b, cross-sections). The protofibril assembly is based solely on A₁₁ tetramers[62,66]. In this tetramer-binding mode, the two vimentin dimers are in an antiparallel arrangement, and their 1B domains are aligned with each other (Fig. 4a); this is the prevalent oligomeric species of vimentin in solution[67]. Along a protofibril, the 1A–1B region (Fig. 4b, blue helices) is constructed from successive A₁₁ tetramers. Therefore, the 1A–1B region is formed from two antiparallel 1A–1B dimers (Fig. 4b, cross-sections). The 2A–2B region (Fig. 4b, red helices) is constructed from two antiparallel 2A–2B dimers (Fig. 4b, cross-sections), but each 2A–2B dimer belongs to one of the two neighboring A₁₁ tetramers (Extended Data Fig. 9 and Supplementary Video 8).

The construction of an interlocking mechanism in VIFs is intriguing. The 1A (residues 100–125) and 2B (380–411) domains are highly conserved among the numerous types of IFs (Supplementary Fig. 8)[29,54]. Although they are distributed over the full length of the tetramer, in the fully formed protofibril they are brought into close proximity (Fig. 4, green colored α-helical segments), promoting interactions and interlocking successive tetramers together. This process is facilitated by the flexibility of the L12 linker domain, which allows the 2A–2B dimers to align parallel to the 1A–1B sections and eventually find the right position to form the interlocking region between the 1A and 2B domains.

In the fully assembled repeating unit of a protofibril (Fig. 4c), both 2A–2B antiparallel dimers are laterally aligned with the 1A–1B section, and both interlocked regions are fully formed ~21 nm apart on either

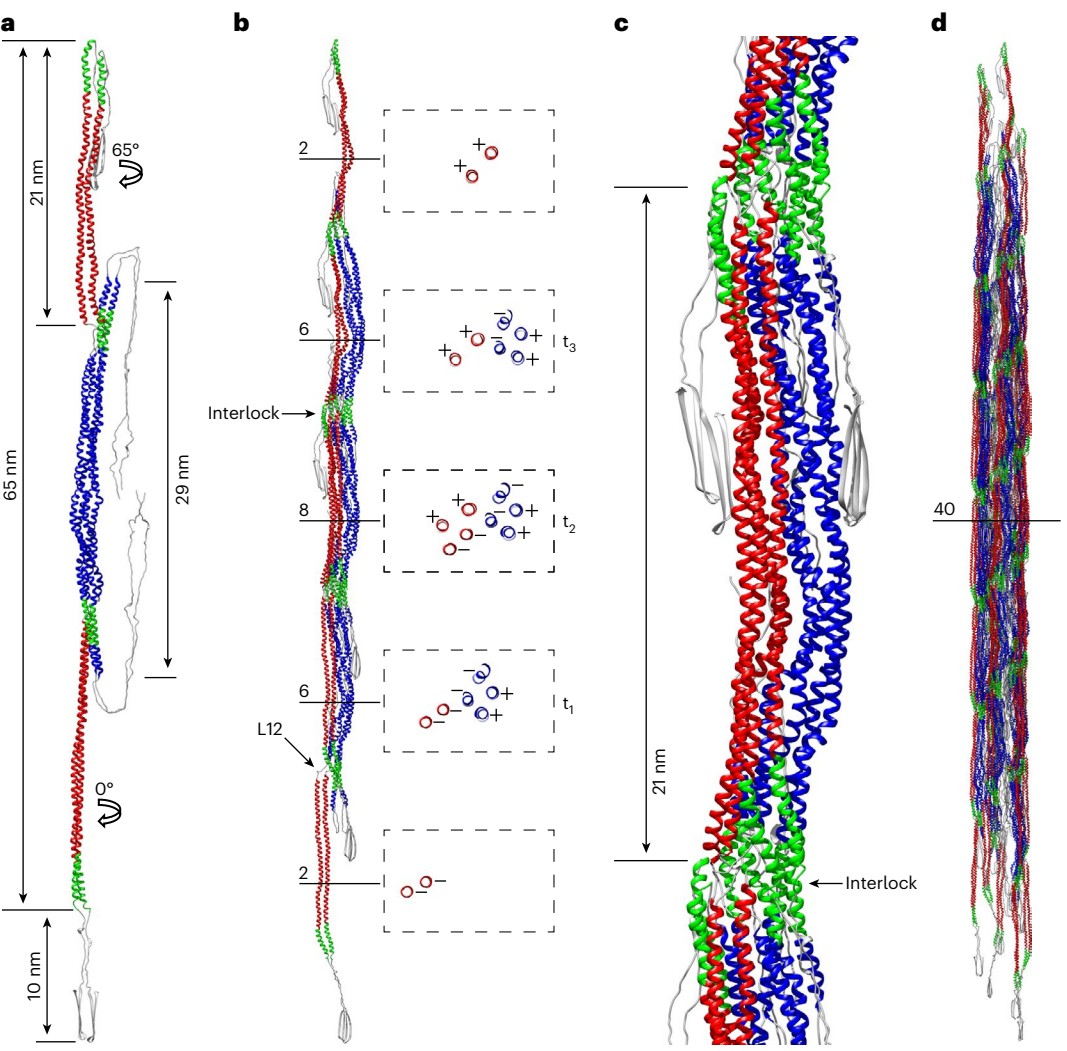

**Fig. 4 | Building blocks of VIFs. a**, Tetramer model. The α-helices shown in red are the 2A–2B dimers, and those in blue are the 1A–1B dimers. Green-colored α-helical segments are highly conserved regions between IFs in the 1A and 2B domains, respectively. **b**, Protofibril model. Numbers and horizontal lines indicate the number of α-helices in cross-section at the respective positions, and $t_1$, $t_2$ and $t_3$ mark the successive tetramers forming the protofibril. The cross-sections along the protofibril are shown within the dashed rectangles. The polarity of the individual α-helices is annotated with + and – signs. **c**, In the fully assembled repeating unit of a protofibril, the interlocking regions are formed on both sides of the assembly and are spaced ~21 nm along the protofibrils. **d**, Extended VIF model constructed from 30 tetramers with a length of ~190 nm.

side of the assembly. The repeating unit unifies the three basic molecular interactions of vimentin dimers forming tetrameric units, defined as the $A_{11}$, $A_{22}$ and $A_{12}$ binding modes, as shown by cross-linking analysis[62]. In the 1A–1B region, the $A_{11}$ cross-links are formed; in the 2A–2B region, the $A_{22}$ cross-links are formed; and between the 1A–1B and 2A–2B regions, the $A_{12}$ cross-links are formed. Mapping these cross-links on our model shows its excellent agreement with the original biochemical data, with a mean distance between the cross-linked lysine residues of 9.4 Å (Supplementary Fig. 9)[64]. The $A_{CN}$ overlap[62] of VIFs between the NTEs of the 1A–1B domains and the CTEs of the 2A–2B domains is ~30–40 Å.

Finally, for a minimal-length VIF model with 40 polypeptide chains per cross-section in its center, the same rule applies: at least 15 tetramers (3 tetramers per minimal-length protofibril, and five protofibrils) are needed to form such a fully polymerized VIF (Fig. 4d, Extended Data Fig. 10 and Supplementary Video 9).

## Discussion

Advances in cryo-FIB and cryo-ET methodologies, in combination with extensive statistical analyses and single-particle cryo-EM, enabled us to visualize and describe the molecular architecture of VIFs in unprecedented detail.

To obtain a physiologically relevant VIF structure, we measured protofibril stoichiometry in situ. It was important to address this question, because of the significant structural polymorphism of VIFs[38,40]. In light of this, cryo-FIB milling in conjunction with cryo-ET is a powerful approach, because it minimizes potential uncertainties in protofibril stoichiometry introduced by VIF extraction from cells or in vitro polymerization of VIFs. Compared with the cellular data, we detected an increased variation in the diameter of VIFs assembled in vitro (Supplementary Fig. 5a). In addition, we observed unraveling of VIFs into individual protofibrils in the cryo-EM micrographs, as has previously been reported[49]. Therefore, it was crucial to directly observe the protofibril stoichiometry of VIFs in cells. On the basis of these results, we conclude that our VIF single-particle structure obtained from in-vitro-polymerized VIFs, which shows that there are five protofibrils, is representative of the structure of VIFs in cells.

In this context, it is noteworthy that unprocessed tomograms have anisotropic resolution. Because most of the VIFs in our datasets are aligned parallel to the tomographic $xy$ plane, their cross-sections

are located in the *xz* planes of the tomograms. As a consequence, these perspectives are affected by the missing-wedge artifact, which can degrade the quality of the observed structures[68].

To overcome this problem, we followed three experimental strategies. First, we used subtomogram averaging[45,46]. It has previously been shown in the in situ structural analysis of F-actin that if filaments are tilted less than 30° out of the *xy* plane, then molecular resolution 3D averages[48] can be reconstructed from projected subvolumes[47,69,70]. Applying this method to VIF segments yields a subtomogram average clearly showing that the structural arrangement contains five protofibrils, without requiring any prior knowledge of the assembly's symmetry (Extended Data Fig. 1c).

As a second strategy to overcome the missing-wedge problem, we deconvolved the missing wedge with IsoNet[71], which has been used to successfully restore microtubule cross-sections in tomograms. We averaged the missing-wedge-corrected cross-sections of VIFs. The cross-section average further confirmed that VIFs contain five protofibrils (Extended Data Fig. 2d). In our third strategy, we searched for bundles of VIFs in the tomograms in which the filaments run parallel to the *z* axis, thereby presenting cross-sections that are naturally unaffected by the missing wedge. These cross-sections also show the five-protofibril architecture (Fig. 1e and Extended Data Fig. 3).

In comparison to F-actin and microtubules (Fig. 5), the basic building blocks of IFs are not globular proteins, but instead are elongated, flexible tetramers. The tetramers intertwine during polymerization, which complicates structural characterization of IFs. Unlike F-actin and microtubules, the proteins forming IFs contain large stretches of head and tail domains with low sequence complexity. Therefore, the structure of VIFs is substantially different, because VIFs integrate low-complexity domains within a highly ordered helical structure.

At the level of the individual protofibrils, classical molecular interactions (α-helices, coiled coils) are prevalent. The lateral alignment of the 1A–1B and 2A–2B dimers and the formation of a molecular interlocking region composed of the highly conserved 1A and 2B domains intertwine the tetramers to form a protofibril. However, at the level of the fully assembled filament, molecular interactions between intrinsically disordered protein domains play an important part. The protofibrils interact laterally through the tail domains, and the head domains form a fiber in the VIF lumen.

The resolution of our electron density map allows the detection and precise positioning of the helical domains of VIFs, in particular in the less flexible regions of the structure (Fig. 2e). However, side chains and their interactions were not resolved. Moreover, we have not yet been able to directly observe the folding of the head domains, owing to the relatively low resolution of the luminal fiber region. Nevertheless, the higher resolution of the 1A–1B domains and their orientation suggests that the head domains protrude into the lumen of the filament. Therefore, it seems likely that the head domains form the luminal fiber.

Indeed, a feature of low-complexity domains is that they can form amyloid-like fibers. For example, this has been shown for the low-complexity domains of the nucleoporin Nup98 (ref. [72]) and the RNA-binding protein FUS[73]. On a molecular level, these fibers are stabilized by transient cross-β interactions; therefore, unlike pathogenic fibers, they are dynamic structures and fully labile to disassembly[74]. Of particular relevance to this study, it has been shown, using electron microscopy, X-ray diffraction and nuclear magnetic resonance analyses, that for vimentin and other IF proteins, including desmin and neurofilaments, the purified head domains form such fibers[65,75]. On the basis of these results, we hypothesize that the luminal fiber is amyloid-like, formed from the head domains of vimentin. This would introduce an additional layer of structural complexity in VIFs that is based on transient molecular interactions within the luminal fiber.

Phosphorylation has a central role in the regulation of IF assembly and disassembly during the cell cycle and cell differentiation[76,77] and the modulation of their mechanical properties[78]. In VIFs, numerous

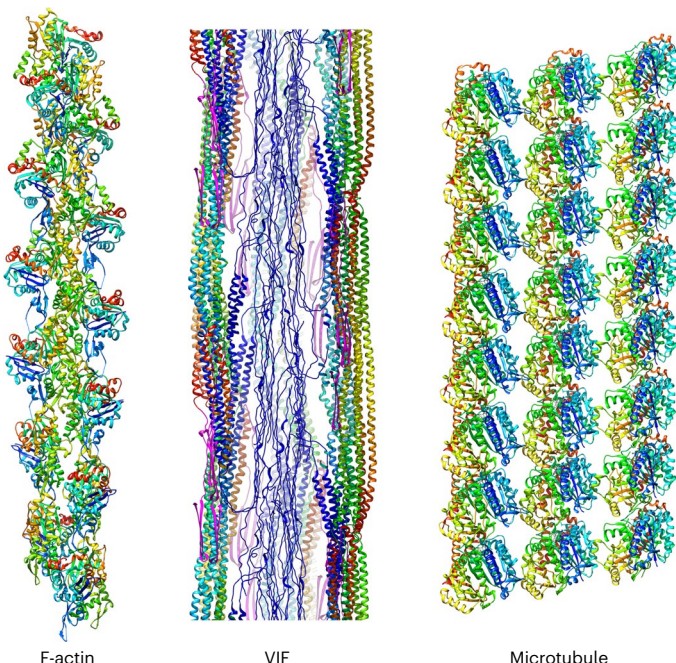

F-actin VIF Microtubule

**Fig. 5 | Complete structural picture of the cytoskeleton.** The functions of the cytoskeleton in cells of mesenchymal origin are based on three biopolymers. F-actin and microtubules are built from compact, globular proteins; VIFs are built from elongated, intertwining tetramers. All three filaments are assembled with helical symmetry. However, in VIFs, a significant part of the structure is formed from domains with low sequence complexity, which condense to connections between the protofibrils and form an amyloid-like fiber in the lumen of VIFs. This introduces an additional layer of structural complexity, based on transient molecular interactions. Three protofilaments of the microtubule are shown. Scale bar, 10 nm.

phosphorylation sites are located in the low-complexity head and tail domains[79]. We speculate that phosphorylation of specific sites in the head domain could gradually destabilize transient cross-β interactions in the luminal fiber and thereby facilitate the disassembly of VIFs[80,81].

A clear implication of our model is that kinases targeting the head domains need to access the VIF lumen[79,82,83]. We suggest that, to facilitate this process, the tail domains add a structural regulatory layer to the filaments in order to control their permeability toward the luminal fiber. In the VIF structure, the tail domains are evenly distributed along the filaments, thereby maintaining close contact between the protofibrils[38]. However, assuming that the tail domains reorganize and form clusters along the filaments, the separation between these clusters of tail domains would be around 47 nm (Supplementary Fig. 10), consistent with previous measurements of the distance between tail domains in VIFs[65,84]. In this tail-domain configuration, the protofibrils could be more separated, and the filament would likely exhibit a more open structure toward the luminal fiber. Ultimately, if the tail domains were completely detached from the protofibrils, substantial gaps between the protofibrils could form, significantly facilitating the access of kinases to the luminal fiber.

Considering the high sequence conservation in the 1A and 2B regions across IFs[29,54], we hypothesize that the molecular interlocking mechanism is similar across IFs. The distance between the interlocking regions is ~21 nm. This axial repeat distance is also conserved across IFs[1,30,52–54]. It has been shown that perturbing the interlocking mechanism with a single point substitution (p.Y117L)[85] prevents the elongation of VIFs. This observation can be explained by our VIF model: without a functional interlocking mechanism, the octameric repeating unit of VIFs cannot form. Several diseases caused by structural defects in IFs are associated with substitutions in the interlocking region[8], including

Alexander disease[86,87], Charcot–Marie–Tooth disease[88] and Epidermolysis bullosa simplex[89,90]. In the future, elucidating the interlocking region of various IFs at atomic resolution will contribute to a better understanding of the molecular basis of these pathologies.

The lateral alignment of the 1A–1B and 2A–2B dimers, the formation of a molecular interlocking region and the tail domain interactions between the protofibrils can occur only in VIFs at higher order polymerization states than the tetramer. However, the interactions of the head domains aggregating to form the luminal fiber are possible at the tetramer stage. Therefore, shorter, minimal-length VIFs would also be stable (Extended Data Fig. 10). Such vimentin assemblies have been experimentally characterized and termed unit-length filaments[38,39]. We suggest that a unit-length VIF is assembled from five tetramers. The initial association of the tetramers could be facilitated by the molecular interactions forming the luminal fiber[75]. Therefore, the head domains may act as nucleators of VIF assembly, in agreement with previous results showing that the head domain is essential for VIF assembly[38].

This unit-length VIF would have a length of ~80 nm (Extended Data Fig. 10a)[38]. Because the tail-domain sites of contact between the tetramers would not yet be formed in this assembly, it would lack diameter control and tend to have a larger diameter than that of VIFs. It has been shown that tail-less VIFs lack diameter control[38] and that a compaction step occurs between unit-length VIFs and VIFs[39]. We suggest that if the tail-domain contact sites and octamer interlocks are formed, the VIF undergoes compaction, reaching a final diameter of 11 nm.

Our results complete the structural picture of the cytoskeleton in cells of mesenchymal origin (Fig. 5). A comprehensive understanding of the mechanical properties of the cytoskeletal network cannot be achieved without first obtaining structural information for all of its components. With the structure of VIFs that we uncovered, it is now possible to obtain an integrative view of the cytoskeleton. Further understanding the unique utilization of low-complexity domains in VIFs, as well as similar domains in other IFs, could begin to explain how IFs complement F-actin and microtubules to generate cell-type-specific mechanical properties.

## Online content

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

## Methods

### Cell lines and cell culture

MEFs[91] (received from the Eriksson lab, Åbo Akademi University) were cultured in DMEM (Sigma-Aldrich, D5671), supplemented with 10% FCS (Sigma-Aldrich, F7524), 2 mM L-glutamine (Sigma-Aldrich, G7513) and 100 µg ml$^{-1}$ penicillin–streptomycin (Sigma-Aldrich, P0781), at 37 °C and 5% $CO_2$ in a humidified incubator.

### 3D-SIM imaging

Sub-confluent cultures of MEFs growing on no. 1.5 glass coverslips were fixed with 4% paraformaldehyde for 10 min at room temperature (RT). The fixed cells were permeabilized with 0.1% Triton X-100 for 10 min at RT and then were incubated with chicken anti-vimentin (1:200, 919101, Biolegend) for 30 min in phosphate-buffered saline (PBS) containing 5% normal goat serum. This was followed by staining with Alexa-Fluor-488-conjugated goat anti-chicken antibody (1:400, A-11039, Invitrogen) and DAPI in PBS for 30 min. The stained cells were mounted with ProLong Glass Antifade Mountant (Life Technologies).

Moreover, MEFs were seeded on no. 1.5 glass coverslips; the next day, they were washed with PBS and 2 mM $MgCl_2$ for 5 s, followed by incubation with PBS containing 0.1% Triton X-100, 10 mM $MgCl_2$, 0.6 M KCl and protease inhibitors for 25 s at RT. The extracted cells were rinsed with PBS and 2 mM $MgCl_2$ for 10 s and subsequently incubated with 2.5 units µl$^{-1}$ benzonase for 30 min at RT. After rinsing with PBS and 2 mM $MgCl_2$, the cells were fixed with 4% paraformaldehyde for 5 min at RT. The fixed cells were then stained with chicken anti-vimentin (1:200, 919101, Biolegend) and rabbit anti-lamin-A/C (to determine the location of the nucleus; 1:100, sc-376248, Santa Cruz Biotechnology) antibodies in PBS containing 5% normal goat serum for 30 min at RT, followed by incubation with goat anti-chicken and anti-rabbit secondary antibodies (1:400, A-11039, A-11011, Invitrogen) for 30 min at RT. After being washed in PBS, stained cells were mounted with Prolong Glass Antifade Mountant (Life Technologies).

3D-SIM imaging was done using an N-SIM Structured Illumination Super-Resolution Microscope system (Nikon) using an oil immersion objective lens (SR Apo TIRF100X, 1.49 numerical aperture (NA), Nikon). For 3D-SIM, 26 optical sections were imaged at 100-nm intervals. The raw SIM images were reconstructed with the N-SIM module of Nikon Elements Advanced Research with the following parameters: illumination contrast, 1.00; high-resolution noise suppression, 0.75; and out-of-focus blur suppression, 0.25. Brightness and contrast were adjusted for image presentation.

### Cryo-FIB and cryo-ET in situ analysis of VIFs

MEFs were seeded onto glow-discharged, carbon-coated, gold EM grids (Quantifoil holey carbon R 2/1, Au 200) in standard medium overnight. After a single wash in 1× PBS (Fisher Bioreagents, BP399-1), the grids were vitrified in liquid ethane using a manual plunge freezing device.

The grids were coated with 5–10 nm platinum on carbon with a Leica BAF060 system cooled to −160 °C, prior to cryo-FIB milling. Next, the grids were transferred to the focused ion beam scanning electron microscope (FIB-SEM) (Zeiss Auriga 40 Crossbeam), equipped with a Leica cryo-stage. After transfer, the grids were coated with two 5-s flashes of organometallic platinum, using the internal gas-injection system, set at 28 °C. The cells were milled with a focused gallium ion beam at a constant voltage of 30 kV and a current between 10 and 240 pA at a stage angle of 18°. The process was controlled by the Nano Patterning and Visualization Engine (Zeiss) and monitored with the SEM at 5 kV. The resulting FIB milled lamellae were 100–200 nm thick (Extended Data Fig. 2b).

The cryo-FIB milled grids were transferred to a Titan Krios 300 kV cryo-TEM (Thermo Fisher Scientific) equipped with a K2 summit direct electron detector and a Quantum energy filter (Gatan). The tilt series were acquired using SerialEM[92] with a 100-µm objective aperture at a magnification of ×64,000, a pixel size of 2.21 Å and −4-µm defocus; a dose-symmetric tilt scheme was used[93], from −60° to +60° in 3° increments starting from 0°. Overall, 102 tilt series were acquired, each with a total dose of ~160 e$^-$/A$^2$.

The tomograms were reconstructed and binned four times (pixel size, 8.84 Å) with weighted back-projection (WBP) using IMOD[94] or AreTomo[95]. If platinum remnants were present in the tilt series, IMOD was used for tomogram reconstruction. The platinum particles were used as fiducial markers for tilt-series alignment. AreTomo was used for marker-free tilt series alignment.

Platinum particles were typically highly concentrated at the front edges of the lamellae and served as fiducial markers for precise tilt-series alignment (Extended Data Fig. 1b, blue dashed rectangle). However, in the tomograms, these high-contrast particles introduce a high density of distorting back-projection rays in the tomographic volume above and below the front edge of the lamellae (Extended Data Fig. 1b, red dashed rectangle). These regions were excluded from subsequent data analysis. Therefore, the processed volume of a tomogram was cut out from the original tomogram (Extended Data Fig. 1b, yellow dashed rectangle, and Supplementary Video 1).

### Subtomogram averaging of VIF segments

First, to determine the native protofibril stoichiometry of VIFs, we applied a subtomogram averaging approach. From the dataset, we selected seven representative tomograms with precise tilt-series alignment (residual error, ~3 Å) and a Thon ring spectrum featuring up to seven non-astigmatic Thon rings visible at 0° tilt. Subsequently, we trained a crYOLO network to detect the VIFs. We localized 14,273 subtomograms ($35 \times 35 \times 35$ nm$^3$) containing VIF segments, which were distributed over 1,114 filaments.

We then applied the Actin Polarity Toolbox (APT) pipeline[47,69,70]. Here, the subtomograms were projected with a projection thickness of 16 nm. False positive (for example, actin filaments, microtubule walls or vesicle membranes) and low-quality VIF segments were excluded by multiple rounds of unsupervised 2D classification in RELION[50,56]. This procedure yielded a particle stack of 2,371 good VIF segments. On the basis of the geometrical orientation of the picked filaments in the tomograms, crYOLO calculates initial priors for the in-plane rotation and the tilt angle for each subtomogram (referred to as psi and tilt priors in the RELION convention). The third Euler angle (rot angle) remains a priori unknown. Therefore, we assigned to each VIF segment a random rot prior, and used these priors to reconstruct a rotationally symmetric, tube-like VIF 3D template[4]. Next, we exported the VIF segments and their priors from RELION to cryoSPARC[96] and performed a 3D reconstruction, using the previously generated rotational symmetric structure as initial reference. Without applying any assumptions on the symmetry of the assembly, the average converged to a resolution of ~20 Å and clearly shows the five-protofibril architecture of VIFs (Extended Data Fig. 1c).

Assuming that VIFs are assemblies with helical symmetry, an initial hypothesis about the helical-symmetry parameter of VIFs can be derived from this result, because a five-protofibril architecture would directly translate to a helical twist angle of 360° / 5 = 72°. Second, a characteristic feature of IFs is their ~210-Å axial periodicity[1,30,52–54]. This would directly translate to a helical rise of 210 Å / 5 = 42 Å. We then repeated the previous 3D reconstruction in cryoSPARC, but with application of helical symmetrization, with a helical rise of 42 Å and a helical twist of 72° (Fig. 1d, Extended Data Fig. 1d and Supplementary Fig. 1).

### Cross-section analysis of VIFs

An alternative method to determine the native protofibril stoichiometry of VIFs is to directly evaluate their tomographic cross-sections[4,49]. However, most of the VIFs in the dataset are oriented in the *xy* plane of the tomograms, with a maximal out-of-plane tilt of ~15°, as measured previously by the tilt-angle distribution of the filaments during crYOLO picking and subtomogram averaging of VIF segments. Therefore, their

cross-sections are contained mainly in the $xz$ planes of the tomograms and cannot be directly analyzed, because they are distorted by the missing-wedge artifact[68].

To overcome this problem, we used IsoNet[71] to compensate for the missing-wedge-induced resolution anisotropy in VIF cross-sections. IsoNet is a convolutional neural network that can be trained to learn missing-wedge deconvolution. We used the seven tomograms that we had already selected as input for IsoNet. The network was trained on the full tomograms containing the edge of the lamellae with a high number of platinum particles, which were previously used as fiducial markers for tilt-series alignment. In this case, they serve as high contrast, point-like features that clearly reproduce the missing-wedge convoluted point spread function in the tomograms. Therefore, they could have an advantageous effect on the convergence of the deconvolution network, but at the very least clearly show reduction of the missing wedge after missing-wedge deconvolution (Extended Data Fig. 2b, blue dashed rectangle). After missing-wedge deconvolution, the tomograms were cut to the same volumes as before (Extended Data Fig. 2b, yellow dashed rectangle, and Supplementary Video 2) to exclude regions with a high concentration of back-projection rays caused by the large number of platinum particles at the edge of the lamellae (Extended Data Fig. 2b, red dashed rectangle). The following parameters were used during the IsoNet workflow: for contrast transfer function (CTF) correction, a signal-to-noise ratio fall off parameter of 0.7 was used. For automatic mask creation, a density percentage of 50% and s.d. percentage of 50% were chosen. Training of the missing-wedge deconvolution neural network was based on randomly extracted 2,000 subtomograms (box size, $64^3$ voxels) per tomogram within the automatically created mask, with the following training parameters: 30 iterations; learning rate, 0.0004; dropout rate, 0.5.

The VIF cross-sections (Extended Data Fig. 2c) were extracted from one of the missing-wedge-corrected tomograms. These cross-sections confirm the five-protofibril architecture of VIFs. However, to extract a meaningful statistical conclusion from the cross-section data, we developed the following procedure. We used the 3D coordinates of the previously established 2,371 good VIF segments and extracted their related subtomograms from the missing-wedge-corrected tomograms. Then, the initial crYOLO priors (in-plane rotation and tilt angle) and random rot angles were used to align the VIF segments. An average of these subvolumes yields a rotational symmetric, tube-like 3D structure of VIFs, similar to that used before, as an initial reference for 3D reconstruction. However, in this case, the subvolumes were extracted from the missing-wedge-corrected tomograms. We next extracted the central cross-section from each of the aligned subvolumes, and applied a 2D classification in RELION on these images without alignment. Because the cross-sections are missing-wedge corrected, they can be classified without using a metric that explicitly models the missing-wedge effect[97]. Because the alignment was fixed in this 2D classification, the cross-sections were combined into classes that are similar, under the application of a random rot angle. In this manner, we identified a subset of 444 aligned cross-sections. Finally, these cross-sections were extracted from the original WBP subvolumes and averaged. This cross-section average (Extended Data Fig. 2d) also confirms the five-protofibril architecture of VIFs.

By contrast, a cross-section located in the $xy$ plane of a tomogram exhibits isotropic resolution and can be evaluated directly. Therefore, we identified a tomogram that contained a vimentin bundle oriented approximately parallel to the electron beam (Extended Data Fig. 3a,b, left column). For contrast enhancement, the tomogram was binned four times and denoised using Topaz-Denoise[98]. To ensure that the denoising operation did not create artificial density, we compared the denoised tomogram with its original version (WBP reconstruction in AreTomo), low-pass filtered to 30 Å (Extended Data Fig. 3a,b, right column). In total, we extracted 150 cross-section images (35 × 35 nm$^2$, 8.84-Å thick) that were located in the $xy$ planes of the denoised

tomogram. The $xy$ cross-sections also show the five-protofibril architecture of VIFs (Fig. 1e and Extended Data Fig. 3c).

## Cryo-ET of detergent-treated MEFs
MEFs were grown to ~80% confluency on glow-discharged holey carbon EM grids (R2/1, Au 200 mesh; Quantifoil) prior to preparation for cryo-ET analysis. Grids with a relatively homogeneous distribution of cells were selected using fine tweezers and then were washed in PBS and 2 mM MgCl$_2$ for 5 s. The grids were treated for 20–40 s in pre-permeabilization buffer (PBS containing 0.1% Triton X-100, 10 mM MgCl$_2$, 600 mM KCl and protease inhibitors) and then rinsed in PBS and 2 mM MgCl$_2$ for 10 s. Next, the grids were incubated with benzonase (2.5 units μl$^{-1}$ in PBS and 2 mM MgCl$_2$; Millipore, Benzonase Nuclease HC, purity >99%) for 30 min at RT. After the grids were washed with PBS and 2 mM MgCl$_2$, a 3-μl drop of 10 nm fiducial gold markers (Aurion) was applied to the grids. For vitrification, the grids were manually blotted for ~3 s from the reverse side and plunge frozen in liquid ethane.

Tilt-series acquisition was conducted using a Titan Krios transmission electron microscope equipped with a K2 Summit direct electron detector and Quantum energy filter. The microscope was operated at 300 keV with a 100-μm objective aperture. In total, 225 tilt series were collected at a nominal magnification of ×42,000, and the slit width of the energy filter was set to 20 eV. Super-resolution videos were recorded within a tilt range from −60° to +60° with 2° increments using SerialEM[92]. The image stacks were acquired at a frame rate of 5 frames per second, with an electron flux of ~2.5 e$^-$/pixel/s. The tilt series were recorded with a total electron dosage of ~125 e$^-$/Å$^2$ and within a nominal defocus range of −2 μm to −6 μm. The super-resolution image stacks were drift-corrected and binned twice using MotionCorr[99], resulting in a pixel size of 3.44 Å for the tilt series. For each projection, the defocus was measured, and the CTF was corrected by phase-flipping. Then, from each tilt series, an overview tomogram that had been binned four times was reconstructed (Supplementary Fig. 2b and Supplementary Video 3). CTF correction and tomogram reconstruction were performed using MATLAB scripts (MathWorks), derived from the TOM toolbox[100,101].

Algorithms from EMAN2 (ref. [102]) were used to train a convolutional neural network that could segment VIFs in the overview tomograms. The segmentations were manually checked and cleaned from obvious false-positive VIF detections in Chimera[103]. On the basis of scripts derived from APT[47], two sets of segment coordinates were extracted from the segmentations. In the first set (values for the second set are in parentheses), the picking distance along the VIFs was set to 165 Å (55 Å), resulting in 390,297 (1,148,072) segment coordinates. Next, on the basis of the segment coordinates, two stacks of subtomograms were reconstructed from the CTF-corrected tilt series with the TOM toolbox. The dimensions of the subtomograms were 65 × 65 × 65 nm$^3$ and 38 × 38 × 38 nm$^3$, respectively, and the voxel size was 3.44 Å.

Subsequently, APT scripts were applied to project the subtomograms, using a projection thickness of 331 Å for the first set and 220 Å for the second set. The size of the subtomogram projections derived from the first and second coordinate sets were 65 × 65 nm$^2$ and 38 × 38 nm$^2$, respectively.

## Initial estimate of helical symmetry
The VIF segments were subjected to multiple rounds of unsupervised 2D classifications in RELION[50,56]. Two-dimensional classes in which single VIFs were not combined (for example, segments containing multiple VIFs running parallel or crossing on top of each other) or those containing false-positive VIF detections (for example, actin filaments or vesicle membranes), determined by visual inspection, were excluded. Consequently, the first particle set was concentrated to 133,780 (and the second to 615,106) VIF segments.

On the basis of the resulting 2D class averages from the first set of particles (Fig. 1f), 2D autocorrelation functions of the class averages were calculated using the TOM toolbox function tom_corr, which were

then displayed as profile plots and averaged (Supplementary Fig. 2c), to measure the distance between similar features observed in the class averages (Fig. 1f, yellow asterisks). The resulting 2D class averages from the first particle set were also used to calculate an averaged power spectrum with the TOM toolbox function tom_ps, to prove that the VIFs had helical symmetry and to obtain an estimate of their helical symmetry parameter (Fig. 1g). This initial estimate was 72° for the helical twist angle and 37 Å for the helical rise.

### Computational filament assembly

The assembly of extended stretches of VIFs (computationally assembled VIFs, ca-VIFs) was based on the 2D classification of the second particle set (Supplementary Fig. 3a). For this purpose, the 2D transformation calculated for each segment (namely its in-plane rotation angle and $xy$ translation) was inverted and applied to the respective class averages, so that the inversely transformed class averages matched the position and orientation of the segments in the tomogram image frame[4,104]. As a result of this operation, the ca-VIFs are represented by a series of class averages (Supplementary Fig. 3b,c), which significantly improves their signal-to-noise ratio compared with that of the raw filaments. Additionally, the ca-VIFs were unbent (Supplementary Fig. 3d), determined using a MATLAB algorithm derived from the ImageJ[105] straighten function[1,106].

In total, 5,205 ca-VIFs of different lengths were assembled (Extended Data Fig. 4a). We selected a subgroup of 389 ca-VIFs that were ≥353 nm long. These were boxed to a uniform length of 353 nm and were measured using autocorrelation (Extended Data Fig. 4b), to determine whether the previously detected periodicity (Fig. 1f, yellow asterisks) persists over much longer distances. This result showed a long-range periodic pattern in 2D projections of VIFs that repeats every 186.5 Å ± 26.0 Å, supporting the previous measurements taken from single class averages.

### Determination of helical symmetry

Next, we calculated a combined power spectrum of the ca-VIFs (Extended Data Fig. 4c). Owing to the increased resolution, the previously detected layer lines (Fig. 1g) are split into fine layer lines, and a dense spectrum of layer lines is revealed. For determination of the helical symmetry of VIFs on the basis of the power spectrum of the computationally assembled filaments, the following procedure was developed in MATLAB.

Firstly, 1-pixel-wide rows in the interval between 1/30 Å and 1/69 Å were sequentially extracted from the power spectrum and compared by cross-correlation with a zero-order Bessel function (assuming a filament radius of 55 Å). The similarity of the extracted rows with a meridional reflection was measured, indicative of the helical rise of the underlying helical assembly[51]. As a result, the layer line at 1/42.5 Å was identified as being related to the helical rise of VIFs (Extended Data Fig. 4d,f,g).

In the next step, the sequence of the layer lines (1/207.4 Å, 1/195.9 Å, 1/185.6 Å, 1/176.3 Å; shown in Extended Data Fig. 4c, inset), which are organized around the layer line that reflects the long-range periodic pattern found previously (Extended Data Fig. 4b), were related to a similar sequence of layer lines found around the meridional reflection (1/42.5 Å, 1/40.1 Å, 1/37.9 Å, 1/36.3 Å; shown in Extended Data Fig. 4d). To this end, the relationship $n = P / h_r$ was applied, which connects the number of asymmetric units ($n$) with the helical pitch ($P$) and helical rise ($h_r$) of a helical assembly. In particular, we searched for the optimal $n$ value that relates the two given sequences of layer lines by $P = n \times h_r$, interpreting the first sequence as layer lines associated with the helical pitch of VIFs and the second sequence as layer lines associated with the helical rise of VIFs. The result of this calculation was $n = 4.8824$ (Extended Data Fig. 4e). A helical twist angle of 73.7° (that is 360° / $n$) and a helical rise of 42.5 Å were determined on the basis of the power spectrum of ca-VIFs.

However, owing to the high density and complexity of the layer line spectrum, we could not index the Bessel peaks[51], so the uniqueness of this solution needed to be proved by other means. Therefore, we conducted extensive helical 3D classifications and helical-symmetry searches on the basis of single-particle data, which converged independently to a numerically identical solution (see below).

### Cryo-EM of human VIFs polymerized in vitro

Human full-length vimentin was expressed in a transformed BL21 *Escherichia coli* strain and isolated from inclusion bodies, as described previously[107]. The protein was stored in 8 M urea, 5 mM Tris HCl (pH 7.5), 1 mM EDTA, 10 mM methylamine hydrochloride (Merck) and approximately 0.3–0.5 mM KCl at −80 °C.

To reassemble human vimentin, the protein was subjected to a stepwise dialysis in phosphate buffer at pH 7.5. A dialysis tube with a 12–14-kDa cutoff (Serva) was rinsed three times with dialysis buffer 1 (6 M urea, 2 mM phosphate buffer). Next, 200 µl of purified vimentin (0.2 mg ml⁻¹) in 8 M urea was added to the dialysis tube, which was dialyzed for 1 h at RT against dialysis buffer 1 under gentle magnetic mixing. The buffer was exchanged three times in 1-h intervals with fresh dialysis buffer, with decreasing concentrations of urea (dialysis buffer 2: 4 M urea, 2 mM phosphate buffer; dialysis buffer 3: 2 M urea, 2 mM phosphate buffer; dialysis buffer 4: 1 M urea, 2 mM phosphate buffer). In the last step, the buffer was exchanged with 2 mM phosphate buffer, and vimentin was incubated for 2 h at RT.

VIF formation was initiated by dialyzing vimentin with 2 mM phosphate buffer containing 100 mM KCl at pH 7.5 at 37 °C overnight. To prepare samples for cryo-EM, 4 µl of filamentous vimentin at a concentration of 0.1 mg ml⁻¹ was applied to glow-discharged, carbon-coated copper grids (Cu R2/1, 200 mesh, Quantifoil) and vitrified in liquid ethane using a manual plunge freezing device.

The VIF grids were imaged using a 300 kV Titan Krios G3i cryo-electron microscope (Thermo Fisher Scientific) equipped with a K3 direct electron detection camera (Gatan) mounted on a Bio Quantum Energy Filter (Gatan). Dose-fractionated micrographs were acquired with a 100-µm objective aperture and zero-loss energy filtering using a 20-eV slit width, at a magnification of ×130,000, with a pixel size of 0.34 Å in super-resolution mode using EPU (Thermo Fisher Scientific). A defocus range of −0.8 to −2.8 µm was chosen. The frame exposure time was set to 0.013 s, with a total exposure time of 1 s per frame (75 frames in total), resulting in a total electron dosage per dose-fractioned micrograph of ~62 e⁻/Å². The micrographs were corrected for beam-induced motion with RELION[56] and binned by Fourier cropping, resulting in 12,160 micrographs (Supplementary Fig. 4) with a pixel size of 0.68 Å. For CTF estimation, Gctf[108] was used. The estimated maximal resolution per micrograph histogram was optimized as a function of the amplitude contrast fraction parameter. This procedure yielded an improved CTF model to describe the data with an amplitude contrast fraction of 0.2, as compared with the default value of 0.1.

### Single-particle reconstruction of VIFs

Filament picking was performed with crYOLO[109]. We used a neural network trained on in-vitro-polymerized keratin K5 and K14 IFs[4], which were acquired using identical cryo-EM microscopy parameters to those used to acquire the VIFs, and we applied this neural network to the VIF micrographs. In the prediction step, the filament mode was activated, with an initial box distance between the VIF segments of 27.2 Å. Directional method convolution was used, and a search range factor of 1.41 and a filament and mask width of 14 nm were specified. As a result, 130,094 VIFs were picked from the micrographs, which were subdivided in 1,462,717 VIF segments with a particle box size of 38 × 38 nm². For initial particle sorting and cleaning, the segments were extracted with a pixel size of 6.8 Å and subjected to multiple rounds of unsupervised 2D classification in RELION[56,57]. In between rounds of 2D classification, the 2D classes combining segments of low quality (for

example, segments containing overlapping or unraveling VIFs or carbon edges), determined by visual inspection, were excluded, therefore reducing the number of particles gradually to 801,585 good VIF segments. At this stage, the 2D classes (Supplementary Fig. 5a) already showed the characteristic pattern described above, in which one filament wall appears more pronounced in projection than does its counterpart (Fig. 1f and Supplementary Figs. 2d and 3a). However, at the same time, the 2D classes showed that VIFs exhibit pronounced variability in diameter, similar to what we previously observed for keratin K5 and K14 IFs[4].

Because the diameter of a helical filament is coupled to its helical symmetry, multiple rounds of helical 3D classifications were performed in RELION[56,57] with the aim of reducing heterogeneity in the particle set. As initial 3D template for helical 3D classifications a rotational symmetric filament tube was reconstructed from the VIF segments using 90° as approximation for the tilt angle, the refined psi angle from the previous 2D classification step and a random angle as the rot angle[4]. As a starting point for the helical symmetry search, an initial twist angle of 72° and an initial helical rise of 37 Å were set (Fig. 1g), and this helical symmetry was also imposed on the initial 3D template. The actual helical-symmetry search was performed in an interval between 50° to 100° for the helical twist and 30 Å to 57 Å for the helical rise, capturing possible helical assemblies based on four or six protofibrils.

In between the rounds of helical 3D classification, the 3D classes that converged to the borders of the search interval were removed, therefore reducing the number of particles gradually to 520,902 good VIF segments. In the final round of helical 3D classification (Supplementary Fig. 5b,c), the mean helical symmetry calculated from all the VIF 3D classes was 73.4° for the helical twist angle and 42.1 Å for the helical rise. These parameters were used as the initial twist and rise values for subsequent local helical symmetry searches during 3D refinement.

The processing of the particles was iteratively switched in RELION between 3D auto-refine and 3D classification jobs[56,57]. VIF segments at this stage were extracted with a pixel size of 1.0 Å. For 3D auto-refine jobs, the angular sampling was reduced to 1.8°, with local angular searches starting from 0.9°. Local helical symmetry search was switched on. Subsequently, a 3D classification job without image alignment was conducted. The 3D classes that were carried over to the next 3D auto-refine job were selected on the basis of visual inspection. Additionally, particles converging to a distance of <12 Å between neighboring segments were removed. The number of particles was gradually reduced to a final number of 236,920 good VIF segments. The final local helical-symmetry search converged to a helical twist angle of 73.7° and a helical rise of 42.5 Å, which is numerically identical to the helical-symmetry values previously determined by power spectrum analysis (Extended Data Fig. 4).

To improve the resolution of the VIF structure, a reference mask was applied during subsequent 3D auto-refine runs (Supplementary Fig. 6c). This mask was based on an intermediate VIF structure that was low-pass-filtered to 30 Å. In this structure, the luminal fiber was removed using UCSF Chimera[103], with the aim of reducing the structural heterogeneity in the averaging volume. The RELION command relion_mask_create was used to extend the binary reference mask by 6 voxels and add a soft edge of 12 voxels. The result of this masked 3D auto-refine job with applied helical symmetry (helical twist 73.7°, helical rise 42.5 Å) was sharpened with LocalDeblur[110]. The LocalDeblur parameter $\hat{\lambda}$ and $K$ were set to 1.0 and 0.025, respectively, and the local-resolution map was calculated using ResMap[111].

For the final 3D auto-refine job, the reference mask described above was kept constant and the sharpened structure described above was used as initial reference. However, in contrast to previous 3D auto-refine jobs, helical symmetry was relaxed and not applied during final 3D refinement. The resulting VIF 3D structure reached a resolution of 7.2 Å (Supplementary Fig. 6f).

To visualize the increased homogeneity of the VIF segments that were used in the final 3D auto-refine job, the segments were combined with a 2D classification job in RELION. Here, the 2D alignment of the particles was fixed to the translations and in-plane rotations that were calculated in the final 3D auto-refine job (Supplementary Fig. 5d).

The final VIF 3D structure (Fig. 2 and Supplementary Fig. 6) was sharpened using LocalDeblur ($\hat{\lambda} = 1.0$, and $K = 0.025$), helical symmetry was imposed (helical twist 73.7°, helical rise 42.5 Å), and the structure was low-pass-filtered to 7.2 Å. The local-resolution distribution of the map was calculated with ResMap (the mean resolution of all analyzed voxels was 5.2 Å) and interpreted as a measure for local structural plasticity (Fig. 2e and Supplementary Fig. 6e).

### Cryo-EM of in-vitro-polymerized human VIFs-ΔT

The gene encoding human vimentin ΔT (amino acids 1–410) was cloned in pEt24d(+), and the protein was expressed in Rosetta2 pLysS as inclusion bodies[107]. The protein was purified in multiple steps, consisting of isolation of the inclusion bodies, guanidinium chloride-based solubilization and size-exclusion chromatography of the recombinant ΔT vimentin. In brief, the cell extract obtained after sonication of the frozen bacteria pellet in a buffer containing Tris HCl 50 mM pH 8, NaCl 200 mM, glycerol 25%, EDTA 1 mM, lysozyme 10 mg ml$^{-1}$, MgCl$_2$ 20 mM, DNase 1 8 µg ml$^{-1}$, RNase A 40 µg ml$^{-1}$, NP40 1%, deoxycholic acid sodium 1% and one cOmplete Protease Inhibitor Cocktail tablet (Roche) was centrifuged for 30 min at 12,000$g$ and 4 °C. The inclusion-bodies pellet was washed three times (or more if dark material persisted) using cycles of resuspension and centrifugation (30 min at 12,000$g$, at 4 °C) and a buffer containing Tris HCl 10 mM, pH 8, Triton X-100 0.5%, EDTA 5 mM, DTT 1.5 mM and one cOmplete Protease Inhibitor Cocktail tablet. As a last step, the inclusion-bodies pellet was washed in a buffer containing Tris HCl, pH 8 10 mM, EDTA 1 mM, DTT 1.5 mM and one cOmplete Protease Inhibitor Cocktail tablet. The vimentin ΔT protein was then solubilized with 6 M guanidine hydrochloride in 10 mM Tris HCl pH 7.5, and clarified for 30 min at 10,000$g$ at 4 °C. The vimentin-containing supernatant was collected. A size-exclusion chromatography using Superdex 200 Increase 10/300 GL (Cytiva) was performed. Protein purity was checked using SDS–PAGE.

The protein concentration was adjusted to 0.2–0.4 mg ml$^{-1}$, and VIF-ΔT filaments were reconstituted by serial dialysis of 30 min at 22 °C using buffers of decreasing concentration of guanidine hydrochloride, followed by an overnight dialysis step at 4 °C in guanidine-hydrochloride-free buffer. The dialysis buffers were composed of 5 mM Tris HCl pH 7.5, with 1 mM EDTA, 0.1 mM EGTA and 1 mM DTT, containing 4, 2 and 0 M guanidine hydrochloride. The dialyzed protein solution was used for VIF-ΔT assembly by dialysis using a high-salt buffer containing 10 mM Tris HCl pH 7.5 with 100 mM KCl.

To prepare samples for cryo-EM, 3 µl of the VIF-ΔT solution was applied onto glow-discharged holey carbon EM grids (Cu R2/1, 200 mesh, Quantifoil), which were subsequently blotted manually and plunge frozen in liquid ethane.

For imaging, the VIF-ΔT grids the same EM setup were used as before, with the only difference being that the dose-fractionated micrographs were recorded in counting mode. The micrographs were corrected for beam-induced motion with MotionCor2 (ref. 112) without binning, resulting in 19,534 micrographs with a pixel size of 0.68 Å. For CTF estimation, Gctf[108] was used, with the same amplitude contrast fraction as for the CTF estimation of the VIF dataset.

### Single-particle reconstruction of VIFs-ΔT

Filament picking was performed with crYOLO[109], with the same neural network and parameter settings as for the VIF dataset. In the VIF-ΔT dataset, 1,019,393 VIF segments were initially picked, with a particle box size of 38 × 38 nm$^2$. Particle sorting and cleaning of the segments was performed as described for the VIF dataset, with multiple rounds of unsupervised 2D classifications in RELION[56,57] and low-quality particle exclusion on the basis of visual inspection of 2D classes, therefore reducing the number of particles gradually to 143,270 good VIF-ΔT segments.

The VIF-ΔT structure was calculated in RELION with a 3D auto-refine job, without applying helical symmetry. Subsequent to 3D refinement, helical symmetry was imposed (helical twist 73.7°, helical rise 42.5 Å), and the structure was low-pass-filtered to 14 Å for visualization (Extended Data Fig. 5d).

### Model building

In the first stage, AlphaFold[59] was used to predict the model of a tetramer consisting of two 1A–1B chains and two 2A–2B chains. Two copies of this tetramer were docked using ClusPro[113] to form an initial octameric model of the repeating unit of a VIF protofibril. The previously published cross-links[62,63] were mapped[64] on this model, and the distance between the cross-linked lysine residues was minimized by a simple geometric rotation and translation of the dimeric entities (namely the 1A–1B and 2A–2B dimers).

One 2A–2B dimer was rotated −5° around its center and translated +20 Å in the z direction; the second was rotated +5° around its center and translated −20 Å in the z direction. Then, one 1A–1B dimer was translated −20 Å in the z direction, and the second +20 Å in the z direction. These simple geometric transformations of the 1A–1B and 2A–2B dimers relative to each other improved the mean distance between the cross-linked residues to 15.6 Å.

Next, the cross-link corrected model was rigid-body docked to the VIF 3D structure using UCSF Chimera[103] and adapted to the electron density map by molecular dynamics flexible fitting using namdinator[61,114].

Then, the helical symmetry of VIFs was applied to form a model of a protofibril. From this model, an initial tetramer model was extracted and converted to a simulated electron density map, which was used in combination with molecular dynamics flexible fitting to transform two copies of the AlphaFold model of the full-length vimentin dimer (Extended Data Fig. 6 and Supplementary Fig. 7) to the overall shape of the tetramer. This initial complete tetramer model (including the head, tail and linker L12 domains) was then further refined with the VIF electron density map using molecular dynamics flexible fitting, resulting in the final tetramer model (Fig. 4a). To assemble the complete VIF atomic model (Fig. 3), the helical symmetry of VIFs was applied to the final tetramer model. Refinement and validation statistics of the VIF model and VIF tetramer model (Table 2) were calculated using Phenix[115].

### Sequence alignment

Sequence alignment of IFs (Supplementary Fig. 8) was performed using the Clustal Omega algorithm[116]. The alignment was obtained after ten combined guide-tree–HMM iterations.

### Visualization

All isosurface representations of the VIF structures and visualizations of the VIF atomic model were rendered with UCSF Chimera[103]. The biopolymer segmentation (Fig. 1c) was manually conducted in 3dmod[94]. Coordinates were picked along the different biopolymers in the tomogram and used as supporting points to render the filaments as tubes reflecting their different diameters. Protofibril segmentation and the segmentation of the repeating unit of a protofibril was conducted with Segger[117]. For the visualization of the complete cytoskeleton, PDB: 8A2R ref. 118 was used for the actin filament, and PDB: 6DPU ref. 119 for the MT.

### Reporting summary

Further information on research design is available in the Nature Portfolio Reporting Summary linked to this article.

### Data availability

The helical symmetrized VIF subtomogram average is deposited in the Electron Microscopy Data Bank under the accession code EMD-19562. The VIF and VIF-ΔT single-particle structures are deposited in the Electron Microscopy Data Bank under the accession codes EMD-16844 and EMD-19563, respectively. The VIF model is deposited in the Protein Data Bank under the accession code PDB: 8RVE. The integrative modeling protocol and AlphaFold initial models of VIFs and the VIF tetramer model are deposited in the PDBDEV under the accession code PDBDEV-00000212. Source data are provided with this paper.

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

## Acknowledgements

This work was funded by grants from the Swiss National Science Foundation (SNSF 310030_207453) and the Mäxi Foundation to O.M. We thank the Center for Microscopy and Image Analysis at the University of Zurich. R.D.G. and S.S. were supported by an NIH program project grant awarded to R.D.G.

## Author contributions

M.E. conceived the research, analyzed all data, developed data analysis methods, built the VIF model and wrote the manuscript. R.K.-T. prepared the cryo-FIB samples and recorded the cryo-ET data. J.K. and S.K. provided the purified full-length vimentin, M.S.W. prepared the samples and recorded the cryo-EM data. R.B.-P. cloned, produced and polymerized the delta-tail vimentin and prepared the cryo-EM samples. C.T.B. recorded the cryo-EM data of the delta-tail vimentin and edited the manuscript. Y.T. prepared the detergent-treated MEF samples and recorded the cryo-ET data. S.S. prepared MEF samples and recorded the light microscopy data. R.D.G. conceived the research and edited the manuscript. O.M. conceived the research, supervised the project, edited the manuscript and acquired the funding.

## Funding

## Competing interests

The authors declare no competing interests.

## Additional information

**Extended data** is available for this paper at https://doi.org/10.1038/s41594-024-01261-2.

**Correspondence and requests for materials** should be addressed to Matthias Eibauer or Ohad Medalia.

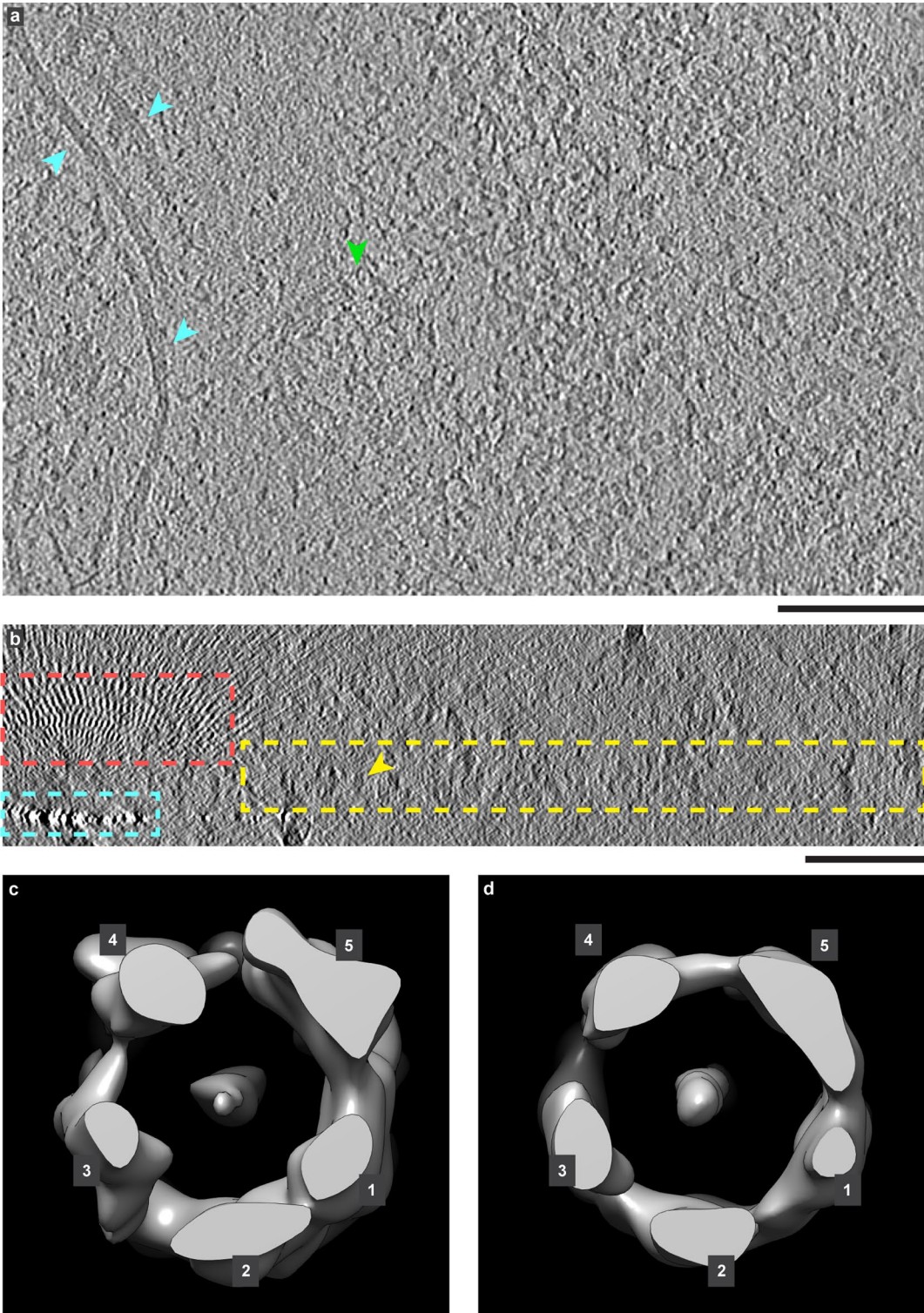

**Extended Data Fig. 1 | Subtomogram averaging of VIFs.** (**a**) Slice (xy-plane, 8.84 Å thick) through a tomogram of cryo-FIB milled MEFs (n = 7). The tomogram was reconstructed with WBP and low-pass filtered to 30 Å for visualization. The cyan arrowheads point to VIFs running in the tomographic xy-plane and the green arrowhead marks a nuclear pore complex. Scale bar is 100 nm. (**b**) The tomographic slice (xz-plane, 8.84 Å thick; n = 7) shows the high contrast and density of the platinum particles at the front edge of the lamella (cyan dashed rectangle), which cause a high density of distorting back-projection rays (red dashed rectangle). These regions in the tomograms were excluded from subsequent data analysis, and only unaffected regions in the tomograms were used for subtomogram averaging of VIFs (yellow dashed rectangle). The yellow arrowhead points to a VIF cross-section in the xz-plane. Cross-sections in this plane are distorted by the missing wedge. Scale bar is 100 nm. (**c**) Isosurface visualization of the 3D reconstruction of VIF segments (2371 particles extracted from 7 cryo-FIB tomograms), calculated without applying any a priori knowledge of the symmetry of the assembly. The average clearly shows the 5 protofibril architecture of VIFs. (**d**) Isosurface visualization of the 3D reconstruction of VIF segments using helical symmetry (42 Å helical rise, 72° helical twist). The average resembles the structure which was obtained without applying symmetry. Scale bar 10 nm.

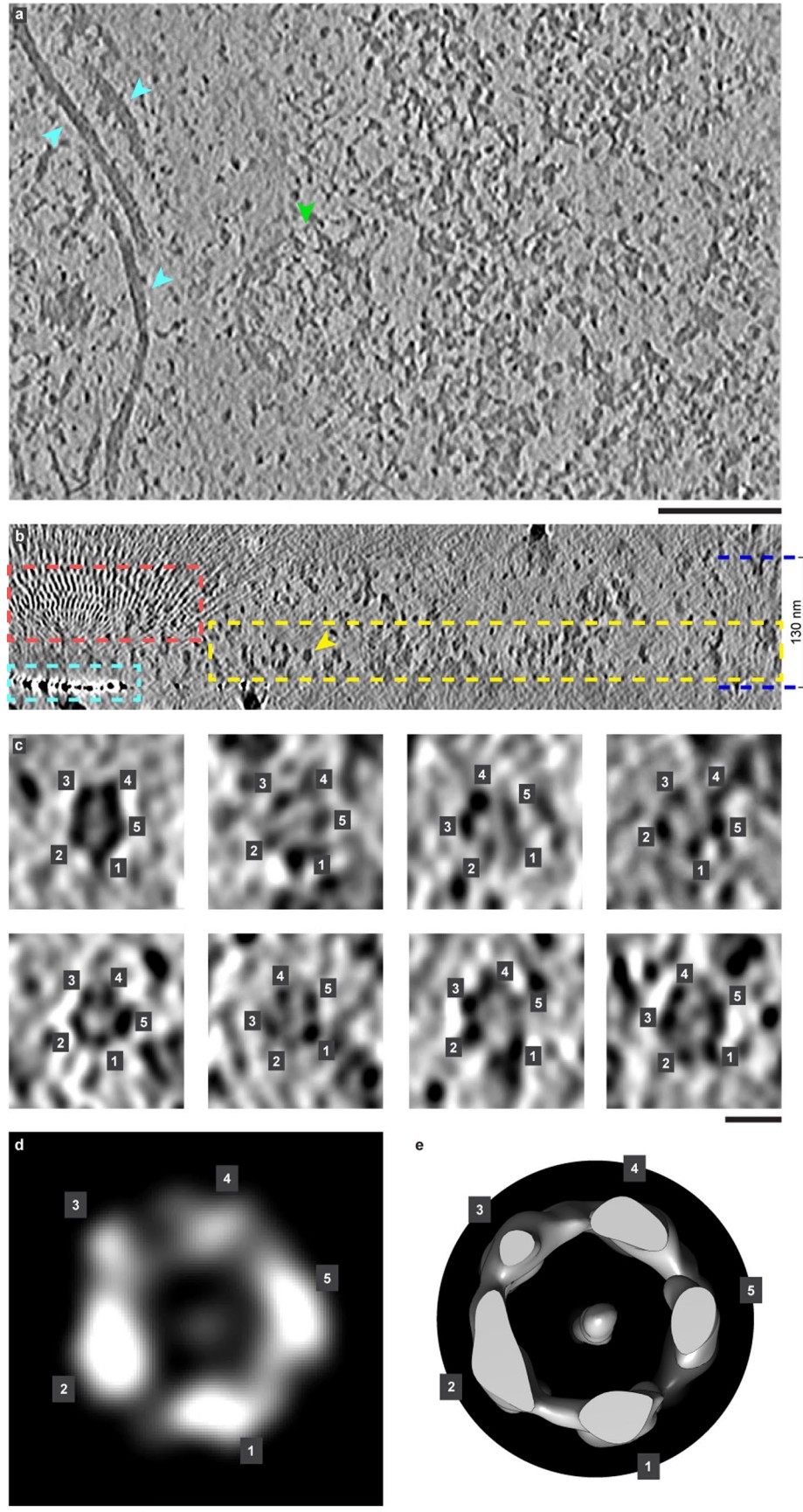

**Extended Data Fig. 2 | See next page for caption.**

**Extended Data Fig. 2 | Cross-section averaging of VIFs. (a)** Slice (xy-plane, 8.84 Å thick) through the same tomogram as shown before (Extended Data Fig. 1a), but the tomogram was missing wedge corrected with the IsoNet algorithm (n = 7). The cyan arrowheads point to VIFs running in the tomographic xy-plane and the green arrowhead marks a nuclear pore complex. Scale bar is 100 nm. **(b)** The tomographic slice (xz-plane, 8.84 Å thick; n = 7) shows the result of the missing wedge correction. As a result, the platinum depositions on the front edge of the lamella (cyan dashed rectangle) appear globular and VIF cross-sections (yellow arrowhead) reveal their pentameric shape. Regions with a high density of distorting back-projection rays (red dashed rectangle) were excluded from subsequent data processing. Only unaffected regions in the tomograms were used for cross-section averaging (yellow dashed rectangle). The lamella has a thickness of -130 nm. **(c)** More examples of pentameric cross-sections of VIFs (n = 8) as seen in the IsoNet corrected tomogram. Scale bar 10 nm. **(d)** VIF cross-section average calculated without any assumptions on the symmetry of the assembly from 444 cross-sections, extracted from the xz-planes of 7 tomograms. The cross-section average clearly confirms the 5 protofibril architecture of VIFs, as seen before with subtomogram averaging. Scale bar 10 nm. **(e)** For comparison the helical symmetrized subtomogram average was rotated and the protofibrils numbered to match the cross-section average.

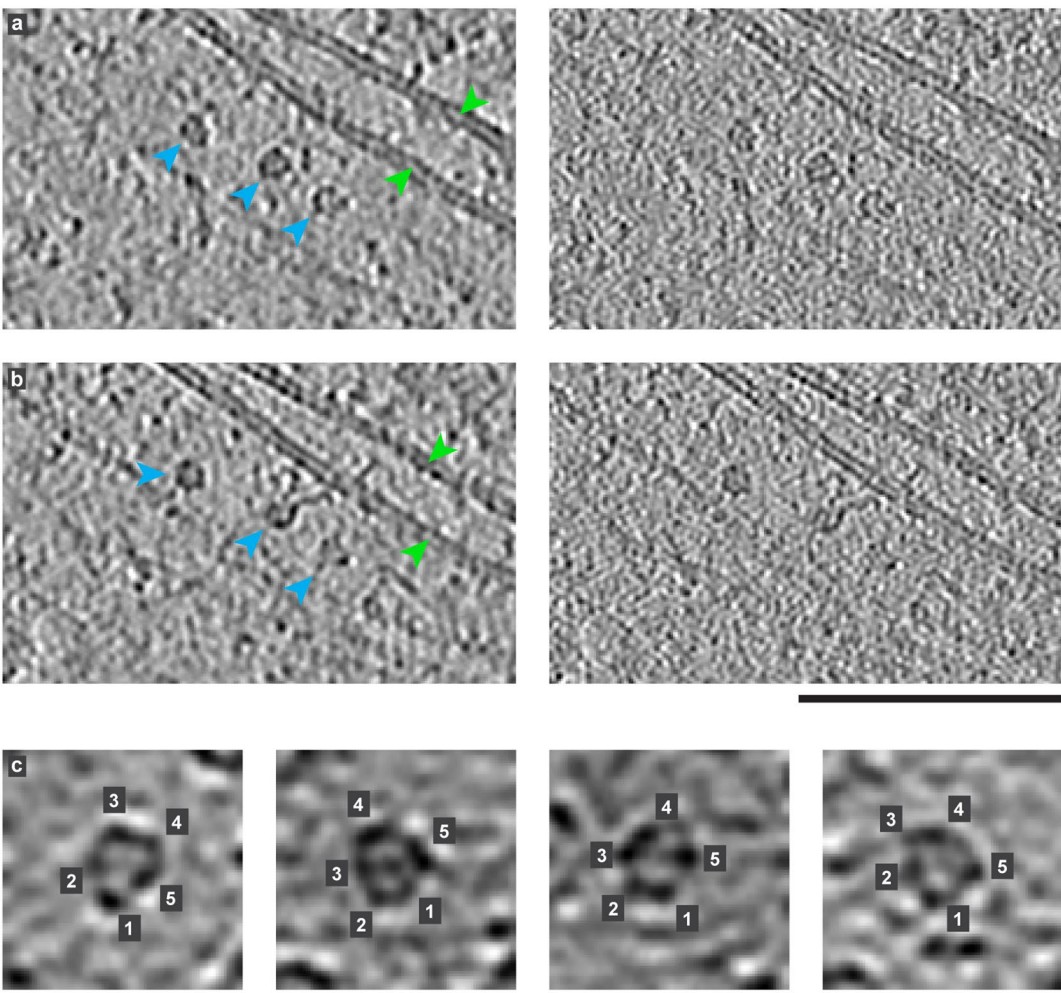

**Extended Data Fig. 3 | Gallery of xy-cross-sections of VIFs. (a)** Slice (xy-plane, 8.84 Å thick) through a tomogram of cryo-FIB milled MEFs. The cyan arrowheads point to cross-sections of VIFs, which are located in the xy-plane of this tomogram (n = 1). The green arrowheads mark the outer and inner nuclear membranes. The slice shown on the left was extracted from the denoised version of the tomogram, the slice on the right from the original WBP tomogram, low-pass filtered to 30 Å. **(b)** Another example for a tomographic slice (n = 50) containing VIF xy-cross-sections (cyan arrowheads). Left side, denoised tomogram, right side, WBP tomogram, low-pass filtered to 30 Å. Scale bar 100 nm. **(c)** More examples for pentameric xy-cross-sections of VIFs (n = 150) extracted from the denoised version of the tomogram. Scale bar 10 nm.

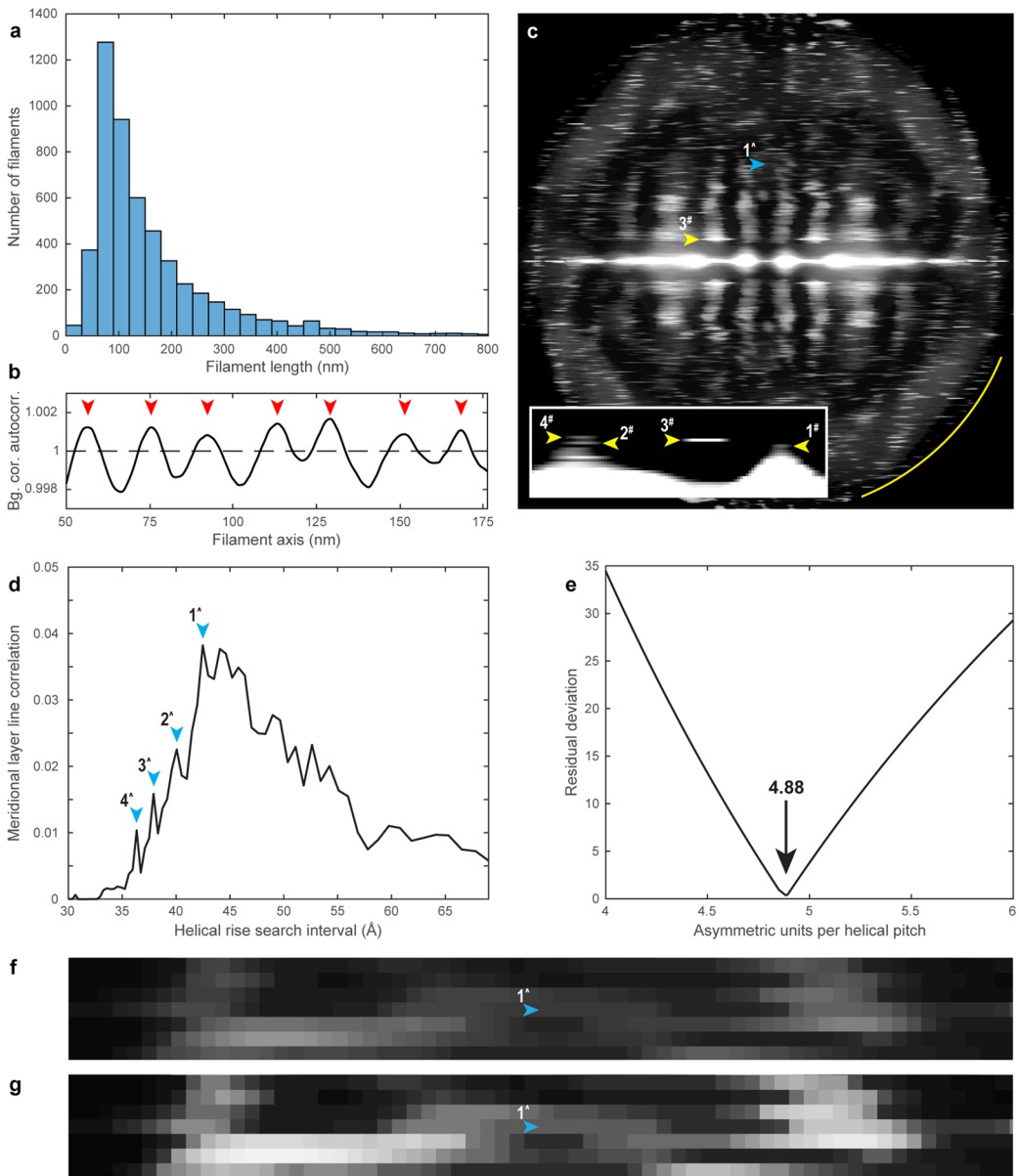

**Extended Data Fig. 4 | Helical parameter determination of VIFs. (a)** Length histogram of the ca-VIFs (n = 5205). **(b)** Profile plot showing the averaged, background corrected autocorrelation signal of ca-VIFs with a minimal length of 353 nm (n = 389). The mean distance between the red arrowheads is 186.5 Å ± 26.0 Å. **(c)** These ca-VIFs were combined into one power spectrum. The meridional reflection at 1/42.5 Å (blue arrowhead, 1˚) indicates the helical rise of VIFs. Around the distinct layer line at 1/185.6 Å (yellow arrowhead, 3#), the following sequence of layer lines was extracted, indicated by 1#-4# in the inset: 1/207.4 Å, 1/195.9 Å, 1/185.6 Å, 1/176.3 Å. The yellow arc indicates 1/16 Å. **(d)** The

meridional reflection at 1/42.5 Å (blue arrowhead, 1˚) was identified by cross-correlation of the layer lines with a zero order Bessel function within a search interval between 1/30 Å and 1/69 Å. In the proximity of the meridional reflection the following sequence of layer lines was extracted, indicated by 1^-4^ in the plot: 1/42.5 Å, 1/40.1 Å, 1/37.9 Å, 1/36.3 Å. **(e)** The optimal number of asymmetric units per helical pitch that relates both sequences of layer lines was determined. **(f)** Magnified cutout image of the meridional layer line (blue arrowhead, 1˚) from the power spectrum shown in (**c**). In (**g**) the same magnified cutout image of the layer line is shown, but with an additional contrast enhancement.

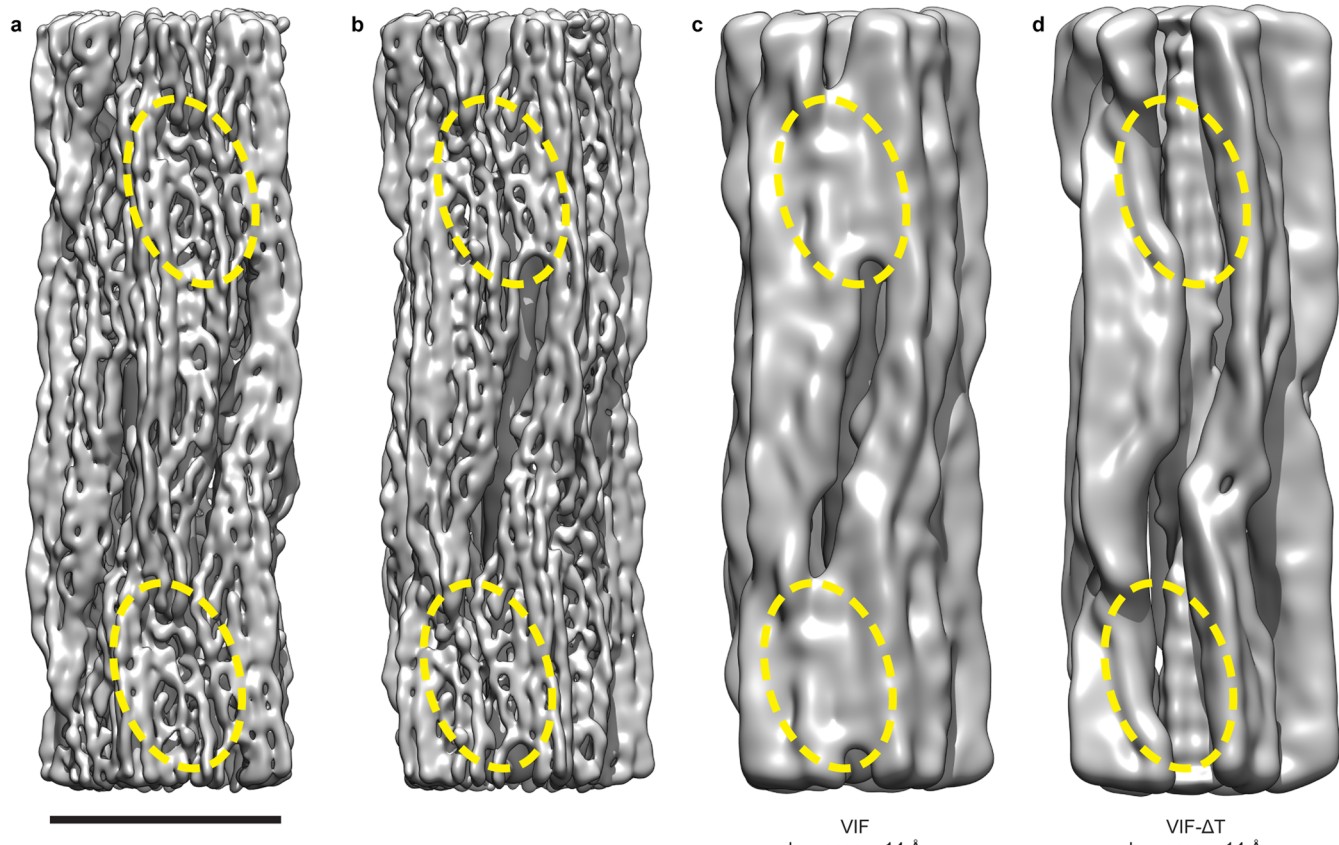

VIF
low-pass = 14 Å

VIF-ΔT
low-pass = 14 Å

**Extended Data Fig. 5 | Comparison between the wildtype VIF and VIF-ΔT structures.** (**a**) Isosurface rendering of the VIF structure. The contact sites between the protofibrils are circled by yellow dashed ovals. Scale bar is 10 nm. (**b**) Identical structure as shown in (**a**), but rotated -36° around the helical axis (**c**) Identical view as shown in (**b**), but the VIF structure was low-pass filtered to 14 Å. (**d**) Isosurface rendering of the VIF-ΔT structure, which is displayed with identical view and low-pass filter settings as the VIF structure depicted in (**c**). Both structures (VIF and VIF-ΔT) are similar to a great extent, and both structures contain a luminal fiber of similar volume. However, their striking difference, as indicated by the yellow dashed ovals in (**c**) and (**d**), is that the VIF-ΔT structure is clearly lacking the contact sites between the adjacent protofibrils.

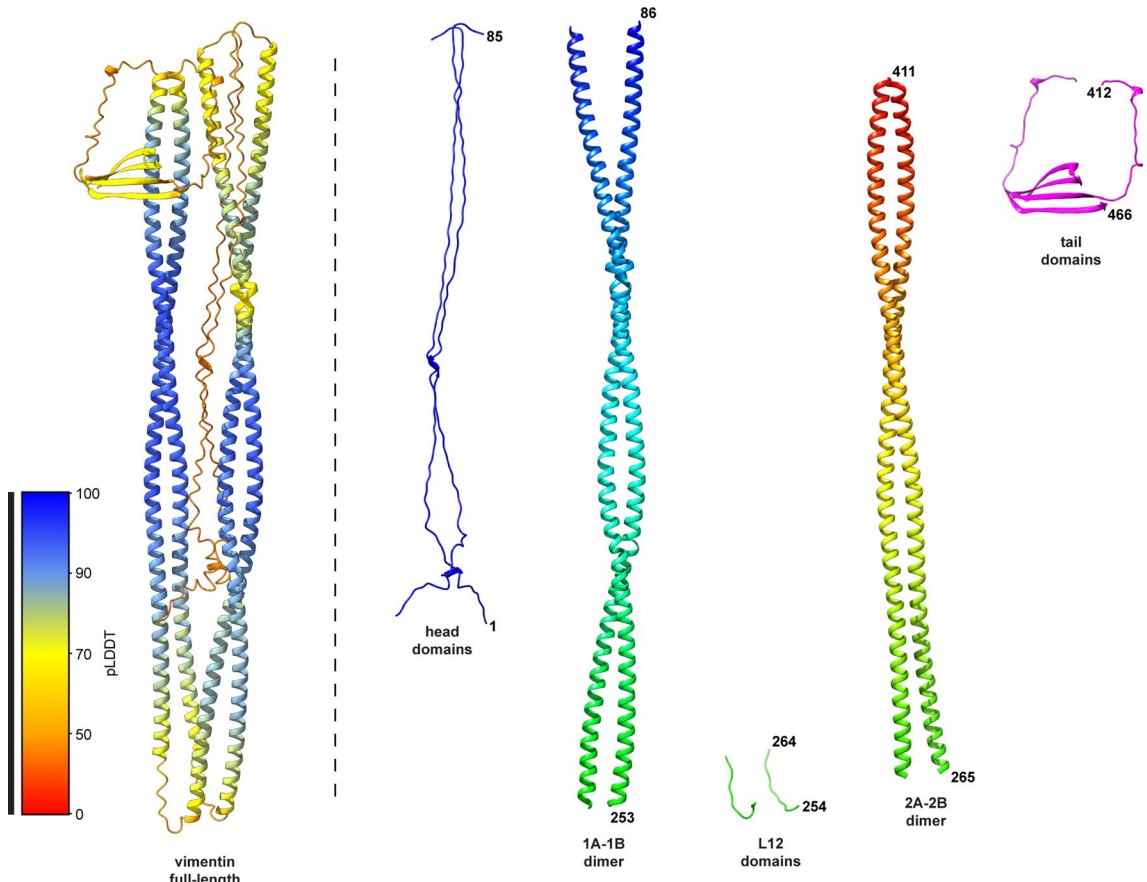

**Extended Data Fig. 6 | Alphafold prediction of the vimentin full-length dimer.** The atomic model shown on the left is the ranked-0 alphafold prediction result of the vimentin full-length dimer. The residues are colored according to the pLDDT confidence score of the alphafold prediction. Based on this model we define the vimentin protein domains as follows: head, residues 1-85, colored blue; 1 A-1B, residues 86-253, colored from blue[NTE] to green[CTE]; linker L12, residues 254-264, colored green; 2A-2B, residues 265-411, colored from green[NTE] to red[CTE]; tail, residues 412-466, colored magenta. Based on this definition the full-length dimer model was dissected into dimeric models of the vimentin protein domains (shown in the figure to the right of the dashed line; labelled with head domains, 1A-1B dimer, L12 domains, 2A-2B dimer, and tail domains). These models constituted the initial folds for the subsequent building of the VIF atomic model. Scale bar 10 nm.

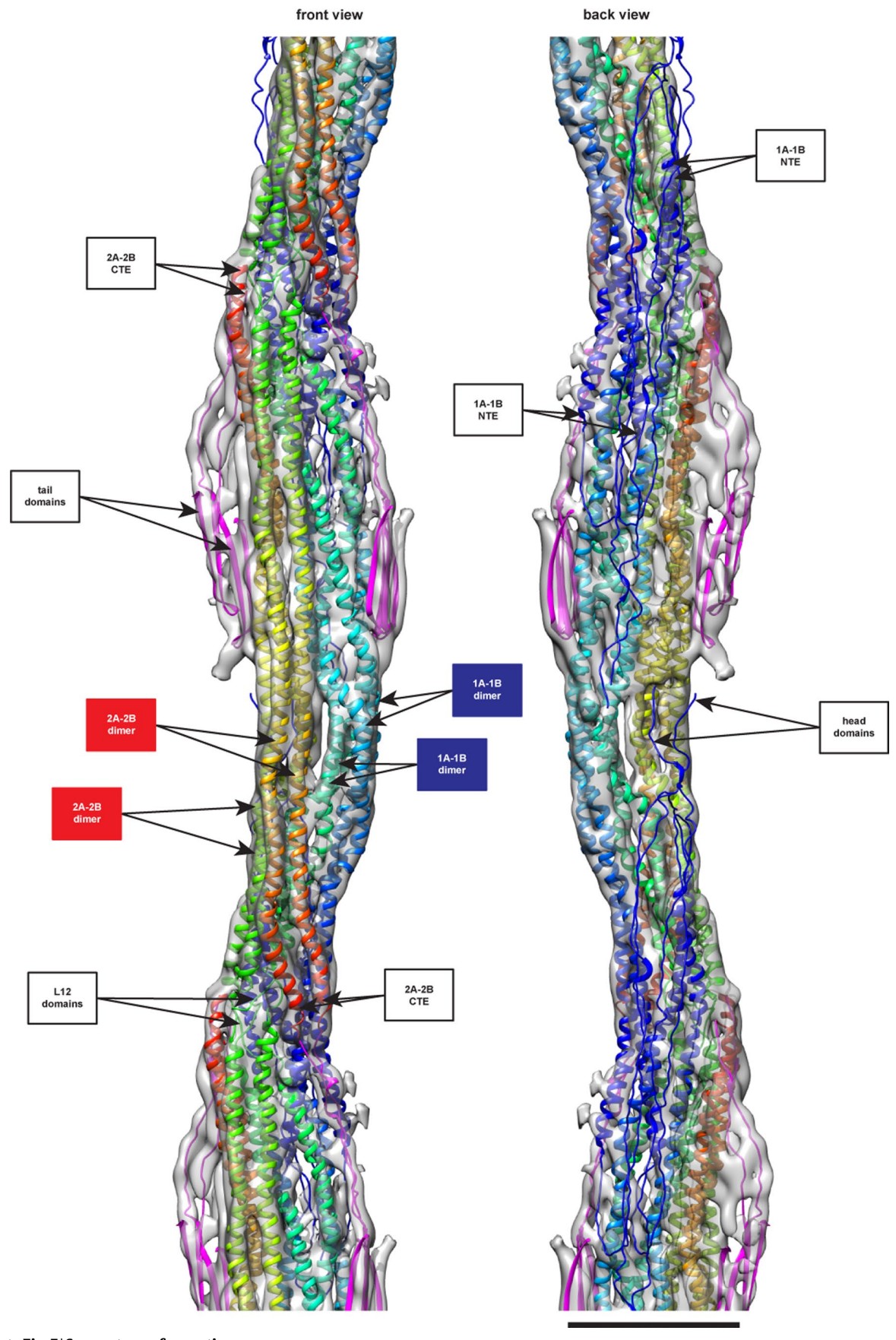

**Extended Data Fig. 7 | See next page for caption.**

**Extended Data Fig. 7 | VIF 3D atomic model zoomed on protofibril.** The complete VIF 3D atomic model was simplified to one repeating unit of a protofibril. The front view of this reduced model is displayed on the left, the view from the luminal face on the right, and the scale bar is 5 nm. The α-helical 2A-2B domains are colored from green[NTE] to red[CTE], and the α-helical 1A-1B domains are colored from blue[NTE] to green[CTE]. The tail domains are colored magenta, the head domains blue, and the linker L12 domains green. The main constituents of the repeating unit of a protofibril are 2 antiparallel 2A-2B dimers (red labels) and 2 antiparallel 1A-1B dimers (blue labels). Along a protofibril, each of the 2A-2B dimers are connected by the L12 linker domains with one of the previous and one of the subsequent 1A-1B dimers. The CTEs of each 2A-2B dimer position the tail domains to form the lateral contact sites between the protofibrils, and the NTEs of each 1A-1B dimer position the head domains to form the luminal fiber.

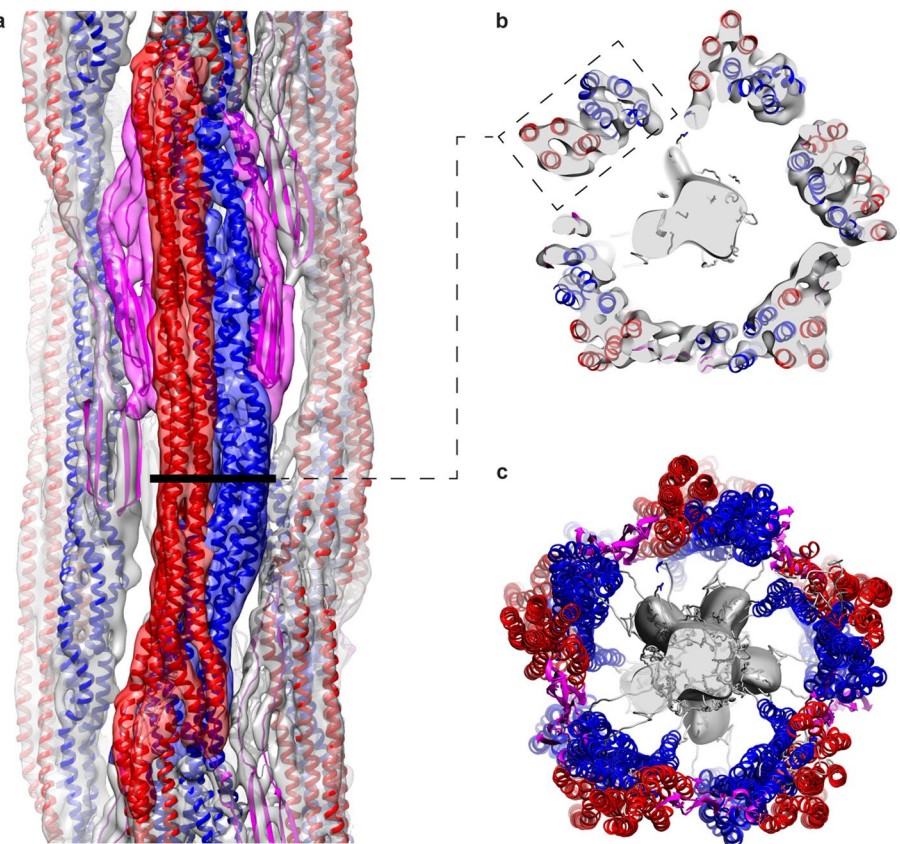

**Extended Data Fig. 8 | Outer and inner surface of VIFs.** (**a**) The frontal straight (red transparent density) and lateral curved (blue transparent density) tetrameric regions in the electron density map are formed from 2 antiparallel 2A-2B dimers (red helices) and 2 antiparallel 1A-1B dimers (blue helices), respectively, and the contact sites between the protofibrils (magenta transparent densities) are formed by the tail domains (magenta chains). (**b**) The horizontal line in (**a**) indicates the position of the respective octameric cross-section (shown within the dashed rectangle). (**c**) The outer surface of VIFs is mainly coated with the 2A-2B dimers and the inner surface with the 1A-1B dimers. The tail domains form the contact sites between the protofibrils. The head domains (grey chains) emanate into the lumen of VIFs and form the luminal fiber (grey density).

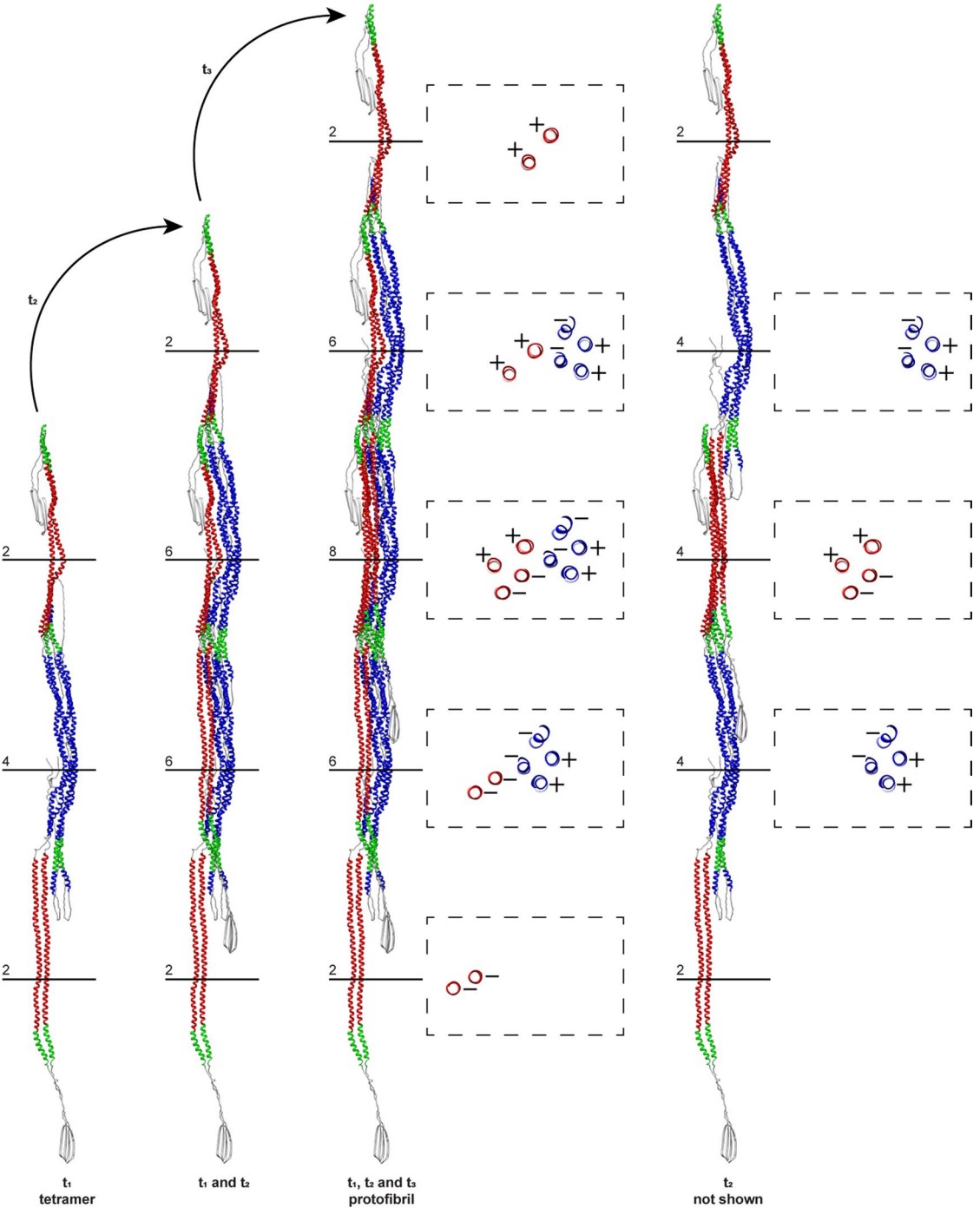

**Extended Data Fig. 9 | Construction of a protofibril.** Numbers and horizontal lines indicate the number of α-helices in cross-section at the respective positions. Cross-sections are shown within dashed rectangles. The polarity of the individual α-helices is annotated with + and - signs. Starting from a first tetramer $t_1$ (shown on the left), a second tetramer $t_2$ attaches with its 1A-1B section (blue helices) laterally to one of the flanking 2A-2B dimers (red helices) of $t_1$. This creates an intermediate assembly ($t_1$ and $t_2$) with 6 chains in cross-section. Subsequently, a third tetramer $t_3$ binds to either side of this assembly. This creates a minimal length (~110 nm) protofibril ($t_1$, $t_2$ and $t_3$) with its basic repeating unit (8 chains in cross-section) fully assembled in its center. Therefore, the full assembly of a protofibril with 8 polypeptide chains in cross-section requires the sequential interaction of at least 3 tetramers. The model shown on the right is derived from the protofibril model by removing the central tetramer $t_2$. This model shows that the central 2A-2B region is formed from 2 antiparallel 2A-2B dimers, which are provided from the 2 neighboring tetramers.

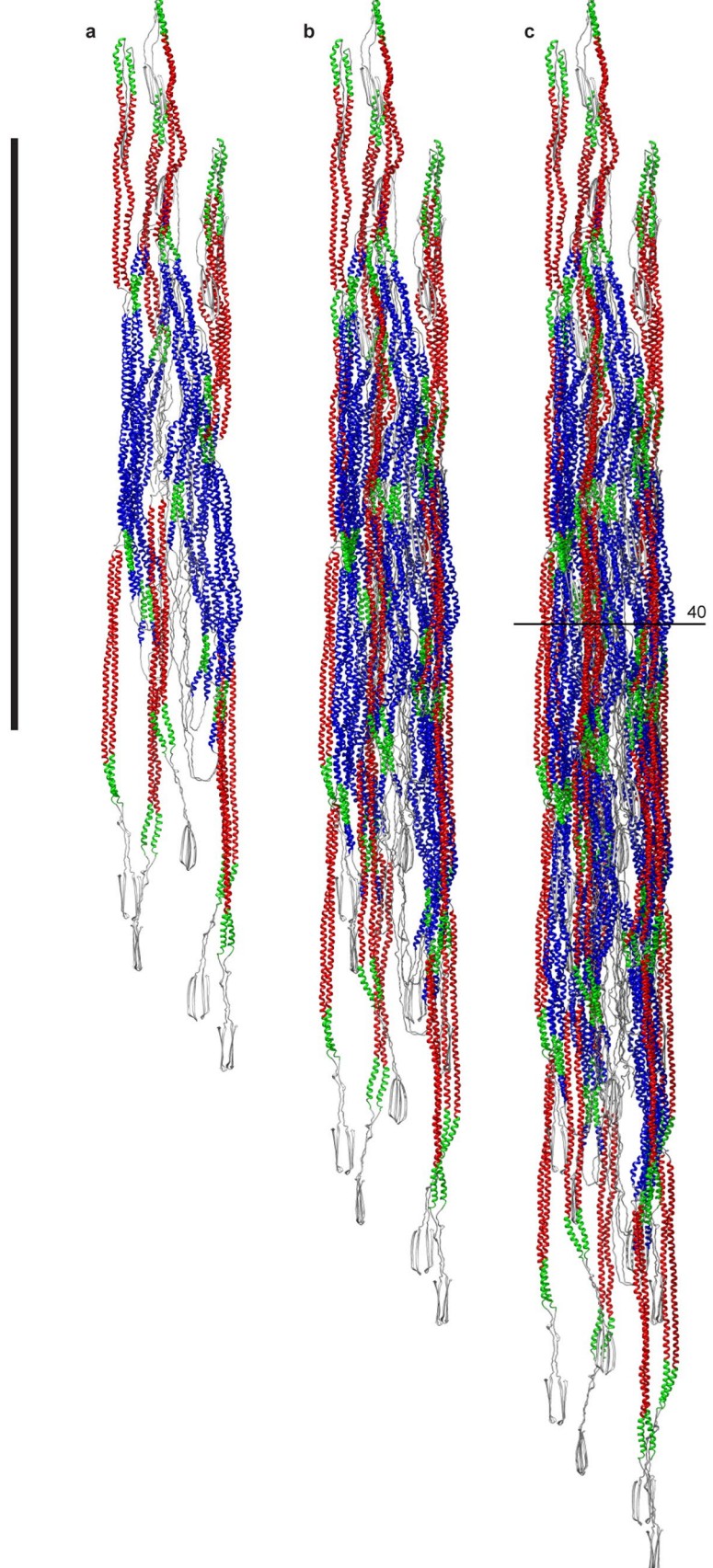

**Extended Data Fig. 10 | VIF models with increasing numbers of tetramers.** (**a**) VIF model constructed from 5 tetramers. Scale bar is 50 nm. (**b**) VIF model constructed from 10 tetramers. (**c**) VIF model constructed from 15 tetramers. In-situ VIFs assembled from at least 15 tetramers exhibit 40 polypeptide chains in cross-section through their central regions.

# Reporting Summary

## Statistics

For all statistical analyses, confirm that the following items are present in the figure legend, table legend, main text, or Methods section.

| n/a | Confirmed | |
|---|---|---|
| ☐ | ☒ | The exact sample size ($n$) for each experimental group/condition, given as a discrete number and unit of measurement |
| ☐ | ☒ | A statement on whether measurements were taken from distinct samples or whether the same sample was measured repeatedly |
| ☒ | ☐ | The statistical test(s) used AND whether they are one- or two-sided<br>*Only common tests should be described solely by name; describe more complex techniques in the Methods section.* |
| ☒ | ☐ | A description of all covariates tested |
| ☒ | ☐ | A description of any assumptions or corrections, such as tests of normality and adjustment for multiple comparisons |
| ☐ | ☒ | A full description of the statistical parameters including central tendency (e.g. means) or other basic estimates (e.g. regression coefficient) AND variation (e.g. standard deviation) or associated estimates of uncertainty (e.g. confidence intervals) |
| ☒ | ☐ | For null hypothesis testing, the test statistic (e.g. $F$, $t$, $r$) with confidence intervals, effect sizes, degrees of freedom and $P$ value noted<br>*Give P values as exact values whenever suitable.* |
| ☒ | ☐ | For Bayesian analysis, information on the choice of priors and Markov chain Monte Carlo settings |
| ☒ | ☐ | For hierarchical and complex designs, identification of the appropriate level for tests and full reporting of outcomes |
| ☒ | ☐ | Estimates of effect sizes (e.g. Cohen's $d$, Pearson's $r$), indicating how they were calculated |

*Our web collection on statistics for biologists contains articles on many of the points above.*

## Software and code

Policy information about availability of computer code

| | |
|---|---|
| Data collection | SerialEM (3.4.9, 3.8.0, 3.9.0), EPU (2.14) |
| Data analysis | MATLAB (R2019b)<br>TOM toolbox (1.0)<br>Actin Polarity Toolbox (APT, 1.0)<br>IMOD (4.7.15, 4.9.12)<br>IsoNet (0.1)<br>MotionCorr (2.1)<br>MotionCor2 (1.4.0)<br>Gctf (1.06)<br>crYOLO (1.8.2)<br>EMAN2 (1.9, 2.3)<br>RELION(3.0.8, 4.0.0)<br>UCSF Chimera (1.15)<br>Segger (2.5.3)<br>LocalDeblur (as implemented in scipion (3.0.11))<br>ResMap (1.95)<br>alphafold (2.1.2)<br>ClusPro (2.0)<br>AreTomo (1.3.3)<br>cryoSPARC (4.3) |

Clustal Omega (1.2.2)
Nikon Elements (4.5)
Phenix (1.21)
Topaz (0.2.4)

For manuscripts utilizing custom algorithms or software that are central to the research but not yet described in published literature, software must be made available to editors and reviewers. We strongly encourage code deposition in a community repository (e.g. GitHub). See the Nature Portfolio guidelines for submitting code & software for further information.

## Data

Policy information about availability of data

All manuscripts must include a data availability statement. This statement should provide the following information, where applicable:
- Accession codes, unique identifiers, or web links for publicly available datasets
- A description of any restrictions on data availability
- For clinical datasets or third party data, please ensure that the statement adheres to our policy

The VIF subtomogram average, the single particle VIF and single particle VIF-deltaT structures have been deposited in the Electron Microscopy Data Bank under the accession codes: EMD-19562, EMD-16844, EMD-19563. The VIF model have been deposited in the Protein Data Bank under the accession code PDB-8RVE. The integrative modelling protocol, initial alphafold models and the VIF tetramer model have been deposited in the PDBDEV under the accession code PDBDEV-00000212.
The UniProt code of human vimentin protein used for structure prediction with alphafold is P08670. For visualization of the complete cytoskeleton PDB models of F-actin (PDB-8A2R) and microtubules (PDB-6DPU) were used.

## Human research participants

Policy information about studies involving human research participants and Sex and Gender in Research.

| Reporting on sex and gender | N/A |
|---|---|
| Population characteristics | N/A |
| Recruitment | N/A |
| Ethics oversight | N/A |

Note that full information on the approval of the study protocol must also be provided in the manuscript.

# Field-specific reporting

Please select the one below that is the best fit for your research. If you are not sure, read the appropriate sections before making your selection.

☒ Life sciences        ☐ Behavioural & social sciences        ☐ Ecological, evolutionary & environmental sciences

For a reference copy of the document with all sections, see nature.com/documents/nr-reporting-summary-flat.pdf

# Life sciences study design

All studies must disclose on these points even when the disclosure is negative.

| Sample size | In total, 102 tomograms were acquired for the cryo-FIB/cryo-ET analysis from >3 different batches of cells. A subset of 7 tomograms was analyzed, which was sufficient to generate a subtomogram average showing the VIF protofibril stoichiometry.<br>In total, 225 tomograms were acquired of detergent-treated MEFs from >3 different batches of cells, which was sufficient to determine the helical parameters of VIFs.<br>For the VIF single particle structure 12,160 micrographs were recorded, which was sufficient to generate a subnanometer resolution structure. For the VIF-deltaT single particle structure 19,534 micrographs were recorded, which was sufficient to generate a structure which showed the position of the tail domains. |
|---|---|
| Data exclusions | Individual VIF subtomograms or single particles, which did not contribute to an improvement of resolution in subtomogram averaging or single particle analysis were excluded based on unsupervised 2D classification and 3D classification. |
| Replication | At least 3 independent, successful replications were performed for all EM measurements. |
| Randomization | For subtomogram averaging and single particle analysis the 3D refinement and averaging procedures were performed with random individual half sets. |
| Blinding | Given the fact that the structure of VIFs was unknown to any researcher involved in this study, blinding was not relevant. |

# Reporting for specific materials, systems and methods

We require information from authors about some types of materials, experimental systems and methods used in many studies. Here, indicate whether each material, system or method listed is relevant to your study. If you are not sure if a list item applies to your research, read the appropriate section before selecting a response.

## Materials & experimental systems

| n/a | Involved in the study |
|-----|----------------------|
| ☐ | ☒ Antibodies |
| ☐ | ☒ Eukaryotic cell lines |
| ☒ | ☐ Palaeontology and archaeology |
| ☒ | ☐ Animals and other organisms |
| ☒ | ☐ Clinical data |
| ☒ | ☐ Dual use research of concern |

## Methods

| n/a | Involved in the study |
|-----|----------------------|
| ☒ | ☐ ChIP-seq |
| ☒ | ☐ Flow cytometry |
| ☒ | ☐ MRI-based neuroimaging |

## Antibodies

| Antibodies used | - anti-vimentin antibody (dilution 1:200, catalog number 919101, Biolegend, USA)<br>- lamin A/C antibody (dilution 1:100, catalog number sc-376248, Santa Cruz Biotechnology, USA)<br>- anti-chicken secondary antibody (dilution 1:400, catalog number A-11039, Invitrogen, USA)<br>- anti-rabbit secondary antibody (dilution 1:400, catalog number A-11011, Invitrogen, USA) |
|---|---|
| Validation | - Validation for anti-vimentin antibody: https://www.biolegend.com/de-de/products/purified-anti-vimentin-antibody-11598<br>- Validation for lamin A/C antibody: https://www.scbt.com/p/lamin-a-c-antibody-e-1?gad_source=1&gclid=EAIaIQobChMI8N7noNiqhAMVboVoCR1qQwRwEAAYASAAEgIz2vD_BwE<br>- Validation for anti-chicken secondary antibody: https://www.thermofisher.com/antibody/product/Goat-anti-Chicken-IgY-H-L-Secondary-Antibody-Polyclonal/A-11039<br>- Validation for anti-rabbit secondary antibody: https://www.thermofisher.com/antibody/product/Goat-anti-Rabbit-IgG-H-L-Cross-Adsorbed-Secondary-Antibody-Polyclonal/A-11011 |

## Eukaryotic cell lines

Policy information about cell lines and Sex and Gender in Research

| Cell line source(s) | The MEF cell line was received from the Eriksson lab, Åbo Akademi University, Turku, Finland. |
|---|---|
| Authentication | The MEF cell line was authenticated in Virtakoivu et al. Cancer Res (2015) 75 (11): 2349–2362 (https://doi.org/10.1158/0008-5472.CAN-14-2842). |
| Mycoplasma contamination | The MEF cell line tested negative for mycoplasma contamination. |
| Commonly misidentified lines (See ICLAC register) | No commonly misidentified cell lines were used in this study. |

