## [Peer Review File · Nature Structural & Molecular Biology]

Peer Review Information

Manuscript Title: Vimentin filaments integrate low complexity domains in a complex helical structure

Corresponding author name(s): Ohad Medalia, Matthias Eibauer

Editorial Notes:

**Redactions –
unpublished data**

Reviewer Comments & Decisions:

Decision Letter, initial version:

Message: 11th Jul 2023

Dear Professor Medalia,

Thank you again for submitting your manuscript "Vimentin filaments integrate low complexity domains in a highly complex helical structure". I apologize for the delay in responding, which resulted from the difficulty in obtaining suitable referee reports. Nevertheless, we now have comments (below) from the 3 reviewers who evaluated your paper. In light of those reports, we remain interested in your study and would like to see your response to the comments of the referees, in the form of a revised manuscript.

You will see that while reviewers appreciate the results, they raise several concerns which will need to be addressed in a revision. Specifically, we agree with reviewer #2 that further investigation into the dynamics of disassembly and involvement of phosphorylation of the head domains in the process, would be of interest, if feasible. At the very least we would encourage further discussion. Moreover, both reviewers #1 and #2 ask for a clarification on the fact that tail domain removal doesn't affect filament formation. Finally, reviewer #3 lists several serious concerns regarding data quality and interpretation, which we would need to see addressed before further consideration of the manuscript, especially pertaining to experiments forming the basis of the conclusions of the protofilament stoichiometry. We agree that employing native MS could be useful in this context.

Please be sure to address/respond to all concerns of the referees in full in a point-by-point response and highlight all changes in the revised manuscript text file.

We appreciate the requested revisions are extensive. We thus expect to see your revised manuscript within 6 months. If you cannot send it within this time, please let us know. We will be happy to consider your revision as long as nothing similar has been accepted for publication at NSMB or published elsewhere. Should your manuscript be substantially delayed without notifying us in advance and your article is eventually published, the received date would be that of the revised, not the original, version.

Reporting Summary:

When submitting the revised version of your manuscript, please pay close attention to our [href="https://www.nature.com/nature-portfolio/editorial-policies/image-integrity">Digital Image Integrity Guidelines](https://www.nature.com/nature-portfolio/editorial-policies/image-integrity). and to the following points below:

Please note that all key data shown in the main figures as cropped gels or blots must be presented in uncropped form, with molecular weight markers. These data can be aggregated into a single supplementary figure. While these data can be displayed in a relatively informal style, they must refer back to the relevant figures.

SOURCE DATA: we request that authors provide, in tabular form, the data underlying the

graphical representations used in figures. This is to further increase transparency in data reporting, as detailed in this editorial (<http://www.nature.com/nsmb/journal/v22/n10/full/nsmb.3110.html>). Spreadsheets can be submitted in excel format. Only one (1) file per figure is permitted; thus, for multi-paneled figures, the source data for each panel should be clearly labelled in the Excel file; alternately the data can be provided as multiple, clearly labelled sheets in an Excel file. When submitting files, the title field should indicate which figure the source data pertains to. We ask our authors to provide source data at the revision stage, so that they are part of the peer-review process.

While we encourage the use of colour in preparing figures, please note that this will incur a charge to partially defray the cost of printing. Information about colour charges can be found at <http://www.nature.com/nsmb/authors/submit/index.html#costs>

We require deposition of coordinates (and, in the case of crystal structures, structure factors) into the Protein Data Bank with the designation of immediate release upon publication (HPUB). Electron microscopy-derived density maps and coordinate data must be deposited in EMDB and released upon publication. Deposition and immediate release of NMR chemical shift assignments are highly encouraged. Deposition of deep sequencing and microarray data is mandatory, and the datasets must be released prior to or upon publication. To avoid delays in publication, dataset accession numbers must be supplied with the final accepted manuscript and appropriate release dates must be indicated at the galley proof stage. Please find the complete NRG policies on data availability at <http://www.nature.com/authors/policies/availability.html>.

Nature Structural & Molecular Biology is committed to improving transparency in authorship. As part of our efforts in this direction, we are now requesting that all authors identified as 'corresponding author' on published papers create and link their Open Researcher and Contributor Identifier (ORCID) with their account on the Manuscript Tracking System (MTS), prior to acceptance. This applies to primary research papers only. ORCID helps the scientific community achieve unambiguous attribution of all scholarly contributions. You can create and link your ORCID from the home page of the MTS by clicking on 'Modify my Springer Nature account'. For more information please visit please visit www.springernature.com/orcid.

<https://mts-nsmb.nature.com/cgi-bin/main.plex?el=A6J7Yp5A3oal6J2A9ftduauBXfv8a6bHJt2keTnAZ>

Sincerely,

Katarzyna Ciazynska
(she/her)

Associate Editor
Nature Structural & Molecular Biology
<https://orcid.org/0000-0002-9899-2428>

Referee expertise:

Referee #1: Biochemistry, structural biology, intermediate filaments

Referee #2: Biochemistry, structural biology, intermediate filaments

Referee #3: Structural biology, integrative modelling

Reviewers' Comments:

Reviewer #1:

Remarks to the Author:

The head domains of the desmin and neurofilament light chain (NFL) IF proteins form cross-beta interactions close to the N-terminus of the two proteins, perhaps between residues 1 and 40. The head domain regions between residue 40 and the start of the alpha-helical rod domains are disordered. It is also the case that disease causing proline 8 and proline 22 mutations within the NFL head domain enhance the avidity of otherwise labile cross-beta interactions in this N-terminal portion of the NFL head domain. Figure 3E appears to show part of the head domains unstructured, and part structured. Is the part that is unstructured the part closest to the alpha-helical rod domain? Is the part that is structured in the center of the lumen interpreted to be at the N-terminal end of the vimentin protein? If so, this structure would appear to fit beautifully with both human genetic and pre-existing biophysical and biochemical data.

Reviewer #2:

Remarks to the Author:

Eibauer, Medalia and colleagues describe the 3D structure of mature vimentin intermediate filaments at 7.2 Angstroms resolution, identifying that the N-terminal heads form a central luminal fibre and the C-terminal tails regulate lateral interactions necessary for protofibril assembly. These insights provide an important advance in understanding the architecture of vimentin. I appreciated the clear explanation of why one filament wall appears brighter in the EM class averages – your non-structural biology readers will need this level of clarity to understand your manuscript. Along those lines, there are several items that need to be addressed to make this manuscript suitable for a general scientific audience.

Major:

1. Intro, Page 4, line 61. Please clarify why adding ion-beam milling to the procedural protocol with Cryo-ET helped you obtain previously unknown helical symmetry parameters – explain to the readers why this approach succeeded where others did not.

2. Intro, Page 4, line 82. It seems appropriate to also cite: Sokolova AV, Kreplak L, Wedig

T, Mücke N, Svergun DI, Herrmann H, Aebi U, Strelkov SV. Monitoring intermediate filament assembly by small-angle x-ray scattering reveals the molecular architecture of assembly intermediates. *Proc Natl Acad Sci U S A*. 2006 Oct 31;103(44):16206-11.

3. Do you believe the use of urea and guanidine to refold recombinant vimentin and tailless vimentin altered the structure?

4. Page 6, line 135. Not all IF enthusiasts will recognize that at 7.2 Å resolution we can understand global protein properties, but the exact side chain interactions driving the filament assembly remain unknown at this level of resolution. Please provide clarity to the readers the limitation of the current model resolution (it doesn't necessarily have to be at this exact location in the manuscript – often limitations are described in the Discussion).

5. Page 7, lines 148-159. A strength of this manuscript is the extra effort to analyze the tail-less VIFs and precisely determine their location and function. However, in Figure 2 the lateral contacts/tails are depicted as yellow, but in Figure 3 they are magenta, and in Figure 4 they are gray. One of the major impediments to an average reader understanding your manuscript is the inconsistent use of colors. I highly recommend you pick one set of colors to represent the head, 1A-1B, L12, 2A-2B, and tail throughout all structural images in the manuscript, including the Extended Data/Supplemental Figures. This is absolutely necessary for readers to build/layer knowledge points you make onto one another and more easily transition figure to figure without feeling lost. In other words, find one color scheme and make it stick throughout the whole manuscript.

6. Figure 2, panel E. The red coloring supposedly representing increased plasticity is barely visible at all. This image does not appear to convey the information you are trying to share. Please find an alternate way to show this, or else your point is lost on the reader.

7. The authors go to significant lengths to explain the frontal straight tetrameric helix bundle compared to the curved tetrameric helix bundle (Figure 2, 3). However, the tetrameric composition of these species are poorly illustrated, and there is little clarity as to how the 1A-1B helices relate to the 2A-2B helices – are the straight and curved tetrameric species from the same 4 vimentin proteins? Or are they each formed from 4 different polypeptide strands, meaning 8 total vimentin proteins? Given how important this architecture is to the formation of the outer vs inner surface of the VIF, there needs to be additional images explaining how each of these species is formed and what constitutes each. Please create a figure that transitions the reader from Fig 2D to 3B/C – right now the leap is not entirely clear.

8. Along the same lines as the previous comment, this manuscript would be greatly improved by having a figure that shows, within the VIF fibre, each of the following states: single vimentin polypeptide chain, vimentin heterodimer, vimentin tetramer, and vimentin octamer. A panel like Figure 3B all in gray except highlighting the path a single protein, then a dimer, a tetramer, and octamer would help synthesize decades of models of VIF assembly and more clearly help readers understand the filament formation and distinguish the single polypeptide path from the straight and curved tetrameric species discussed above. If the authors wanted to go the extra mile, a supplemental figure highlighting single A11, A12, and A22 tetramers within the fiber model could help unify decades of biophysics. There was no comment on ACN interaction – do the authors conclude this does not exist in VIFs?

9. Figure 4B would be more effective if you had two panels side by side – one with the consistent color scheme throughout the paper and one where each tetramer was a solid single color, so the layering of tetramers would be more clear.
10. Page 9, lines 210-211 (and lines 265-266): a multiple sequence alignment of type III IFs, a few select Type I/II, IV, and V IFs should be shown to demonstrate the conservation of the interlocking mechanism at the primary sequence level.
11. For extended data Fig 9 please color heads and tails the appropriate “consistent” color you choose for the manuscript so that the polarity of the protofibril assembly (parallel vs antiparallel) is readily apparent (since there are no N- and C- terminal labels).
12. Ext Data Fig 12 – the labels at the top (“no missing wedge”...) seem to be reversed based on the Fig legend description.
13. Page 11, lines 265-271. One of the most important findings of the manuscript is the molecular interlock region, yet other than the mention of Y117L, there is no other mutational information discussed. This is a major oversight – the next paragraph between line 271 and 272 should discuss the human diseases caused by mutation specifically in the molecular interlock region, and a supplementary table added listing the known mutations located in the interlock region. This information will provide more substantial clinical impact to this work.
14. Lines 27, 68: “form amyloid-like fiber” – this term is used, but it is not really shown in any images that the head regions are truly amyloid like. At current resolution beta-sheets can’t be resolved. Consider “proposed to be amyloid-like structures” or acknowledge the limitation. If you have other evidence it is “amyloid fiber” like, then please present it.
15. Line 71: “Mouse embryonic fibroblasts (MEFs) are widely used...” Do they (MEFs) have other IFs expressed, if yes, how authors are sure that they are looking at VIMs? If only VIM expressed, I suggest to mention it.
16. Line 168: “...on the alphafold prediction of the full-length vimentin dimer (Extended Data Fig. 7) “ On this figure the Alpha-fold model should be originally colored from the program and the confidence bar (colors) should be presented.
17. Line 277: “We suggest that a unit-length VIF is assembled from 5 tetramers and that this assembly is driven by a phase-separation process which facilitates the initial association of the tetramers via their head domains. Therefore, the head domains may act as initial nucleators of VIF filament assembly”. When two tetramers come together, is concentration of the head domains high enough for phase-separation? Can the authors be more specific on phase-separation and how it results in “aggregating to the luminal fiber” (line 275). If this is true, can authors also have some model for VIM disassembly and dynamics, which require head domain phosphorylation? It looks like phase separated head domains are inside the filament and can’t be accessed by kinases?
18. One curious question arising from the manuscript, that is not discussed, and follows the above remark on the head domains, is that there is a substantial amount of literature discussing the regulation of VIM assembly through phosphorylation of the head domains. It would seem that the current structural model with the head domains forming an interior

fiber, would make phosphorylation regulation of VIFs difficult. How do the authors reconcile the body of literature regarding head domain phosphorylation and the model? Is all phosphorylation occurring pre-VIF? How could post-VIF phosphorylation work in this model? If phosphorylation takes place after assembly, the heads should be accessible for PKA (cAMP- dependent kinase), correct?

Some literature to consider on phosphorylation:

Eriksson, J.E., et al., Specific in vivo phosphorylation sites determine the assembly dynamics of vimentin intermediate filaments. *J Cell Sci*, 2004. 117(Pt 6): p. 919-32.
 ...IFs are highly dynamic structures, the constituent proteins of which undergo active exchange between a major compartment of assembled IF polymers and a very small fraction of disassembled IF subunits. In order to achieve these dynamic properties, IF proteins require an active support mechanism that maintains the exchange of subunits between the assembled and disassembled protein fractions. Reversible phosphorylation is a mechanism that could plausibly maintain this kind of exchange. ...
 ...both the equilibrium between vimentin polymers and depolymerized subunits and the turnover of subunit exchange are regulated by kinase-phosphatase equilibria. This regulation is based upon reversible phosphorylation of a number of phosphorylation sites located mainly in the N-terminal region of vimentin. (HEAD DOMAIN)
 ... results show that PP1 and PP2A inhibition results in rapid elevation of vimentin phosphorylation, with subsequent disassembly of the vimentin IFs.
 From our results, it appears that the elevated phosphorylation first induces a net fragmentation of the polymers and, when the phosphorylation is more markedly elevated, then induces a net release of soluble subunits.
 Our results demonstrated that phosphorylation releases the same oligomeric peptide composition both in vitro and in vivo; tetrameric oligomers were released both by PKA phosphorylation in vitro and by inhibition of dephosphorylation in vivo. ...Our present results, showing that phosphorylation induces disassembly into tetrameric subunits, could imply that the phosphorylation- driven subunit exchange between IF polymers and soluble subunits would take place in the form of a tetramer.
 Ser-4, Ser-6, Ser- 7, Ser-8, Ser-9, Ser-38, Ser-41, Ser-71, Ser-72, Ser-418, Ser- 429, Thr-456, and Ser-457 as significant in vivo phosphorylation sites.
 ...However, the absence of two major phosphorylation sites, Ser-38 and Ser-72, did not have a major effect on filament assembly, indicating that in regulating assembly equilibria, the role of phosphorylation is primarily to drive disassembly.

Kraxner, J., et al., Post-translational modifications soften vimentin intermediate filaments. *Nanoscale*, 2021. 13(1): p. 380-387.
 ... We find that the filaments soften with increasing amount of phosphorylated protein within the filament and that interaction with the protein 14-3-3 further enhances this effect.
 phosphorylated sites are dispersed through- out the whole protein, but the most abundant ones ($\log_2(I) > 0$, red lines) are all found in the head region of vimentin. ...Strikingly, the positions S71 and S72 are always phosphorylated simultaneously and show the highest degree of phosphorylation.
 When vimentin becomes phosphorylated, the positive charges of the head domain are flanked by negative charges of the phosphorylated amino acids as shown in Fig. 4a, which diminishes the electrostatic attraction between the head and the coiled-coils.

Head domain of vimentin is positively charged (Kraxner, J. and S. Koster, Influence of

phosphorylation on intermediate filaments. *Biol Chem*, 2023) is the surface charge of the interior wall of the filament negative?

...Phosphorylation at specific sites of vimentin leads to disassembly, which is a necessary prerequisite for cell division during mitosis (Eriksson et al. 1992; Inagaki et al. 1996). In both vimentin and desmin the preferred phosphorylation sites lie in the head domain (Geisler et al. 1989; Geisler and Weber 1988) that is also crucial for filament assembly as shown in experiments with headless vimentin, which is unable to assemble (Rogers et al. 1995).

Early studies revealed that phosphorylation of vimentin at Ser55 by p34cdc2 kinase is responsible for the disassembly during mitosis (Chou et al. 1990), and later studies identified additional interphase-specific phosphorylation sites (Sihag et al. 2007). Phosphorylation of IF proteins does not solely provide a mechanism for reversible assembly of filaments but rather offers a huge variety of regulation mechanisms ranging from remodeling of IF networks (Chou et al. 1989; Eriksson et al. 2004), cell migration (Chung et al. 2013), changes in mechanical properties (Kraxner et al. 2021), interactions with other protein structures (Sihag et al. 2007) or roles in signaling pathways (van Engeland et al. 2019).

Minor:

19. Figure 1 C – how was this image created? Please clarify in figure legend or in methods.

20. Figure 3E is my favorite image in the whole manuscript. With a little tweaking of the colors so that the lateral tail contacts pop out more, this could be worthy of a journal cover image.

Reviewer #3:

Remarks to the Author:

The results presented by Eibauer et al. are of interest to people in the intermediate filament/cytoskeleton, cryoEM helical reconstruction and cancer research fields. If all above remarks will be addressed, the manuscript can be considered for publication. The text is very readable and concise enough.

However, aside from the very detailed modelling part, that was extremely interesting and convincing, the quality of the data needs to improve substantially, and results should be more convincing.

The manuscript presents results of the yet elusive full 3D structure of assembled vimentin intermediate filaments (VIFs). From cross sections of an in situ tomogram, the authors claim to VIFs composed of 5 protofibrils, stoichiometry that is in contrast to the previously reported 4-protofibril VIFs structure seen previously in cryoEM by Goldie et al., 2007. The claim is not convincing, though, and that is the main flaw: both power spectra are confusing and layer lines hard to discern. In the first one (Fig.1F), very pixelated probably because of the noise in the rest of the tomogram, lines 3 and 4 are not lying on defined layer lines, as for line 1 and 2; in the supplementary power spectrum (Extended Data Fig. 4C), the cyan arrow at helical rise points to no clear line, while the first visible meridian layer line is localized lower than the cyan arrow.

Moreover, the use of a software (IsoNet) to overcome the missing wedge limitation in the cryo-FIB milled tomogram (Movie 1) seem to enhance missing-wedge artefacts instead of attenuating them (striped lines at the left edge of the tomogram Movie 1, middle frames, and in the same region in Extended Data Fig. 1A).

Thus, the analysis of features from missing-wedge corrected tomogram might not be reliable. For this reason, lines 236-237: "Without correction, the difference between 4 and 5 protofibrils could not be resolved" and Extended Data Figure 12: "[...] cryo-ET without missing wedge deconvolution can resolve only 4 protofibrils in cross-section", are dangerous statements in light that the tomogram with artificially filled missing wedge displays more artefacts. Also, the overall quality of the cryoEM tomogram and the fibers within is lower compared to the cryoEM tomogram and VIFs in detergent-treated cells, that has no reported missing wedge "correction": VIFs there look much more detailed and organic. It would be good to show the original unprocessed tomogram, or more, and what stoichiometry is extrapolated from it. Native mass-spec can possibly help redeem the 4 or 5 protofilament stoichiometry conundrum of VIFs, despite them being reported to be structurally very polymorphic.

Another point that should be clarified in the study is what are the contacts that keep the structure intact in the tailless maps and models (Extended Data Fig. 6)? Only the low complexity head domains that aggregate in the central amyloid-like fibre of the filament? What is then the function of tail domains in VIMs if VIM- Δ T is still able to form filaments?

The final point is regarding the discussion lines 278-280, where authors declare that "VIF [...] assembly is driven by a phase-separation process which facilitates the initial association of the tetramers via their head domains.". This is an unannounced speculation that was never introduced nor discussed previously in the manuscript as an investigation hypothesis and has no supporting data from the static three-dimensional structure and model reported. And no references are provided to previous publications where VIFs nucleation and assembly via phase-separation was observed. Therefore, it should either be proven or removed, or at least greatly toned down.

other points:

- Structure-based predictions should be provided along with confidence scores
- A few more tomograms could be analyzed/shown and/or number of experiments should be indicated.

Reviewer #1 attachment:

The manuscript authored by Eibauer and colleagues reports upon structural studies of vimentin intermediate filaments. The authors utilized methods of cryo-focused ion beam milling, cryo-electron microscopy and tomography to resolve the three-dimensional structure of assembled vimentin intermediate filaments. The manuscript describes exceptionally rigorous and beautiful science. To my knowledge, this study represents the first structural model for assembled intermediate filaments that is likely to be accurate.

The structural model deduced by the authors shows that fully assembled intermediate filaments are composed of five proto-filaments, each containing eight individual polypeptides. Each protofilament is held together primarily via interactions among α -helical segments long appreciated to represent the dominant structural feature of intermediate filaments. The five protofilaments are organized to form a cylindrical structure. The lumen of the structure contains IF head domains believed to be organized via

labile cross- β interactions to form an amyloid-like core. Finally, partially ordered tail domains are shown to be positioned so as to facilitate favorable interactions among proto-filaments. Tail domains are further interpreted to decorate the exterior surface of assembled vimentin intermediate filaments.

Unless I am mistaken, this work represents a landmark contribution to the field of cell biology. The structures of polymerized actin filaments and assembled microtubules were resolved decades ago as watershed discoveries in the field of cell biology. Whereas many conceptual models of assembled intermediate filaments have been proposed over the years, this paper hits the jackpot in showing us how the third and final form of cytoskeletal polymers is assembled at a molecular level. From the perspective of this reviewer, this paper should appear in Nature – it is of huge significance.

Beyond resolving the molecular architecture of assembled intermediate filaments, this work offers the first example showing how disordered protein domains affect the morphology of a defined cellular structure. It has long been known that the head and tail domains of all of our 75 intermediate filament proteins represent intrinsically disordered polypeptide segments. Functional studies have shown that IF proteins cannot assemble if the head domain is removed. This study offers an incredibly simple description of how IF head domains actually work – they protrude into the lumen created by the five proto-filaments and there assemble into an amyloid-like polymer. This represents a particularly exciting discovery that will have implications far beyond the intermediate filament field. Eukaryotic cells contain thousands of protein domains that are of intrinsic disorder. It is highly likely that many such domains will utilize labile, amyloid-like interactions to affect cell morphology, including the permeability channel of nuclear pores and scores of nuclear and cytoplasmic puncta not surrounded by investing membranes. I have but three recommendations for revision. First, the authors should establish whether their own data reveal amyloid-like interactions by head domains within the lumen of assembled intermediate filaments. Figure 3E appears to show ribbon-like representations organized in a cylindrical conformation. Are cross- β interactions observed to underlie the formation of this cylindrical structure, or is it instead the case that the authors offer this conclusion based solely upon work from the McKnight lab showing that the head domains of many different IF proteins self-interact via the formation of labile cross- β interactions? Whereas either of these options will be fine, it would be useful for this uncertainty to be clarified.

My second recommendation is that the authors consider commenting upon the fact that removal of the tail domain does not impede formation of their assembled intermediate filaments. I say this for two reasons. First, this observation gives evidence that the observed tail domain interactions among protofilaments are not essential for IF assembly. Second, the McKnight lab reported in Cell evidence that the fused-in-sarcoma (FUS) RNA binding protein can interact with assembled vimentin intermediate filaments in a manner yielding repetitive puncta separated by 45 nanometers (Lin et al., 2016). Such interactions were postulated to be representative of the sorts of interactions that allow RNA granules to bind assembled IFs as demonstrated by Pondel and King in their PNAS paper published in 1988. The observed, periodic interaction of FUS along the surface of assembled filaments was lost if filaments were prepared from a tail-deleted form of the vimentin protein. Is the architectural disposition of tail segments

in the authors' molecular model concordant with the pattern of FUS binding described by the McKnight laboratory?

Third, and finally, the authors should recognize that the lava-hot "phase separation" field has fully rejected the idea that protein domains of low sequence complexity self-associate in a biologically valid manner via the formation of labile cross- β structures. As such, it can be anticipated that many people will dismiss the claims of the authors that IF head domains function by forming a labile, amyloid-like polymer within the lumen of their assembled vimentin IFs.

Anticipating this blowback, the authors should add to their discussion several of the following points. First, amyloid-like polymers made from LC domains, unlike pathogenic prions, are fully labile to disassembly. This feature of LC domain polymers was first demonstrated by the Kato and Han papers published by the McKnight laboratory in *Cell* in 2012. Second, a high resolution structure of polymers made from the FUS LC domain shows precisely why the polymers are labile to disassembly – they have zero hydrophobic amino acid residues lining the protomer:protomer interface. This feature differs drastically as compared with pathogenic prions, including the $\alpha\beta$ and α -synuclein polymers that are replete with hydrophobic residues at the protomer:protomer interface. The solid-state NMR structure of FUS polymers was resolved by the laboratory of Robert Tycko (Murray et al., 2017). The Tycko *Cell* paper offers multiple lines of evidence explaining why FUS polymers are labile to disassembly and likely to be biologically valid. Finally, the authors should make reference to a beautiful structural study showing that the intrinsically disordered FG domains of nucleoporin proteins likely function via the formation of labile cross- β interactions. This study came from the laboratories of Gunnar Schroder and Markus Zweckstetter as published in *Nature Chemistry* (Opakua et al., 2022).

I conclude with the prediction that the present contribution by Eibauer and colleagues will emerge as a landmark discovery clarifying how enigmatic protein domains of low sequence complexity actually work. The proteomes of eukaryotic cells contain thousands of LC domains, many of which are likely to function in a similar manner as to what the authors propose in this beautiful piece of science. If placed in proper context, the impact of this study will be justifiably enhanced.

Author Rebuttal to Initial comments

Reviewer #1:

The manuscript authored by Eibauer and colleagues reports upon structural studies of vimentin intermediate filaments. The authors utilized methods of cryo-focused ion beam milling, cryo-electron microscopy and tomography to resolve the three-dimensional structure of assembled vimentin intermediate filaments. The manuscript describes exceptionally rigorous and beautiful science. To my knowledge, this study represents the first structural model for assembled intermediate filaments that is likely to be accurate.

The structural model deduced by the authors shows that fully assembled intermediate filaments are composed of five protofilaments, each containing eight individual polypeptides. Each protofilament is held together primarily via interactions among α -helical segments long appreciated to represent the dominant structural feature of intermediate filaments. The five protofilaments are organized to form a cylindrical structure. The lumen of the structure contains IF head domains believed to be organized via labile cross- β interactions to form an amyloid-like core. Finally, partially ordered tail domains are shown to be positioned so as to facilitate favorable interactions among protofilaments. Tail domains are further interpreted to decorate the exterior surface of assembled vimentin intermediate filaments.

Unless I am mistaken, this work represents a landmark contribution to the field of cell biology. The structures of polymerized actin filaments and assembled microtubules were resolved decades ago as watershed discoveries in the field of cell biology. Whereas many conceptual models of assembled intermediate filaments have been proposed over the years, this paper hits the jackpot in showing us how the third and final form of cytoskeletal polymers is assembled at a molecular level. From the perspective of this reviewer, this paper should appear in *Nature* – it is of huge significance.

Beyond resolving the molecular architecture of assembled intermediate filaments, this work offers the first example showing how disordered protein domains abet the morphology of a defined cellular structure. It has long been known that the head and tail domains of all of our 75 intermediate filament proteins represent intrinsically disordered polypeptide segments. Functional studies have shown that IF proteins cannot assemble if the head domain is removed. This study offers an incredibly simple description of how IF head domains actually work – they protrude into the lumen created by the five protofilaments and there assemble into an amyloid-like polymer. This represents a particularly exciting discovery that will have implications far beyond the intermediate filament field. Eukaryotic cells contain thousands of protein domains that are of intrinsic disorder. It is highly likely that many such domains will utilize labile, amyloid-like interactions to abet cell morphology, including the permeability channel of nuclear pores and scores of nuclear and cytoplasmic puncta not surrounded by investing membranes.

We thank the reviewer for realizing the importance of this work and for the kind words.

I have but three recommendations for revision. First, the authors should establish whether their own data reveal amyloid-like interactions by head domains within the lumen of assembled intermediate filaments. Figure 3E appears to show ribbon-like representations organized in a cylindrical conformation. Are cross- β interactions observed to underlie the formation of this cylindrical structure, or is it instead the case that the authors offer this conclusion based solely upon work from the McKnight lab showing that the head domains of many different IF proteins self-interact via the formation of labile cross- β interactions? Whereas either of these options will be fine, it would be useful for this uncertainty to be clarified.

We thank the reviewer for this comment. We cannot currently directly observe cross- β interactions and the precise folding of the head domains in our map, due to the relatively low resolution of the luminal fiber region. Nevertheless, the well-resolved 1A-1B domains clearly indicate that the head domains protrude into the lumen of the filament, so it fits that they form the luminal fiber. However, the conclusion that the luminal fiber is an amyloid-like fiber and therefore stabilized on a molecular level by cross- β interactions is based on the work by McKnight and coworkers (Lin et al., 2016). We found the similarity of the vimentin-head filaments (shown in Figure S5 in this article) with the luminal fiber very intriguing. The fact that vimentin-head filaments show the typical X-ray diffraction pattern of cross- β interactions is the basis of our speculation.

We have modified the text in the discussion accordingly (see lines 304-321 in the revised manuscript), so that the resolution limit of our structure and the basis of the amyloid-like fiber possibility is clarified for the reader.

The head domains of the desmin and neurofilament light chain (NFL) IF proteins form cross-beta interactions close to the N-terminus of the two proteins, perhaps between residues 1 and 40. The head domain regions between residue 40 and the start of the alpha-helical rod domains are disordered. It is also the case that disease causing proline 8 and proline 22 mutations within the NFL head domain enhance the avidity of otherwise labile cross-beta interactions in this N-terminal portion of the NFL head domain. Figure 3E appears to show part of the head domains unstructured, and part structured. Is the part that is unstructured the part closest to the alpha-helical rod domain? Is the part that is structured in the center of the lumen interpreted to be at the N-terminal end of the vimentin protein? If so, this structure would appear to fit beautifully with both human genetic and pre-existing biophysical and biochemical data.

Yes, that is exactly how the model of the luminal fiber is built. The part that appears structured spans residues 1-60, the unstructured part bridges between the luminal fiber to the inner wall of the filament (residues 61-85), and the alpha-helical rod domain starts with residue 86.

My second recommendation is that the authors consider commenting upon the fact that removal of the tail domain does not impede formation of their assembled intermediate filaments. I say this for two reasons. First, this observation gives evidence that the observed tail domain interactions among protofilaments are not essential for IF assembly. Second, the McKnight lab reported in *Cell* evidence that the fused-in-sarcoma (FUS) RNA binding protein can interact with assembled vimentin intermediate filaments in a manner yielding repetitive puncta separated by 45 nanometers (Lin et al., 2016). Such interactions were postulated to be representative of the sorts of interactions that allow RNA granules to bind assembled IFs as demonstrated by Pondel and King in their *PNAS* paper published in 1988. The observed, periodic interaction of FUS along the surface of assembled filaments was lost if filaments were prepared from a tail-deleted form of the vimentin protein. Is the architectural disposition of tail segments in the authors' molecular model concordant with the pattern of FUS binding described by the McKnight laboratory?

We thank the reviewer very much for this comment. We believe that the tail domains add a structural regulatory layer to the filaments, controlling how accessible the luminal fiber is. In our structure the tail domains are evenly distributed between the protofibrils. Assuming that the tail domains interact to form a minimal number of clusters along the protofibrils, the distance between these tail domain clusters would be around 47 nm, in perfect agreement with the work of the McKnight lab (Lin et al., 2016). In this tail domain configuration, the filament would be more open and the luminal fiber more accessible.

Ultimately, if the tail domains were completely detached from the protofibrils, access to the luminal fiber would be increased.

We added the tail clustering model as a new figure in the revised manuscript (Extended Data Fig. 18), which is now described in the discussion (see lines 328-339 in the revised manuscript).

Third, and finally, the authors should recognize that the lava-hot “phase separation” field has fully rejected the idea that protein domains of low sequence complexity self-associate in a biologically valid manner via the formation of labile cross- β structures. As such, it can be anticipated that many people will dismiss the claims of the authors that IF head domains function by forming a labile, amyloid-like polymer within the lumen of their assembled vimentin IFs. Anticipating this blowback, the authors should add to their discussion section the following points. First, amyloid-like polymers made from LC domains, unlike pathogenic prions, are fully labile to disassembly. This feature of LC domain polymers was first demonstrated by the Kato and Han papers published by the McKnight laboratory in *Cell* in 2012. Second, a high resolution structure of polymers made from the FUS LC domain shows precisely why the polymers are labile to disassembly – they have zero hydrophobic amino acid residues lining the protomer:protomer interface. This feature differs drastically as compared with pathogenic prions, including the $\alpha\beta$ and α -synuclein polymers that are replete with hydrophobic residues at the protomer:protomer interface. The solid-state NMR structure of FUS polymers was resolved by the laboratory of Robert Tycko (Murray *et al.*, 2017). The Tycko *Cell* paper offers multiple lines of evidence explaining why FUS polymers are labile to disassembly and likely to be biologically valid.

Finally, the authors should make reference to a beautiful structural study showing that the intrinsically disordered FG domains of nucleoporin proteins likely function via the formation of labile cross- β interactions. This study came from the laboratories of Gunnar Schroder and Markus Zwecksteter as published in *Nature Chemistry* (Opakua *et al.*, 2022).

We thank the reviewer very much for these comments and ideas to improve the discussion, which we have added to the revised manuscript (see lines 311-321).

I conclude with the prediction that the present contribution by Eibauer and colleagues will emerge as a landmark discovery clarifying how enigmatic protein domains of low sequence complexity actually work. The proteomes of eukaryotic cells contain thousands of LC domains, many of which are likely to function in a similar manner as to what the authors propose in this beautiful piece of science. If placed in proper context, the impact of this study will be justifiably enhanced.

Thank you very much for your positive feedback. We hope your prediction will come true!

Reviewer #2:

Eibauer, Medalia and colleagues describe the 3D structure of mature vimentin intermediate filaments at 7.2 Angstroms resolution, identifying that the N-terminal heads form a central luminal fibre and the C-terminal tails regulate lateral interactions necessary for protofibril assembly. These insights provide an important advance in understanding the architecture of vimentin. I appreciated the clear explanation of why one filament wall appears brighter in the EM class averages – your non-structural biology readers will need this level of clarity to understand your manuscript. Along those lines, there are several items that need to be addressed to make this manuscript suitable for a general scientific audience.

We thank the reviewer for realizing the importance of this work and the effort that was put into the comments below, which we highly appreciate.

Major:

1. Intro, Page 4, line 61. Please clarify why adding ion-beam milling to the procedural protocol with Cryo-ET helped you obtain previously unknown helical symmetry parameters – explain to the readers why this approach succeeded where others did not.

We thank the reviewer for this comment. We analyzed four datasets in this work, the cryo-FIB/cryo-ET dataset, the detergent-treated MEF dataset, and two single particle datasets obtained from in-vitro polymerized VIFs (full-length and Δ Tail). We used the cryo-FIB/cryo-ET data to study the structure of VIFs in-situ, in particular their native protofibril stoichiometry. In this respect, the cryo-FIB/cryo-ET method is suitable, since it excludes the uncertainty that the detergent treatment in the second dataset or the in-vitro polymerization in the single particle datasets could alter the protofibril stoichiometry. Therefore, we can conclude that our VIF model with 5 protofibrils describes at least one of the native occurring polymerization states of VIFs in cells. We added this point to stress the relevance of the cryo-FIB method (see lines 258-268 in the revised manuscript).

2. Intro, Page 4, line 82. It seems appropriate to also cite: Sokolova AV, Kreplak L, Wedig T, Mücke N, Svergun DI, Herrmann H, Aebi U, Strelkov SV. Monitoring intermediate filament assembly by small-angle x-ray scattering reveals the molecular architecture of assembly intermediates. Proc Natl Acad Sci U S A. 2006 Oct 31;103(44):16206-11.

We added this citation to the revised manuscript (line 84).

3. Do you believe the use of urea and guanidine to refold recombinant vimentin and tailless vimentin altered the structure?

This is clearly an important point. In the micrographs of in-vitro polymerized VIFs, we see features that are not seen in cells. For example, no unwinding of VIFs or unravelling of the filaments is seen inside cells, nor large diameter changes. While these features are only seen in-vitro, by analyzing a large number of VIF segments we extracted a well-structured and uniform VIF assembly, which resembles the native 5 protofibril architecture of VIFs in cells. Nevertheless, we currently do not know the reasons why the in-vitro assembled filaments appear much more polymorphic and heterogeneous in contrast to VIFs polymerized in cells, and to understand this better would be the goal of a future project.

4. Page 6, line 135. Not all IF enthusiasts will recognize that at 7.2 Å resolution we can understand global protein properties, but the exact side chain interactions driving the filament assembly remain

unknown at this level of resolution. Please provide clarity to the readers the limitation of the current model resolution (it doesn't necessarily have to be at this exact location in the manuscript – often limitations are described in the Discussion).

The reviewer is absolutely correct! We added this point to the revised discussion (line 304-310).

5. Page 7, lines 148-159. A strength of this manuscript is the extra effort to analyze the tail-less VIFs and precisely determine their location and function. However, in Figure 2 the lateral contacts/tails are depicted as yellow, but in Figure 3 they are magenta, and in Figure 4 they are gray. One of the major impediments to an average reader understanding your manuscript is the inconsistent use of colors. I highly recommend you pick one set of colors to represent the head, 1A-1B, L12, 2A-2B, and tail throughout all structural images in the manuscript, including the Extended Data/Supplemental Figures. This is absolutely necessary for readers to build/layer knowledge points you make onto one another and more easily transition figure to figure without feeling lost. In other words, find one color scheme and make it stick throughout the whole manuscript.

We apologize for the confusion. It was a mistake that the lateral protofibril contacts (tail domains) were colored yellow in the repeating unit segmentation, and not magenta as they should have been. We have corrected this mistake in the revised manuscript (see Fig. 2D). The main color scheme to present the VIF model is introduced in Fig. 3. In the explanation of the building blocks of VIFs in Fig. 4 we use a simplified version of the main color scheme, to focus the attention on the 1A-1B and 2A-2B domains, and the interlock region.

6. Figure 2, panel E. The red coloring supposedly representing increased plasticity is barely visible at all. This image does not appear to convey the information you are trying to share. Please find an alternate way to show this, or else your point is lost on the reader.

We have improved Fig. 2E in the revised manuscript and now present the front view and a section of the structure side by side to provide a clearer view of the more flexible regions. Please note that we have also adapted the coloring in the revised Fig. 2E so that it is consistent with the color scheme of the article. The frontal straight tetrameric helix bundle harboring the 2A-2B domains now appears in reddish colors (indicating the most rigid region of the protofibril), whereas the curved tetrameric helix bundle harboring the 1A-1B domains now appears in bluish colors (indicating increased flexibility in this region).

7. The authors go to significant lengths to explain the frontal straight tetrameric helix bundle compared to the curved tetrameric helix bundle (Figure 2, 3). However, the tetrameric composition of these species are poorly illustrated, and there is little clarity as to how the 1A-1B helices relate to the 2A-2B helices – are the straight and curved tetrameric species from the same 4 vimentin proteins? Or are they each formed from 4 different polypeptide strands, meaning 8 total vimentin proteins? Given how important this architecture is to the formation of the outer vs inner surface of the VIF, there needs to be additional images explaining how each of these species is formed and what constitutes each. Please create a figure that transitions the reader from Fig 2D to 3B/C – right now the leap is not entirely clear.

We thank the reviewer for this comment. First of all, in the revised manuscript we have improved Fig. 3. In the new Fig. 3D, we show the repeating unit segmentation from Fig. 2D overlaid with the model to provide a better transition for the reader between the electron density map and the model.

Secondly, we fully agree that the rather sparse illustration of the tetrameric composition of the 1A-1B and 2A-2B regions needed improvement. For better visualization, we added a new figure to the revised manuscript (Extended Data Figure 12). In this figure, we use map and model with the simplified color scheme to illustrate the tetrameric composition of the 1A-1B and 2A-2B regions. The figure exemplifies how the 2A-2B helices primarily shape the outer surface, while the 1A-1B helices shape the inner surface of the filament. Moreover, for a complete picture of the composition of the outer and inner surfaces of VIFs we added a new video to the revised manuscript (Supplementary Video 7).

In regards to your question on how the straight and curved tetrameric species are build, we realized how important it is to add cross-section views and to annotate the polarity of the helices in the protofibril model (see revised Fig. 4B and Extended Data Fig. 13). The 1A-1B region (the curved tetrameric species in the electron density map) is formed from the 1A-1B section of one tetramer. The 2A-2B region (the straight tetrameric species in the electron density map) is formed from the two 2A-2B sections of the two neighboring tetramers along the protofibril. The 1A-1B region is tetrameric (2 antiparallel dimers) per default, because it is formed from A_{11} tetramers. The 2A-2B region is also tetrameric, but the 2 antiparallel 2A-2B dimers come from the two neighboring A_{11} tetramers. We hope that this has clarified that the repeating unit of a protofibril (where it comprises 8 single peptide chains in cross-section) is actually an assembly of three tetramers.

8. Along the same lines as the previous comment, this manuscript would be greatly improved by having a figure that shows, within the VIF fibre, each of the following states: single vimentin polypeptide chain, vimentin heterodimer, vimentin tetramer, and vimentin octamer. A panel like Figure 3B all in gray except highlighting the path a single protein, then a dimer, a tetramer, and octamer would help synthesize decades of models of VIF assembly and more clearly help readers understand the filament formation and distinguish the single polypeptide path from the straight and curved tetrameric species discussed above. If the authors wanted to go the extra mile, a supplemental figure highlighting single A_{11} , A_{12} , and A_{22} tetramers within the fiber model could help unify decades of biophysics. There was no comment on CAN interaction – do the authors conclude this does not exist in VIFs?

We hope that the revised Fig. 4B clarifies how VIFs are constructed. The filament is an assembly of 5 protofibrils, which are held together by their head and tail domains. To answer the question of how protofibrils are constructed, it is important to note that only A_{11} tetramers (Fig. 4A) are needed as building blocks for a protofibril (revised Fig. 4B and revised Extended Data Fig. 13). To understand the mechanism, it is critical to see that a minimum of three tetramers are required to fully assemble a protofibril. Therefore, we would like to keep the assembly scheme rather simple, since it just needs an A_{11} tetramer and a fully assembled protofibril (as well as the interactions of the head and tail domain) to explain the structure of VIFs. Here it is very exciting to see how A_{12} and A_{22} interactions are constructed in the fully assembled protofibril, without the need for additional tetrameric species. The A_{22} interactions are created by the two antiparallel 2A-2B dimers in the 2A-2B regions of the protofibril and the A_{12} interactions are formed between antiparallel 1A-1B and 2A-2B dimers in the protofibrils. We also measured the overlap between the start of the 1A domain and the end of the 2B domain in the model (A_{CN} overlap). We added these points to the revised manuscript (see lines 220-228 and 241-246).

9. Figure 4B would be more effective if you had two panels side by side – one with the consistent color scheme throughout the paper and one where each tetramer was a solid single color, so the layering of tetramers would be more clear.

We thank the reviewer for this comment. To improve the visualization of the layering of the tetramers in the protofibril assembly, we added cross-sections and annotated the polarity of the helices in the

revised Fig. 4B and Extended Data Fig. 13. Additionally, we added a new video to the revised manuscript (Supplementary Video 8) that moves in cross-section along a protofibril, to give the full picture of tetramer layering.

10. Page 9, lines 210-211 (and lines 265-266): a multiple sequence alignment of type III IFs, a few select Type I/II, IV, and V IFs should be shown to demonstrate the conservation of the interlocking mechanism at the primary sequence level.

We thank the reviewer for this suggestion and added a multiple sequence alignment to the revised version of the manuscript (Extended Data Fig. 14).

11. For extended data Fig 9 please color heads and tails the appropriate “consistent” color you choose for the manuscript so that the polarity of the protofibril assembly (parallel vs antiparallel) is readily apparent (since there are no N- and C- terminal labels).

We thank the reviewer for this comment. In the revised version of the manuscript, we added the polarity annotations to clarify this point (Fig. 4B and Extended Data Fig. 13). Additionally, we added another model (Extended Data Fig. 13, right side) in which the central tetramer was removed. This further clarifies how the 2A-2B region in the protofibril is formed from two antiparallel 2A-2B dimers derived from the neighboring two tetramers.

12. Ext Data Fig 12 – the labels at the top (“no missing wedge”...) seem to be reversed based on the Fig legend description.

In this figure we show the effect of the missing wedge on a VIF oriented perpendicular to the direction of an electron beam. This is the orientation of most VIFs in cells cultured on a 2D support. The original figure caption was unnecessarily complicated (“without missing wedge deconvolution” means “missing wedge applied”). This has been rephrased in the revised manuscript (see lines 1187-1190).

13. Page 11, lines 265-271. One of the most important findings of the manuscript is the molecular interlock region, yet other than the mention of Y117L, there is no other mutational information discussed. This is a major oversight – the next paragraph between line 271 and 272 should discuss the human diseases caused by mutation specifically in the molecular interlock region, and a supplementary table added listing the known mutations located in the interlock region. This information will provide more substantial clinical impact to this work.

We thank the reviewer for this comment. We have added text and citations that summarize some of the known disease-associated point mutations to emphasize the importance of the interlock region and the connection between mutations in this region and diseases (see lines 346-351 in the revised manuscript).

14. Lines 27, 68: “form amyloid-like fiber” – this term is used, but it is not really shown in any images that the head regions are truly amyloid like. At current resolution beta-sheets can’t be resolved. Consider “proposed to be amyloid-like structures” or acknowledge the limitation. If you have other evidence it is “amyloid fiber” like, then please present it.

We apologize for not being clear about this matter. We have clarified this in the revised version by stating that the assignment of the head domain as amyloid-like, is based on the work of McKnight and colleagues (please see also our response to Reviewer #1). The low resolution of the head domain does

not allow us to unambiguously determine the structure of the luminal fiber (discussed in the revised manuscript, lines 304-321).

15. Line 71: "Mouse embryonic fibroblasts (MEFs) are widely used..." Do they (MEFs) have other Ifs expressed, if yes, how authors are sure that they are looking at VIMs? If only VIM expressed, I suggest to mention it.

In our lamin paper (Turgay et al., 2017), we used vimentin knockout MEFs and could not detect any 10 nm filaments in the cytoplasm. Thus, the only cytoskeletal IFs that could be detected by cryo-ET in MEFs are VIFs.

16. Line 168: "...on the alphafold prediction of the full-length vimentin dimer (Extended Data Fig. 7) "On this figure the Alpha-fold model should be originally colored from the program and the confidence bar (colors) should be presented.

We have revised the figure accordingly and mapped the pLDTT score values on the alphafold full-length vimentin dimer model (see revised Extended Data Fig. 10, left side).

17. Line 277: "We suggest that a unit-length VIF is assembled from 5 tetramers and that this assembly is driven by a phase-separation process which facilitates the initial association of the tetramers via their head domains. Therefore, the head domains may act as initial nucleators of VIF filament assembly". When two tetramers come together, is concentration of the head domains high enough for phase separation? Can the authors be more specific on phase-separation and how it results in "aggregating to the luminal fiber" (line 275). If this is true, can authors also have some model for VIM disassembly and dynamics, which require head domain phosphorylation? It looks like phase separated head domains are inside the filament and can't be accessed by kinases?

We apologize for the lack of clarity in our wording. It was misleading to use the term phase separation here. In the discussion of the revised manuscript (see lines 311-321), we clarify the idea that the head domains could self-assemble into an amyloid-like fiber driven by cross- β interactions. Thereby, the tetramers could be oriented in a position that facilitates the formation of protofibrils around the fiber in the center. In response to Reviewer #1 we clarified the suggestion that the luminal fiber could be an amyloid-like fiber formed by the head domains, and therefore stabilized on a molecular level by cross- β interactions. This is based on the work by McKnight and coworkers (Lin et al., 2016). The basis of our conclusion is our finding that the central fiber is likely composed of the low complexity head domains of vimentin and the findings of the McKnight group, who showed that the head domains of vimentin (and other IFs as well) forms fibers that generate the typical X-ray diffraction pattern of cross- β interactions.

Regarding the point made by this reviewer that kinases which phosphorylate the head domains of vimentin must access the lumen of the filaments - we draw exactly the same conclusion. We explain how this process could work in our answer to the next question.

18. One curious question arising from the manuscript, that is not discussed, and follows the above remark on the head domains, is that there is a substantial amount of literature discussing the regulation of VIM assembly through phosphorylation of the head domains. It would seem that the current structural model with the head domains forming an interior fiber, would make phosphorylation regulation of VIFs difficult. How do the authors reconcile the body of literature regarding head domain phosphorylation and the model? Is all phosphorylation occurring pre-VIF? How could post-VIF phosphorylation work

in this model? If phosphorylation takes place after assembly, the heads should be accessible for PKA (cAMP- dependent kinase), correct?

We thank the reviewer for this exciting comment. Indeed, one of the implications of our model is that kinases targeting the head domains need to access the VIF lumen. Regarding this point, we believe that the tail domains regulate access of kinases to the luminal fiber. In our structure, the tail domains are evenly distributed between the protofibrils. They keep the protofibrils together and the lumen sealed in this configuration. In order to facilitate the phosphorylation of the head domains in the lumen, the VIFs need to increase access to the luminal fiber. We have added text that speculates on potential mechanisms to the discussion of the revised manuscript (see lines 322-339). In response to Reviewer #1, we also added an additional figure (Extended Data Fig. 18), which shows that reorganization of the tail domains could modulate the permeability of the filaments. In the extreme case where all tail domains are detached from the protofibrils, the filaments would be open, presenting the luminal fiber as a target for protein kinases.

Some literature to consider on phosphorylation:

Eriksson, J.E., et al., Specific in vivo phosphorylation sites determine the assembly dynamics of vimentin intermediate filaments. *J Cell Sci*, 2004. 117(Pt 6): p. 919-32.

...IFs are highly dynamic structures, the constituent proteins of which undergo active exchange between a major compartment of assembled IF polymers and a very small fraction of disassembled IF subunits. In order to achieve these dynamic properties, IF proteins require an active support mechanism that maintains the exchange of subunits between the assembled and disassembled protein fractions. Reversible phosphorylation is a mechanism that could plausibly maintain this kind of exchange. ...

...both the equilibrium between vimentin polymers and depolymerized subunits and the turnover of subunit exchange are regulated by kinase-phosphatase equilibria. This regulation is based upon reversible phosphorylation of a number of phosphorylation sites located mainly in the N-terminal region of vimentin. (HEAD DOMAIN)

... results show that PP1 and PP2A inhibition results in rapid elevation of vimentin phosphorylation, with subsequent disassembly of the vimentin IFs.

From our results, it appears that the elevated phosphorylation first induces a net fragmentation of the polymers and, when the phosphorylation is more markedly elevated, then induces a net release of soluble subunits.

Our results demonstrated that phosphorylation releases the same oligomeric peptide composition both in vitro and in vivo; tetrameric oligomers were released both by PKA phosphorylation in vitro and by inhibition of dephosphorylation in vivo. ...Our present results, showing that phosphorylation induces disassembly into tetrameric subunits, could imply that the phosphorylation- driven subunit exchange between IF polymers and soluble subunits would take place in the form of a tetramer.

Ser-4, Ser-6, Ser- 7, Ser-8, Ser-9, Ser-38, Ser-41, Ser-71, Ser-72, Ser-418, Ser- 429, Thr-456, and Ser-457 as significant in vivo phosphorylation sites.

...However, the absence of two major phosphorylation sites, Ser-38 and Ser-72, did not have a major effect on filament assembly, indicating that in regulating assembly equilibria, the role of phosphorylation is primarily to drive disassembly.

Kraxner, J., et al., Post-translational modifications soften vimentin intermediate filaments. *Nanoscale*, 2021. 13(1): p. 380-387.

... We find that the filaments soften with increasing amount of phosphorylated protein within the filament and that interaction with the protein 14-3-3 further enhances this effect.

phosphorylated sites are dispersed throughout the whole protein, but the most abundant ones ($\log_2(I) > 0$, red lines) are all found in the head region of vimentin. ...Strikingly, the positions S71 and S72 are always phosphorylated simultaneously and show the highest degree of phosphorylation.

When vimentin becomes phosphorylated, the positive charges of the head domain are flanked by negative charges of the phosphorylated amino acids as shown in Fig. 4a, which diminishes the electrostatic attraction between the head and the coiled-coils.

Head domain of vimentin is positively charged (Kraxner, J. and S. Koster, Influence of phosphorylation on intermediate filaments. *Biol Chem*, 2023) is the surface charge of the interior wall of the filament negative?

...Phosphorylation at specific sites of vimentin leads to disassembly, which is a necessary prerequisite for cell division during mitosis (Eriksson et al. 1992; Inagaki et al. 1996).

in both vimentin and desmin the preferred phosphorylation sites lie in the head domain (Geisler et al. 1989; Geisler and Weber 1988) that is also crucial for filament assembly as shown in experiments with headless vimentin, which is unable to assemble (Rogers et al. 1995).

Early studies revealed that phosphorylation of vimentin at Ser55 by p34cdc2 kinase is responsible for the disassembly during mitosis (Chou et al. 1990), and later studies identified additional interphase-specific phosphorylation sites (Sihag et al. 2007).

phosphorylation of IF proteins does not solely provide a mechanism for reversible assembly of filaments but rather offers a huge variety of regulation mechanisms ranging from remodeling of IF networks (Chou et al. 1989; Eriksson et al. 2004), cell migration (Chung et al. 2013), changes in mechanical properties (Kraxner et al. 2021), interactions with other protein structures (Sihag et al. 2007) or roles in signaling pathways (van Engeland et al. 2019).

We thank the reviewer for this comprehensive entry point on literature about phosphorylation of VIFs. Several of the above papers are cited in the revised manuscript (see lines 322-339).

Minor:

19. Figure 1 C – how was this image created? Please clarify in figure legend or in methods.

The description was added to the methods section (see lines 939-942 in the revised manuscript).

20. Figure 3E is my favorite image in the whole manuscript. With a little tweaking of the colors so that the lateral tail contacts pop out more, this could be worthy of a journal cover image.

We thank the reviewer for the suggestion and will offer such a figure to the journal.

Reviewer #3:

The results presented by Eibauer et al. are of interest to people in the intermediate filament/cytoskeleton, cryoEM helical reconstruction and cancer research fields. If all above remarks will be addressed, the manuscript can be considered for publication. The text is very readable and concise enough.

We thank the reviewer for appreciating the importance of structural characterization of IFs and the importance of this work to our field.

However, aside from the very detailed modelling part, that was extremely interesting and convincing, the quality of the data needs to improve substantially, and results should be more convincing.

The manuscript presents results of the yet elusive full 3D structure of assembled vimentin intermediate filaments (VIFs). From cross sections of an in situ tomogram, the authors claim to VIFs composed of 5 protofibrils, stoichiometry that is in contrast to the previously reported 4-protofibril VIFs structure seen previously in cryoEM by Goldie et al., 2007. The claim is not convincing, though, and that is the main flaw: both power spectra are confusing and layer lines hard to discern. In the first one (Fig. 1F), very pixelated probably because of the noise in the rest of the tomogram, lines 3 and 4 are not lying on defined layer lines, as for line 1 and 2; in the supplementary power spectrum (Extended Data Fig. 4C), the cyan arrow at helical rise points to no clear line, while the first visible meridian layer line is localized lower than the cyan arrow.

Moreover, the use of a software (IsoNet) to overcome the missing wedge limitation in the cryo-FIB milled tomogram (Movie 1) seem to enhance missing-wedge artefacts instead of attenuating them (striped lines at the left edge of the tomogram Movie 1, middle frames, and in the same region in Extended Data Fig. 1A).

Thus, the analysis of features from missing-wedge corrected tomogram might not be reliable. For this reason, lines 236-237: “Without correction, the difference between 4 and 5 protofibrils could not be resolved” and Extended Data Figure 12: “[...] cryo-ET without missing wedge deconvolution can resolve only 4 protofibrils in cross-section”, are dangerous statements in light that the tomogram with artificially filled missing wedge displays more artefacts. Also, the overall quality of the cryoEM tomogram and the fibers within is lower compared to the cryoEM tomogram and VIFs in detergent-treated cells, that has no reported missing wedge “correction”: VIFs there look much more detailed and organic. It would be good to show the original unprocessed tomogram, or more, and what stoichiometry is extrapolated from it. Native mass-spec can possibly help redeem the 4 or 5 protofilament stoichiometry conundrum of VIFs, despite them being reported to be structurally very polymorphic.

We thank the reviewer very much for these constructive criticisms. In order to base the conclusion of the native protofibril stoichiometry on a solid statistical foundation we improved and extended the analysis of the cryo-FIB data considerably. For the revised manuscript we conducted 3 different and independent data processing workflows to show the native protofibril stoichiometry of VIFs. All three of these data processing methods clearly show the 5 protofibril architecture of VIFs. The new results are shown in the revised manuscript in Fig. 1D, Fig. 1E, Extended Data Fig. 1, Extended Data Fig. 2, and Extended Data Fig. 3. The results section (see lines 84-88), the discussion (see lines 276-290) and the methods section (see lines 547-646) were altered accordingly.

More specifically, we first performed subtomogram averaging of VIF segments from tomograms reconstructed with weighted backprojection (WBP). Initially, we extracted 14,273 subtomograms from 7 of the cryo-FIB tomograms. The final average was calculated from 2371 VIF segments. Without applying any assumptions on the symmetry, the average converged to a resolution of ~ 20 Å and clearly shows the 5 protofibril architecture of VIFs (see new Extended Data Fig. 1). Based on the unsymmetrized average we derived an initial hypothesis for VIF symmetry and applied it during averaging (see Fig. 1D in the revised manuscript).

Secondly, we improved the analysis of VIF cross-sections from the IsoNet corrected tomograms. In the previous manuscript we only looked at a small gallery of IsoNet corrected cross-sections extracted from one tomogram. For the revised version of the paper, we developed a method to average a statistically relevant number of VIF cross-sections, based on the analysis of IsoNet corrected tomograms. The cross-section average we present in the revised manuscript (see new Extended Data Fig. 2D) was calculated from 444 cross-sections extracted from 7 tomograms of the cryo-FIB data. The average converges to a 5 protofibril architecture and resembles the average obtained from sub-tomogram averaging of the WBP reconstructed tomograms (see new Extended Data Fig. 2E).

In this context, the reviewer comment about the enhancement of missing wedge artefacts in the IsoNet corrected tomograms is very important. However, we conclude that these artefacts are not introduced by the IsoNet correction, since they are visible in both the WBP reconstructed and IsoNet corrected tomograms (see Extended Data Figs. 1B and 2B, red dashed rectangles). If platinum remnants are present in the tilt series, IMOD is used for tomogram reconstruction. The platinum particles are used as fiducial markers for precise tilt series alignment. Platinum particles are highly concentrated at the front edges of the lamellae (see Extended Data Figs. 1B and 2B, blue dashed rectangles). These high contrast particles introduce a high density of distorting back-projection rays in the tomographic volume above and below the front edge of the lamellae (see Extended Data Figs. 1B and 2B, red dashed rectangles). This caused the artefacts in the tomogram movie referred to by this reviewer. These regions have been excluded from downstream data analysis (see revised methods section between lines 547-558).

For a direct comparison between the WBP data (which is used for the subtomogram average, see above) and the IsoNet corrected data (which is used for the cross-section average, see above) we have added Supplementary Video 1 and Supplementary Video 2 to the revised manuscript, showing the analyzed region in the respective tomogram (Extended Data Figs. 1B and 2B, yellow dashed rectangles).

As a third and additional independent confirmation of the 5 protofibril architecture, we extracted cross-section images from VIFs running along the z-axis (parallel to the electron beam). These cross-sections are located in the xy-planes of the tomograms and not affected by the missing wedge. We show these images in the revised manuscript in Fig. 1E and Extended Data Fig. 3.

Regarding the reviewer comment about the power spectrum analysis, we would like to emphasize that the question whether VIFs are built in a helical symmetric manner or built only by lateral attachment of the tetramers has been extensively discussed in the IF field. Therefore, the power spectrum shown in Fig. 1G is important, as it is a direct proof that the underlying symmetry of VIFs is helical. In the following of the paper, we used the data from the detergent-treated MEFs to improve the resolution of the power spectrum by calculating it from very long stretches of computationally assembled VIFs.

We thank the reviewer for pointing out the poor visibility of the meridional reflection at $1/42.5$ Å (Extended Data Fig. 6C, blue arrowhead). Indeed, this is a contrast/zoom problem. To better visualize the meridional layer line (with was identified by the meridional layer line correlation search shown in Extended Data Fig. 6D), we added two new images in the revised version of the figure (Extended Data Figs. 6, F and G). In Extended Data Fig. 6F, we show a zoom in the region around the meridional reflection using the same contrast settings as in Extended Data Fig. 6C. In Extended Data Fig. 6G, we show a zoom plus increased contrast of the meridional layer line. We think that with the additional images, the meridional reflection is better visualized.

It is important to see that the helical parameter set we found by this analysis (42.5 Å helical rise and 73.7° helical twist based on the detergent-treated MEF data) can be independently verified using the single particle data obtained from in-vitro polymerized VIFs. Here we derive the helical symmetry again with a different method. In this case, we use helical symmetry searches based on 3D classifications of the single particle data. We found the fact that we identified the same helical parameters from these 2 independent data sets and processing pipelines striking. To better emphasize this independent verification of the helical symmetry measurement, we added Extended Data Figs. 7, B and C, to the revised manuscript.

Since the length of VIFs within cells and in-vitro polymerization reactions are not constant and cannot be controlled (each filament has a different length), results from native mass-spectrometry would be extremely challenging to interpret. Therefore, we performed the experiments explained above to confirm the 5 protofibril architecture of VIFs.

Another point that should be clarified in the study is what are the contacts that keep the structure intact in the tailless maps and models (Extended Data Fig. 6)? Only the low complexity head domains that aggregate in the central amyloid-like fibre of the filament? What is then the function of tail domains in VIMs if VIM- Δ T is still able to form filaments?

Yes, we think this is the case. The protofibrils in the Δ T structure are held together by the head domains, which are forming the luminal fiber in the center of the filaments. The tail domains laterally connect the protofibrils and thereby control their spacing and the diameter of the filaments (shown in (Herrmann et al., 1996)). Therefore, we suggest that the tail domains control the access to the filament lumen, which is critical in the context of phosphorylation of VIFs. We speculate that if the tail domains detach from the protofibrils, the protofibrils would become more separated, and the filament is likely to exhibit a more opened structure, facilitating the access of kinases to the luminal fiber. We discuss these ideas (see also the comments of Reviewer #1 and Reviewer #2) in the revised version of the manuscript (see lines 328-339 in the discussion and Extended Data Fig. 18).

The final point is regarding the discussion lines 278-280, where authors declare that “VIF [...] assembly is driven by a phase-separation process which facilitates the initial association of the tetramers via their head domains.”. This is an unannounced speculation that was never introduced nor discussed previously in the manuscript as an investigation hypothesis and has no supporting data from the static three dimensional structure and model reported. And no references are provided to previous publications where VIFs nucleation and assembly via phase-separation was observed. Therefore, it should either be proven or removed, or at least greatly toned down.

We thank the reviewer for this comment. It was misleading to use the term phase separation here. We corrected this in the revised manuscript (see lines 358-361). Moreover, in the discussion of the revised manuscript (see lines 311-321), we clarify the idea that the head domains could self-assemble into an amyloid-like fiber in the center of VIFs driven by cross- β interactions (see also the comments of Reviewer #1).

other points:

- Structure-based predictions should be provided along with confidence scores

We have revised the figure accordingly and mapped the pLDTT score values on the alphafold full-length vimentin dimer model (see Extended Data Fig. 10, left side).

- A few more tomograms could be analyzed/shown and/or number of experiments should be indicated.

The number of tomograms, subtomograms, and cross-sections, that were analyzed for the determination of the native protofibril stoichiometry from the cryo-FIB data in the revised manuscript are integrated in the methods section (see lines 547-646) and listed in Extended Data Table 1.

References

- Herrmann, H., Haner, M., Brettel, M., Muller, S. A., Goldie, K. N., Fedtke, B., . . . Aebi, U. (1996). Structure and assembly properties of the intermediate filament protein vimentin: the role of its head, rod and tail domains. *J Mol Biol*, *264*(5), 933-953. doi:10.1006/jmbi.1996.0688
- Lin, Y., Mori, E., Kato, M., Xiang, S., Wu, L., Kwon, I., & McKnight, S. L. (2016). Toxic PR Poly-Dipeptides Encoded by the C9orf72 Repeat Expansion Target LC Domain Polymers. *Cell*, *167*(3), 789-802 e712. doi:10.1016/j.cell.2016.10.003
- Turgay, Y., Eibauer, M., Goldman, A. E., Shimi, T., Khayat, M., Ben-Harush, K., . . . Medalia, O. (2017). The molecular architecture of lamins in somatic cells. *Nature*, *543*(7644), 261-264. doi:10.1038/nature21382

Decision Letter, first revision:

Message: 10th Nov 2023

Dear Dr. Medalia,

Thank you again for submitting your manuscript "Vimentin filaments integrate low complexity domains in a complex helical structure". We now have comments (below) from the 3 reviewers who evaluated your paper. In light of those reports, we remain interested in your study and would like to see your response to the comments of the referees, in the form of a revised manuscript.

You will see that while reviewers appreciate the results, they raise several concerns which will need to be addressed in a revision. Specifically, we will ask you for one final effort to address comments pertaining to model building brought up by reviewer #2. In addition, we will ask that you address remaining comments of reviewer #3 and where appropriate and necessary soften the language around the filament stoichiometry where data does not unequivocally support the conclusions.

Please be sure to address/respond to all concerns of the referees in full in a point-by-point response and highlight all changes in the revised manuscript text file. If you have comments that are intended for editors only, please include those in a separate cover letter.

We expect to see your revised manuscript within 6 weeks. If you cannot send it within this time, please contact us to discuss an extension; we would still consider your revision, provided that no similar work has been accepted for publication at NSMB or published elsewhere.

Reporting Summary:

When submitting the revised version of your manuscript, please pay close attention to our [href="https://www.nature.com/nature-portfolio/editorial-policies/image-integrity"](https://www.nature.com/nature-portfolio/editorial-policies/image-integrity)>Digital

Image Integrity Guidelines. and to the following points below:

Please note that all key data shown in the main figures as cropped gels or blots must be presented in uncropped form, with molecular weight markers. These data can be aggregated into a single supplementary figure item. While these data can be displayed in a relatively informal style, they must refer back to the relevant figures.

SOURCE DATA: we urge authors to provide, in tabular form, the data underlying the graphical representations used in figures. This is to further increase transparency in data reporting, as detailed in this editorial (<http://www.nature.com/nsmb/journal/v22/n10/full/nsmb.3110.html>). Spreadsheets can be submitted in excel format. Only one (1) file per figure is permitted; thus, for multi-paneled figures, the source data for each panel should be clearly labeled in the Excel file; alternately the data can be provided as multiple, clearly labeled sheets in an Excel file. When submitting files, the title field should indicate which figure the source data pertains to.

Data availability: this journal strongly supports public availability of data. All data used in accepted papers should be available via a public data repository, or alternatively, as Supplementary Information. If data can only be shared on request, please explain why in your Data Availability Statement, and also in the correspondence with your editor. Please note that for some data types, deposition in a public repository is mandatory - more information on our data deposition policies and available repositories can be found below: <https://www.nature.com/nature-research/editorial-policies/reporting-standards#availability-of-data>

Nature Structural & Molecular Biology is committed to improving transparency in authorship. As part of our efforts in this direction, we are now requesting that all authors identified as 'corresponding author' on published papers create and link their Open Researcher and Contributor Identifier (ORCID) with their account on the Manuscript Tracking System (MTS), prior to acceptance. This applies to primary research papers only. ORCID helps the scientific community achieve unambiguous attribution of all scholarly contributions. You can create and link your ORCID from the home page of the MTS by clicking on 'Modify my Springer Nature account'. For more information please visit please visit www.springernature.com/orcid.

<https://mts-nsmb.nature.com/cgi-bin/main.plex?el=A2J6DKO7A6oal2J3A9ftd24BmXvWYMb8qryijTGskpwZ>

Sincerely,

Katarzyna Ciazynska, PhD
(she/her)
Associate Editor
Nature Structural & Molecular Biology
<https://orcid.org/0000-0002-9899-2428>

Reviewers' Comments:

Reviewer #1:
None

Reviewer #2:
Remarks to the Author:

The authors have put significant effort into addressing multiple weaknesses of the first draft of this manuscript describing the cryo-EM/ET structure of vimentin filaments. I commend the authors for wholeheartedly addressing them.

I have one major point left to address:

1. The pdb files associated with this publication are not appropriate for the resolution of the structure. Viewing of the tetramer and higher order filament in Chimera reveals full side chains are visible. In fact, the filament model shows a lysine side chain penetrating a

phenylalanine ring- this occurs in multiple locations. All pdb models of a 7.2A structure should have the side chains truncated at C-alpha. Otherwise, other investigators may use these models to make conclusions about residue to residue interactions erroneously.

I have a few minor points left to address:

1. Page 9, lines 221-228. Excellent additions to clarify the A11 tetramer building block vs the A12, A22, and ACN interactions. This concept was also proposed in Eldirany et al, EMBO J. 2019 Jun 3;38(11):e100741, which should be cited along with ref 63.

2. The phosphorylation issue, while adequately responded to, still will remain controversial in many eyes. I think it would be helpful to clarify the diameter of the VIF lumen compared to the diameter of the kinases you expect to have access to the heads in the lumen.

Reviewer #3:

Remarks to the Author:

I am very grateful to see that the authors have provided a comparison between the unfiltered and the IsoNet-processed tomogram. I appreciated them cropping the tomogram to minimize the flare artifacts coming from the platinum deposition on the lamellae, that were indeed enhanced in by the IsoNet correction, allowing the viewer to focus on the VIFs and other biological features in the volume.

The results of the comparison between weighted backprojection and IsoNet-processed tomogram are more convincing towards the 5-fold symmetry of the protofilaments in assembled VIFs. In particular, I appreciated the comparison xy-cross-sections of VIFs shown in extended Data Figure 3A and B, from where it is possible to evaluate the extent of the applied missing wedge correction on the left, as well as the cross-section views in Extended Data Figure 1C and D and 2E showing a pentameric helical symmetry for the VIFs in both subtomogram averages. I would like the authors to clarify from which tomogram (denoised or not) have been extracted the cross-sections of pentameric VIFs in Figure 1E and Extended Data Figure 3C, and that should be included in the captions text. Since a lot of work has been dedicated to dispel all doubts on the definitive VIFs stoichiometry, if the resolution/level of noise allows it, it would be nice to have a gallery of xy-cross sections of VIFs as seen in the non-denoised weighted backprojection reconstructed tomogram, which is naturally affected by resolution anisotropy due to the missing wedge, and/or VIF cross-section average calculated without any assumptions on the symmetry of the assembly. In this way, there should be no need to model/simulate the missing wedge effect on VIF cross-sections to prove that only 4 protofilaments instead of 5 can be resolved from the non-denoised tomograms (Extended Data Figure 17). Regarding the power spectrum analysis, I agree it is an important figure to prove the helical nature of the fibrils, that was never questioned in first instance, but hard to infer from the resolution. Anyways, I thank the authors for the effort of improving contrast and visibility of the meridional layer line in Extended Data Figure 6F and G.

Lines 333-339 and Extended Data Figure 18: the authors speculate that "the tail domains reorganize" or "if the tail domains were completely detached from the protofibrils"...have they inferred any of this arrangement in the structure of VIFs, or is there any hint of them repositioning in either conformation somehow? Could the tails be cleaved by a protease to grant access to head-phosphorylating kinases?

Also, in Extended Data Figure 18 why only one tail out of 2 gets rearranged in the model

(only the yellow tail gets repositioned)? Are there two different types of tails? Or is it to fit the 47-nm gap consistent with the tail-to-tail distance measured previously? I am not sure this model makes complete sense in its current form.

Figure 1B: the microtubule that is labelled with an orange arrow on the bottom left appears on the top left in the Supplementary Videos 1 and 2. Either Figure 1B needs to be flipped upside down to match the volumes, or the other way round, as preferred. Figure 1B should be referred to in lines 1220-1223 and 1225-1227 as a central frame of the IsoNet reconstructed tomogram.

Figure 1E and Extended Data Figure 3C: please clarify in the text that the xy-cross sections of pentameric VIFs are extracted from the denoised tomogram.

Extended Data Figure 10: there is no need to show the different domains in shades of colors (blue fading to green and so on): the residue numbers help identifying the different secondary structures already. This is especially confusing when presented next to the pLDDT plot that shows the same colors associated to AlphaFold predictions confidence. I suggest the authors to show the isolated secondary structures using the same colors of their relative pLDDT confidence and show them sequentially from left to right following the residue numbers, to avoid the reader going back and forth between them.

Extended Data Figure 13: "plus" and "minus" texts can be confused for the polarity + and - of the fibres. Therefore, I would suggest keeping the text below, referring to the tetramer number, keep only the arrows above and for the t1+t3 add a text like "t2 not shown" or "hidden".

Extended Data Figure 14: please cite all the homologous structures.

Extended Data Figure 18: needs a scalebar to put the 47 nm gap into context.

Author Rebuttal, first revision:

Reviewer #1:

None

We thank the reviewer for the time and effort to improve our manuscript.

Reviewer #2:

Remarks to the Author:

The authors have put significant effort into addressing multiple weaknesses of the first draft of this manuscript describing the cryo-EM/ET structure of vimentin filaments. I commend the authors for wholeheartedly addressing them.

We thank the reviewer for the time and effort to improve our manuscript, and for appreciating our commitment to follow the reviewers' suggestions.

I have one major point left to address:

1. The pdb files associated with this publication are not appropriate for the resolution of the structure. Viewing of the tetramer and higher order filament in Chimera reveals full side chains are visible. In fact, the filament model shows a lysine side chain penetrating a phenylalanine ring- this occurs in multiple locations. All pdb models of a 7.2A structure should have the side chains truncated at C-alpha. Otherwise, other investigators may use these models to make conclusions about residue to residue interactions erroneously.

We thank the reviewer for this comment. We have further improved the vimentin tetramer model and updated the respective PDBDEV entry (PDBDEV-00000212). As shown in the updated validation report, the clashscore of the tetramer model is now down to 2.67, and therefore of high quality (within the 98th percentile according to molprobity analysis). Since the vimentin filament model is assembled by the application of helical symmetry to the tetramer model, the clashscore of the filament model is consequently also improved, although it is clear that the clashscore of the filament model is higher as additional interactions between side chains are realized in the filament.

However, we are reluctant to truncate the side chains at C-alpha in the uploaded model because of multiple reasons. Since the tetramer model was obtained by molecular dynamics flexible fitting (while embedded in the filament model), the orientations of the side chains are guided by physical principles. The electron density map acts as an external field that directs the model into the map (Trabuco, Villa, Mitra, Frank, & Schulten, 2008). Numerous articles show that models of this type are deposited in the PDB with non-truncated side chains (for example PDB ID code 5FUR as just one illustrative example). Truncating the side chains would imply that our model could not be used further by other scientists in applications where the side chains are essential. These applications include using the model as input for molecular dynamics simulations, estimating the charge and hydrophobicity of different surface regions of the filament, and conducting volume estimations.

I have a few minor points left to address:

1. Page 9, lines 221-228. Excellent additions to clarify the A11 tetramer building block vs the A12, A22, and ACN interactions. This concept was also proposed in Eldirany et al, EMBO J. 2019 Jun 3;38(11):e100741, which should be cited along with ref 63.

The reference was added as suggested (see line 220, citation 66 in the revised manuscript).

2. The phosphorylation issue, while adequately responded to, still will remain controversial in many eyes. I think it would be helpful to clarify the diameter of the VIF lumen compared to the diameter of the kinases you expect to have access to the heads in the lumen.

[Redacted text block]

[Redacted text block]

██
██
██
██

Reviewer #3:

Remarks to the Author:

I am very grateful to see that the authors have provided a comparison between the unfiltered and the IsoNet-processed tomogram. I appreciated them cropping the tomogram to minimize the flare artifacts coming from the platinum deposition on the lamellae, that were indeed enhanced in by the IsoNet correction, allowing the viewer to focus on the VIFs and other biological features in the volume.

The results of the comparison between weighted backprojection and IsoNet-processed tomogram are more convincing towards the 5-fold symmetry of the protofilaments in assembled VIFs. In particular, I appreciated the comparison xy-cross-sections of VIFs shown in extended Data Figure 3A and B, from where it is possible to evaluate the extent of the applied missing wedge correction on the left, as well as the cross-section views in Extended Data Figure 1C and D and 2E showing a pentameric helical symmetry for the VIFs in both subtomogram averages. I would like the authors to clarify from which tomogram (denoised or not) have been extracted the cross-sections of pentameric VIFs in Figure 1E and Extended Data Figure 3C, and that should be included in the captions text.

Since a lot of work has been dedicated to dispel all doubts on the definitive VIFs stoichiometry, if the resolution/level of noise allows it, it would be nice to have a gallery of xy-cross sections of VIFs as seen in the non-denoised weighted backprojection reconstructed tomogram, which is naturally affected by resolution anisotropy due to the missing wedge, and/or VIF cross-section average calculated without any assumptions on the symmetry of the assembly. In this way, there should be no need to model/simulate the missing wedge effect on VIF cross-sections to prove that only 4 protofilaments instead of 5 can be resolved from the non-denoised tomograms (Extended Data Figure 17).

Regarding the power spectrum analysis, I agree it is an important figure to prove the helical nature of the fibrils, that was never questioned in first instance, but hard to infer from the resolution. Anyways, I thank the authors for the effort of improving contrast and visibility of the meridional layer line in Extended Data Figure 6F and G.

We thank the reviewer for the time and effort to improve our manuscript, and for appreciating our commitment to follow the reviewers' suggestions.

The subtomogram (Extended Data Fig. 1C) and cross-section averages (Extended Data Fig. 2D) were both calculated without any assumptions on the symmetry of the assembly. The VIF cross-sections shown in Fig. 1E and Extended Data Fig. 3C are located in the xy-plane of the tomogram, and therefore are isotropic in resolution and not affected by the missing wedge. We only applied denoising to these images to improve the contrast. As requested, we added this information to the figure captions (see lines 391 and 1000 in the revised manuscript).

We agree with the reviewer that the missing wedge simulation (previous Extended Data Fig. 17) is not needed anymore and removed this figure from the revised manuscript.

Lines 333-339 and Extended Data Figure 18: the authors speculate that “the tail domains reorganize” or “if the tail domains were completely detached from the protofibrils”...have they inferred any of this arrangement in the structure of VIFs, or is there any hint of them repositioning in either conformation somehow? Could the tails be cleaved by a protease to grant access to head-phosphorylating kinases?

The quoted text was added to the manuscript as a speculative hypothesis, requested by reviewers #1 and #2. The precise structural effect of phosphorylation of the tail domain is still not well understood and beyond the scope of this manuscript. The tail possibly changes its conformation and orientation due to phosphorylation, however, no cleavage of the tail has been reported.

Also, in Extended Data Figure 18 why only one tail out of 2 gets rearranged in the model (only the yellow tail gets repositioned)? Are there two different types of tails? Or is it to fit the 47-nm gap consistent with the tail-to-tail distance measured previously? I am not sure this model makes complete sense in its current form.

We thank the reviewer for this comment. This figure was added due to a comment from reviewer #1. The figure (Extended Data Fig. 17 in the revised manuscript) illustrates the idea that the tail domains could reorganize and cluster along the filament, thereby creating a distance of ~47 nm between the clustered tail domains. There are numerous possibilities as to how the tail domains could reposition. We illustrate one possibility of this hypothetical tail domain clustering, where the yellow tail domains changed their position. If we assume that the tails are identical, repositioning of the magenta tails would be equal likely. However, it could be that the tails are arranged in an alternating, layered pattern, so that the phosphorylation of one group of tails is more likely because their phosphorylation site is exposed.

Figure 1B: the microtubule that is labelled with an orange arrow on the bottom left appears on the top left in the Supplementary Videos 1 and 2. Either Figure 1B needs to be flipped upside down to match the volumes, or the other way round, as preferred. Figure 1B should be referred to in lines 1220-1223 and 1225-1227 as a central frame of the IsoNet reconstructed tomogram.

We thank the reviewer for this comment. We have corrected this error by flipping the Supplementary Videos 1 and 2 in the revised manuscript and added the link to Figure 1B (see line 1220 in the revised manuscript).

Figure 1E and Extended Data Figure 3C: please clarify in the text that the xy-cross sections of pentameric VIFs are extracted from the denoised tomogram.

The text was added to the figure legends (see lines 391 and 1000 in the revised manuscript).

Extended Data Figure 10: there is no need to show the different domains in shades of colors (blue fading to green and so on): the residue numbers help identifying the different secondary structures already. This is especially confusing when presented next to the pLDDT plot that shows the same colors associated to AlphaFold predictions confidence. I suggest the authors to show the isolated secondary structures using the same colors of their relative pLDDT confidence and show them sequentially from left to right following the residue numbers, to avoid the reader going back and forth between them.

We thank the reviewer for this comment. As suggested, we ordered the isolated secondary structures from left to right according to their residue numbers in the revised Extended Data Figure 10. However, we hope you understand our reservations about changing their coloring. We believe it is helpful for the reader to better identify these structures in Figure 3 and Extended Data Figure 11, if the color scheme is consistent. To better visually separate the isolated secondary structures from the full dimer on the left, colored with the pLDDT confidence score, we added a dashed line in between.

Extended Data Figure 13: “plus” and “minus” texts can be confused for the polarity + and – of the fibres. Therefore, I would suggest keeping the text below, referring to the tetramer number, keep only the arrows above and for the t1+t3 add a text like “t2 not shown” or “hidden”.

As suggested, we removed all “plus” and “minus” texts from the revised Extended Data Figure 13 to avoid confusion with the polarity identifiers.

Extended Data Figure 14: please cite all the homologous structures.

We added the UniProt identifiers of the analyzed intermediate filament sequences to the revised figure caption of Extended Data Figure 14.

Extended Data Figure 18: needs a scalebar to put the 47 nm gap into context.

We placed the scale bar in between the models and increased its thickness to improve visibility (Extended Data Figure 17 in the revised manuscript).

References

- Eriksson, J. E., He, T., Trejo-Skalli, A. V., Harmala-Brasken, A. S., Hellman, J., Chou, Y. H., & Goldman, R. D. (2004). Specific in vivo phosphorylation sites determine the assembly dynamics of vimentin intermediate filaments. *J Cell Sci*, *117*(Pt 6), 919-932. doi:10.1242/jcs.00906
- Geisler, N., Hatzfeld, M., & Weber, K. (1989). Phosphorylation in vitro of vimentin by protein kinases A and C is restricted to the head domain. Identification of the phosphoserine sites and their influence on filament formation. *Eur J Biochem*, *183*(2), 441-447. doi:10.1111/j.1432-1033.1989.tb14947.x
- Trabuco, L. G., Villa, E., Mitra, K., Frank, J., & Schulten, K. (2008). Flexible fitting of atomic structures into electron microscopy maps using molecular dynamics. *Structure*, *16*(5), 673-683. doi:10.1016/j.str.2008.03.005
- Zheng, J., Trafny, E. A., Knighton, D. R., Xuong, N. H., Taylor, S. S., Ten Eyck, L. F., & Sowadski, J. M. (1993). 2.2 Å refined crystal structure of the catalytic subunit of cAMP-dependent protein kinase complexed with MnATP and a peptide inhibitor. *Acta Crystallogr D Biol Crystallogr*, *49*(Pt 3), 362-365. doi:10.1107/S0907444993000423

Decision Letter, second revision:

Message: Our ref: NSMB-A47655B

18th Dec 2023

Dear Dr. Medalia,

Thank you for submitting your revised manuscript "Vimentin filaments integrate low complexity domains in a complex helical structure" (NSMB-A47655B). It has now been seen by the original referees and their comments are below. The reviewers find that the paper has improved in revision, and therefore we'll be happy in principle to publish it in Nature Structural & Molecular Biology, pending minor revisions to satisfy the referees' final requests and to comply with our editorial and formatting guidelines. This especially pertains to the guidelines for structural data deposition.

To facilitate our work at this stage, it is important that we have a copy of the main text as a word file. If you could please send along a word version of this file as soon as possible, we would greatly appreciate it; please make sure to copy the NSMB account (cc'ed above).

Sincerely,

Katarzyna Ciazynska, PhD
(she/her)
Associate Editor
Nature Structural & Molecular Biology
<https://orcid.org/0000-0002-9899-2428>

Reviewer #2 (Remarks to the Author):

The authors have addressed the significant side chain clashes in their vimentin tetramer and filament models, reducing the Molprobit clash scores. The degree of prior clashes was not suitable for deposition, but it is reasonable to proceed with deposition of the current structure so long as the statement(s) in the manuscript about the resolution limitations of the structure remain. It is essential if the model contains all side chains that the readers understand that accurate visualization of side chain atoms was not and could not be done at 7.2Å resolution.

The authors' state "Truncating the side chains would imply that our model could not be

used further by other scientists in applications where the side chains are essential." It is precisely this scenario where I believe caution is needed. Some investigators will use the model as fact and erroneous conclusions might occur when relying on side chain positions, contact residues and distances, surfaces, etc. In the end, though, the rigor of study falls on those investigators and their own reviewers.

I congratulate the authors on a most outstanding structural study.

Christopher G. Bunick, MD PhD
Associate Professor of Dermatology
Program in Translational Biomedicine
Yale University

Reviewer #3 (Remarks to the Author):

I was asked by the editor to comment on the PDB model and this is my response to the comments from Reviewer 2:

Reviewer #2: The pdb files associated with this publication are not appropriate for the resolution of the structure. Viewing of the tetramer and higher order filament in Chimera reveals full side chains are visible. In fact, the filament model shows a lysine side chain penetrating a phenylalanine ring- this occurs in multiple locations. All pdb models of a 7.2Å structure should have the side chains truncated at C-alpha. Otherwise, other investigators may use these models to make conclusions about residue to residue interactions erroneously.

Authors: Since the tetramer model was obtained by molecular dynamics flexible fitting (while embedded in the filament model), the orientations of the side chains are guided by physical principles. The electron density map acts as an external field that directs the model into the map (Trabuco, Villa, Mitra, Frank, & Schulten, 2008). Numerous articles show that models of this type are deposited in the PDB with non-truncated side chains (for example PDB ID code 5FUR as just one illustrative example).

My response: I agree with reviewer #2: models of a 7.2Å resolution (and even below, e.g. PDB ID: 8B5R at 6.10Å) should have the side chains truncated at C-alpha when deposited in PDB. I understand the authors effort of improving the clash score using molecular dynamics (MD), thus simulating the relatively correct positions and interactions of atoms by calculating forces between particles and their potential energies inside the map. The backbone of the ordered domains of vimentin filaments fit correctly in the density and the atomic model is probably also "acceptable" in terms of geometry, as assessed by the low clash score. However, the authors cannot claim that the side chains fit and the registry is correct in the context of this map, due to the limited resolution. In PDB, it is not allowed to deposit MD-only based models, without map. Therefore, I believe that depositing an MD model simulated in a low-resolution map, and therefore prone to bias and overfitting, is not a good scientific practice. The main concern here is that erroneously placed residues in the density, as reviewer #2 pointed out, can be deceiving in further interaction/mutation analysis based on such model.

Moreover, the fact that there are models from even lower resolution maps, which were deposited in PDB with non-truncated side chains (quite some years ago) should not be a justification to continue to submit possibly incorrect models in the database. Furthermore,

the model mentioned by the authors as an example (PDB ID: 5FUR) has an extremely high clash score, that should be enough to discourage any other investigator from using it because of the high probability of being very wrong.

If the authors really want to submit the full model maybe they can try to add it as a supplemental material to the paper.

Author Rebuttal, third revision:

Reviewer #2:

Remarks to the Author:

The authors have addressed the significant side chain clashes in their vimentin tetramer and filament models, reducing the Molprobity clash scores. The degree of prior clashes was not suitable for deposition, but it is reasonable to proceed with deposition of the current structure so long as the statement(s) in the manuscript about the resolution limitations of the structure remain. It is essential if the model contains all side chains that the readers understand that accurate visualization of side chain atoms was not and could not be done at 7.2Å resolution.

The authors' state "Truncating the side chains would imply that our model could not be used further by other scientists in applications where the side chains are essential." It is precisely this scenario where I believe caution is needed. Some investigators will use the model as fact and erroneous conclusions might occur when relying on side chain positions, contact residues and distances, surfaces, etc. In the end, though, the rigor of study falls on those investigators and their own reviewers.

I congratulate the authors on a most outstanding structural study.

Christopher G. Bunick, MD PhD
Associate Professor of Dermatology
Program in Translational Biomedicine
Yale University

Reviewer #3:

Remarks to the Author:

I was asked by the editor to comment on the PDB model and this is my response to the comments from Reviewer 2:

Reviewer #2: The pdb files associated with this publication are not appropriate for the resolution of the structure. Viewing of the tetramer and higher order filament in Chimera reveals full side chains are visible. In fact, the filament model shows a lysine side chain penetrating a phenylalanine ring- this occurs in multiple locations. All pdb models of a 7.2Å structure should have the side chains truncated at C-alpha. Otherwise, other investigators may use these models to make conclusions about residue to residue interactions erroneously.

Authors: Since the tetramer model was obtained by molecular dynamics flexible fitting (while embedded in the filament model), the orientations of the side chains are guided by physical principles. The electron density map acts as an external field that directs the model into the map (Trabuco, Villa, Mitra, Frank, & Schulten, 2008). Numerous articles show that models of this type are deposited in the PDB with non-truncated side chains (for example PDB ID code 5FUR as just one illustrative example).

My response: I agree with reviewer #2: models of a 7.2Å resolution (and even below, e.g. PDB ID: 8B5R at 6.10Å) should have the side chains truncated at C-alpha when deposited in PDB. I understand the authors effort of improving the clash score using molecular dynamics (MD), thus simulating the relatively correct positions and interactions of atoms by calculating forces between particles and their potential energies inside the map. The backbone of the ordered domains of vimentin filaments fit correctly in the density and the atomic model is probably also "acceptable" in terms of geometry, as assessed by the low clash score. However, the authors cannot claim that the side chains fit and the registry is correct in the context of this map, due to the limited resolution. In PDB, it is not allowed to deposit MD-only based models, without map. Therefore, I believe that depositing an MD model simulated in a low-resolution map, and therefore prone to bias and overfitting, is not a good scientific practice. The main concern here is that erroneously placed residues in the density, as reviewer #2 pointed out, can be deceiving in further interaction/mutation analysis based on such model.

Moreover, the fact that there are models from even lower resolution maps, which were deposited in PDB with non-truncated side chains (quite some years ago) should not be a justification to continue to submit possibly incorrect models in the database. Furthermore, the model mentioned by the authors as an example (PDB ID: 5FUR) has an extremely high clash score, that should be enough to discourage any other investigator from using it because of the high probability of being very wrong.

If the authors want really want to submit the full model maybe they can try to add it as a supplemental material to the paper.

We thank the reviewers for the constructive review process. As suggested, we modified the PDB deposition of the VIF filament model (PDB-8RVE) and truncated the side chains at C-alpha position (please see attached PDB validation report D_1292136296_val-report-full-annotate_P1.pdf and model file D_1292136296_model-review_P1.cif).

Final Decision Letter:**Message:** 1st Mar 2024

Dear Dr. Medalia,

We are now happy to accept your revised paper "Vimentin filaments integrate low complexity domains in a complex helical structure" for publication as an Article in Nature Structural & Molecular Biology.

Your paper will be published online soon after we receive proof corrections and will appear in print in the next available issue. You can find out your date of online publication by

contacting the production team shortly after sending your proof corrections.

Please note that *Nature Structural & Molecular Biology* is a Transformative Journal (TJ). Authors may publish their research with us through the traditional subscription access route or make their paper immediately open access through payment of an article-processing charge (APC). Authors will not be required to make a final decision about access to their article until it has been accepted. Find out more about Transformative Journals

Sincerely,

Katarzyna Ciazynska, PhD
(she/her)
Associate Editor
Nature Structural & Molecular Biology
<https://orcid.org/0000-0002-9899-2428>